# Effects of HSP70 chaperones Ssa1 and Ssa2 on Ste5 scaffold and the mating mitogen-activated protein kinase (MAPK) pathway in *Saccharomyces cerevisiae*

**Francis W. Farley¤a, Ryan R. McCully¤b, Paul B. Maslo¤c, Lu Yu¤d, Mark A. Sheff¤e, Homayoun Sadeghi, Elaine A. Elion** *

Department of Biological Chemistry & Molecular Pharmacology, Harvard Medical School, Boston, MA, United States of America

¤a Current address: Somerville, MA, United States of America
¤b Current address: L.E.V. Consulting, London, United Kingdom
¤c Current address: Quinn Emmanuel Urquhart & Sullivan, LLP, Dallas, TX, United States of America
¤d Current address: AstraZeneca, Academy House, Cambridge, United Kingdom
¤e Current address: Greenlight Biosciences, Metabolic Engineering Group, Medford, MA, United States of America
* elaine_elion@hms.harvard.edu

**Data Availability Statement:** All relevant data are within the paper and its supporting information files.

## Abstract

Ste5 is a prototype of scaffold proteins that regulate activation of mitogen-activated protein kinase (MAPK) cascades in all eukaryotes. Ste5 associates with many proteins including Gβγ (Ste4), Ste11 MAPKKK, Ste7 MAPKK, Fus3 and Kss1 MAPKs, Bem1, Cdc24. Here we show that Ste5 also associates with heat shock protein 70 chaperone (Hsp70) Ssa1 and that Ssa1 and its ortholog Ssa2 are together important for Ste5 function and efficient mating responses. The majority of purified overexpressed Ste5 associates with Ssa1. Loss of Ssa1 and Ssa2 has deleterious effects on Ste5 abundance, integrity, and localization particularly when Ste5 is expressed at native levels. The status of Ssa1 and Ssa2 influences Ste5 electrophoresis mobility and formation of high molecular weight species thought to be phosphorylated, ubiquitinylated and aggregated and lower molecular weight fragments. A Ste5 VWA domain mutant with greater propensity to form punctate foci has reduced predicted propensity to bind Ssa1 near the mutation sites and forms more punctate foci when Ssa1 Is overexpressed, supporting a dynamic protein quality control relationship between Ste5 and Ssa1. Loss of Ssa1 and Ssa2 reduces activation of Fus3 and Kss1 MAPKs and *FUS1* gene expression and impairs mating shmoo morphogenesis. Surprisingly, *ssa1*, *ssa2*, *ssa3* and *ssa4* single, double and triple mutants can still mate, suggesting compensatory mechanisms exist for folding. Additional analysis suggests Ssa1 is the major Hsp70 chaperone for the mating and invasive growth pathways and reveals several Hsp70-Hsp90 chaperone-network proteins required for mating morphogenesis.

**Funding:** The National Institutes of Health (N.I.H.) grant RO1 GM46962 from the government of the United States funded salaries and research supplies for Elaine Elion, Francis Farley, Ryan McCully, Paul Maslo, Lu Yu, Mark Sheff and Homayoun Sadeghi either directly or indirectly through tenure transfer funds and provided indirect cost funding for Harvard University. Once N.I.H. government funding ended Elaine Elion was supported by salary support from Harvard University but was required by her Department to pay out of own pocket for cell phone, internet connection, computers, software, computer accessories, printer, cleaning supplies, PPE, and supplies relating to writing and notebook keeping. The funders had no role in study design, data collection and analysis, decision to publish, or preparation of the manuscript.

**Competing interests:** The authors have declared that no competing interests exist.

## Introduction

The mating pathway in *S. cerevisiae* involves differentiative events that include cell cycle arrest in G1 phase, specialized polarized growth towards a pheromone gradient, cell-cell attachment and fusion to form a zygote that reenters the cell cycle (Fig 1A) [1–7]. These events are mediated by a conserved receptor-G protein-coupled MAPK cascade that activates MAPKs Fus3 and Kss1, with Fus3 being more critical [5–7]. Ste5, the first described MAPK scaffold/tether [8], has multiple functions in the mating pathway and is essential for activation of Fus3 and Kss1 [9–12]. During mating in a cells, α factor pheromone binds to dimers of Ste2 (a type D GPCR) [13], which liberate bound Gβ-Ste4/Gγ-Ste18 dimers from Gα-Gpa1 subunits [7,8]. Ste4/Ste18 then binds to and recruits a Cdc42-activated form of PAK Ste20, a MAPKKKK. Ste5 also binds the liberated Gβ/Ste4-GγSte18 dimers and four protein kinases, MAPKKK Ste11, MAPKK Ste7 and MAPK Fus3 and MAPK Kss1, with MAPKK Ste7 bridging and enhancing MAPK association (Figs 1A and S1A [1–13]). Ste5 orchestrates Ste20 activation of Ste11, Ste11 activation of Ste7, Ste7 activation of Fus3 and auto-activation of Fus3 [1,4–7,9,10,14] potentially through cis and trans kinase phosphorylation events on dimers or oligomers of Ste5 [12]. These many functions are mediated through Ste4 interactions, conformational changes, oligomerization and interactions with Cln2/Cdc28, phosphatases, and the Bem1 complex that includes Cdc42 Rho-type GTPase and Cdc24 (Dbl guanine nucleotide exchange factor for Cdc42) [4–7,15–19]. Ste5 also regulates transcription in association with Fus3 [20,21], and may have direct roles in cell polarization and cell fusion [1,3,6,7,22–25]. Fus3 and Kss1 phosphorylate numerous targets required for mating that include transcription factors and repressors (e.g. Ste12, Dig1, Dig2 and others), cytoskeletal targets such as the actin cable nucleator Bni1 and the Far1 cyclin dependent kinase inhibitor (CDKI) that also binds GbGg (Ste4/Ste18), Bem1 and Cdc24 and regulates shmoo formation [1–7,20]. Many of the mating pathway components are used in the invasive growth pathway which is activated by low pheromone concentrations, poor nutrients, and cell wall stress i (Fig 1A) [26–29]. Invasive growth (pseuodohyphal growth in diploids) is activated by plasma membrane mucins linked to Cdc42, Ste20, Ste11, Ste7 and Kss1. Kss1 regulates transcriptional repressors Dig1, Dig2 and transcription activator Ste12 (and represses Ste12 as an inactive kinase via Dig1 [5]), in addition to Tec1 and other transcription factors which together promote invasive growth (Fig 1A). Multiple mechanisms downregulate the mating and invasive growth pathways including degradation of Ste7, Ste11 and Ste12 [26–29]. The mating and invasive growth/pseudohyphal pathways also regulate virulence of pathogenic yeast (e.g. *S. cerevisiae*, *C. glabrata*) that infect humans and crops [28,29].

Scaffold proteins play crucial roles in transmitting signals through MAPK cascades to downstream targets in eukaryotes [4,8,11,12,30,31]. Ste5 is regulated at multiple levels to ensure differentiation occurs at the right time and place. Tight control of the conformation, localization and abundance of Ste5 regulates signaling specificity, proper timing and amplitude of MAPK activation and prevents deleterious misactivation of the mating pathway. These controls include assembly of Ste5 into an active dimer (and possibly higher order oligomers; 12,15,17), relief of autoinhibitory interactions between PH and VWA domains [7,12,16,17], membrane lipids (i.e. phosphatidylinositol-(4,5)-bisphosphate and fatty acids) that bind Ste5 and enhance its interaction with Gβγ [7,32,33] and at least 16 phosphorylation events including overlapping phosphorylation by Fus3 and Cln/Cdc28 that influence Ste5 residency at the plasma membrane and abundance ([34], https://phosphogrid.org/sites/32160). Ste5 is spatially controlled through nucleocytoplasmic shuttling which imports a nuclear pool for transcription, limits the amount of Ste5 available to bind to Gβγ and also exports Ste5 for binding to Gβγ during pheromone signaling [1,35–37]. During vegetative growth, high phosphorylation

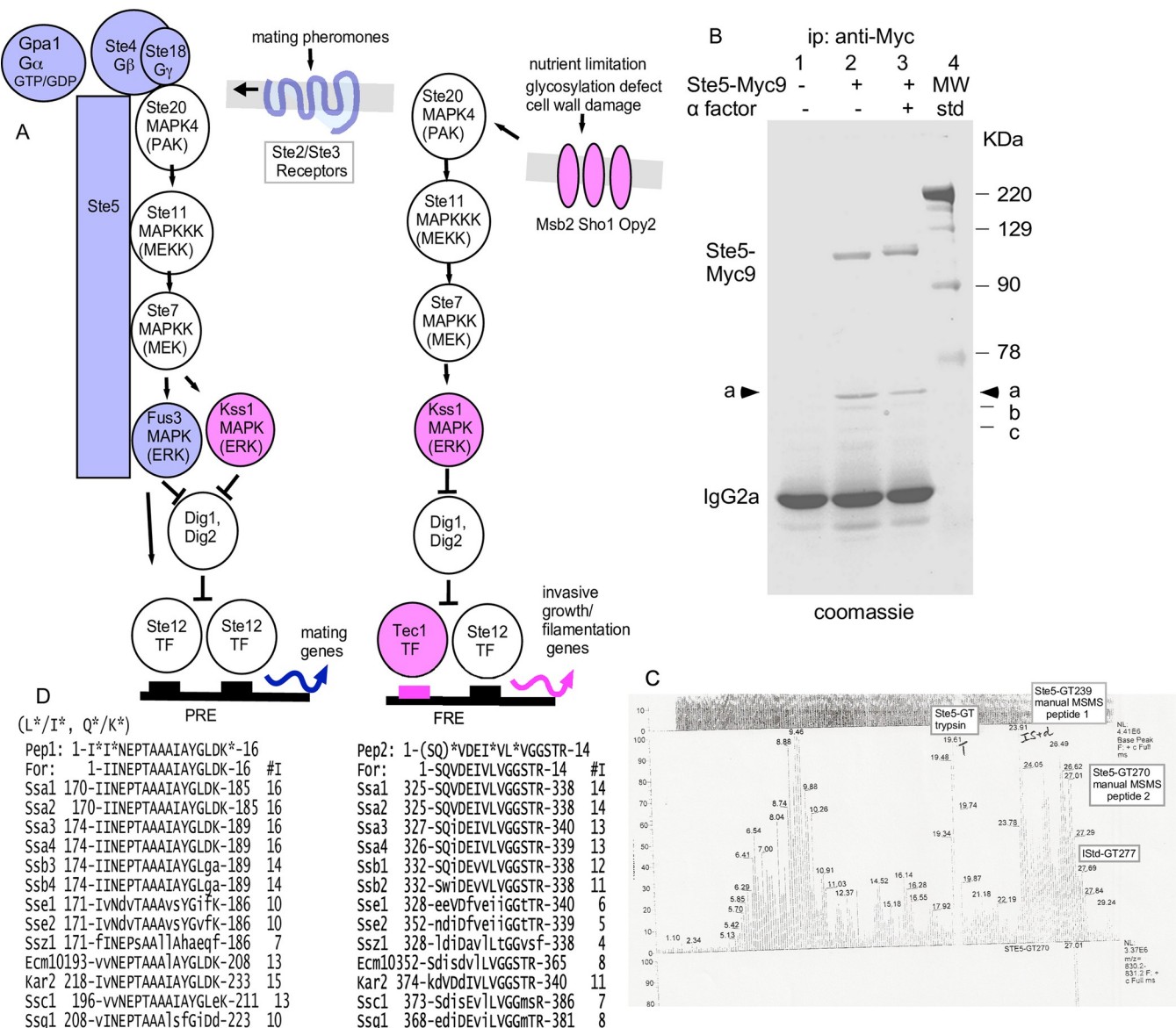

**Fig 1. The majority of immunoprecipitated Ste5-Myc9 is bound to Hsp70 Ssa1 and possibly Ssa2.** A. Schematics of mating pathway and invasive growth pathway MAPK cascades and Ste5 interactions. B. Ste5-Myc9 co-immunoprecipitates (co-IPs) a ~70 KDa protein(s). Coomassie stain of an SDS-polyacrylamide gel of Ste5-Myc9 IP'd with 9E10 monoclonal antibody from yeast whole cell extracts prepared from mitotically dividing EY1775 (*MATa ste5 bar1*) cells expressing *STE5-MYC9-2μ* (pSKM19). Lane 1: EY957 (STE5) cells, lanes 2,3: EY1775 expressing STE5-MYC9 (pSKM19). "+" indicates treatment of cells with 50 nM a factor for 2 hours. C. HPLC profile of tandem mass spectra for the 70 kD protein gel band after treatment with trypsin. The HPLC profile of the LCMSMS base peak trace is from time 0 to time 30 minutes for the tryptic digest of the 70 kDa band labeled as STE5-GT by Harvard Microchemistry. The LCMSMS peaks at times 23.91 minutes (STE5-GT239) and 27.01 minutes (STE5-GT270) were manually picked and subjected to manual MSMS interpretation. The T at peak 19.61 minutes is an autolyzed tryptic artefact. This analysis was performed in 1997. D. The 70kDa protein is most homologous to Ssa1 and Ssa2. Two peptide sequences from the 70 kDa band in A are aligned with *S. cerevisiae* HSP70 proteins. The "*" indicates residue uncertainty, *L/I and *Q/K.

of Ste5 by Cln/CDK adds negative charges that disrupt membrane localization of Ste5 and reduces its stable binding to Gβγ [24,34]. The phosphorylation of Ste5 also triggers its ubiquitinylation and degradation through SCF$^{Cdc4}$ and the proteasome in the nucleus [37]. During pheromone signaling, Ste5 abundance increases through Fus3 and Kss1 [38], in part through inhibition of Cln/Cdk by Far1, a CDK inhibitor. Loss of Fus3 causes enhanced, depolarized accumulation of Ste5 at the plasma membrane [24] whereas loss of both Cdc28 and Fus3

restores polarized localization of Ste5 at the plasma membrane but also causes Ste5 to accumulate in vesicles that that are likely vacuolar [24], possibly because of loss of solubility derived from phosphorylation. Here, we present evidence of novel requirements for Hsp70 chaperones Ssa1 and Ssa2 for Ste5 stability and integrity.

The heat shock 70 (Hsp70) proteins regulate evolutionarily conserved responses of all cells to proteotoxic stress [39–44]. Their misregulation is associated with numerous diseases including cancer [45–47], Alzheimer's Disease [48], Parkinson's Disease [48], and fungal infections [29,49–51]. S. cerevisiae has 14 Hsp70 proteins of which 9 are cytosolic and nuclear. The "classic" cytosolic Hsp70s are Ssa1-Ssa4 in S. cerevisiae and Hsc70s in metazoans and mammals [51]. Ssa1-Ssa4 function with J-domain proteins/Hsp40s (i.e. Ydj1, Sis1) and nucleotide exchange factors (e.g. Fes1/HBp1) [39–44,52] and bind substrates on surface exposed hydrophobic stretches with certain characteristics [39,53]. Ssa1 and Ssa2 are expressed constitutively through heat shock transcription factor 1 (HSF1, 54) and function at cool and ambient temperatures. Heat stress increases the level of Ssa2 but not that of Ssa1. Much of the heat shock response also relies on Msn2/Msn4 transcription factors which modulate membrane composition, carbohydrate flux, and cell wall integrity independently of Hsf1, Ssa1 and Ssa2 [54,55].

The classic human and S. cerevisiae Hsp70 proteins (Ssa1, Ssa2, Ssa3, Ssa4) are together essential for viability and have major functions in folding nascent translated polypeptides, refolding of misfolded proteins and disaggregation of protein aggregates in cooperation with specific co-chaperones [39–44,56–58]. They deliver proteins to translocation machinery at organellar membranes, and manage irretrievably damaged and misfolded proteins (including aggregated proteins, prions and RNA stress granules) by facilitating their degradation by the proteasome or by the lysosome or autophagy [39–41,59,60]. Ssa1 and Ssa2 have many functional similarities and subtle functional differences [40,51,55]. Major functions of Ssa1 and Ssa2 include assisting the ribosome-associated Hsp70s Ssb1 and Ssb2 to fold emerging polypeptides during translation [57,58] and transfer polypeptides to other chaperones such as Hsp90s (i.e. Hsp82 and Hsc82 in S. cerevisiae) which also help fold proteins, into their mature functional forms and collaborate with Hsp70s [61,62]. Ssa1 and Ssa2 help translocate proteins through endoplasmic reticulum and mitochondrial membranes [61,62]. They promote degradation of misfolded and damaged proteins and impact various types of inclusion bodies [63–67], and promote ER-associated degradation of ER proteins in cytosolic proteasomes (ERAD; [68], proteasome degradation of ubiquitinylated and nonubiquinylated proteins [65–67], degradation in vacuoles and autophagosomes [39,68,69]. Ssa1 and Ssa2 also facilitate nuclear import [70,71], assembly and disassembly of multi-protein complexes (e.g. prions [72], proteasomes [73], clathrin coats [74], and can be essential subunits of enzyme complexes (e.g. Ssa1/Ssa2/Rad9/Rad53 checkpoint complex [75].

Despite a multitude of functions, obvious ssa1, ssa2, ssa3 and ssa4 mutant phenotypes are rare; most work has focused on Ssa1. For example, whole cell extracts from a ssa1 null mutant in one study had no obvious differences in individual proteins by 2d gel electrophoresis, [76]. More obvious phenotypes have been seen when both SSA1 and SSA2 are deleted, although pulse labeling experiments indicate small (~3%) global differences in protein degradation rates [77]. The expression of SSA3 and SSA4 is elevated in an ssa1D ssa2D mutant, however, SSA3 and SSA4 cannot substitute for SSA1 and SSA2 and neither can Hsf1 [40,78,79].

Little is known about the role of Ssa proteins during mating. To date, only one mating pathway protein, Dig1, has been categorized as a candidate substrate for Ssa1 in cells treated with α factor mating pheromone, compared to 317 from vegetatively growing cells in the same study [80]. Here we report the purification of Ssa1 with Ste5 during vegetative growth and α factor stimulation. Ssa1 is predominantly cytoplasmic but also accumulates in nuclei. Ssa1 and Ssa2 together enhance Ste5 abundance, integrity and localization in the nucleus and at the plasma

membrane. Ssa1 and Ssa2 prevent degradation of Ste5 at specific proteolytic sites, especially as temperatures rise, and they influence antibody recognition of N- and C-terminal tags on Ste5. By contrast, Ssa1 and Ssa2 may downregulate Fus3, and have no obvious effect on the level of Tcm1, Ste7 or Kss1 by immunoblot analysis. Overexpression of Ssa1 and loss of Ssa1 and Ssa2 influences the localization of wild type and mutant forms of Ste5. Ssa1 and Ssa2 are required for efficient activation of the mating MAPK cascade, *FUS1* expression, and shmoo morphogenesis. Despite these many effects, loss of Ssa1 and Ssa2 does not block mating, suggesting strong compensatory mechanisms exist to tolerate reduced protein quality control. Analysis of other Hsp70 network mutants supports the existence of compensatory mechanisms.

## Materials and methods

### Yeast strains and media

See S1 Table for a list of the yeast strains and plasmids used in this study. Yeast strains were propagated and stored following standard procedures [81–83]. Null deletion alleles of *ssa1*, *ssa2*, *ssa3 and ssa4* were used for all analyses. If no allele is specified, then it is a null deletion allele (indicated as a "*D*" or a "*D*" as in *ste5D and ste5D*). *S. cerevisiae* and *E. coli* strains were grown in standard selective media (termed SC) and yeast extract peptone (YEP) media following standard protocols [81,82]. Cells with *GAL1prom*-driven genes were pre-grown in media containing 2% raffinose before induction in medium containing 2% galactose. Strains were grown at room temperature and 30˚C unless noted otherwise. Mating type was determined by a qualitative patch mating assay and by halo assay tests of a factor and a factor sensitivity [83]. Colony size was measured by enlarging the image of the petri plate to a diameter of 240 mm, measuring 50 colonies per strain and determining mean value with standard error. Previously constructed *ste5* mutant proteins [16,23,35,36] were screened for aberrant localization or aggregates of Ste5-Myc9 or Ste5-GFP2 with or without excess Ssa1, including Ste5K49,50A-Myc9 (EY3398), Ste5K64-66A-Myc9 (EY3399), Ste5K49,50,64-66A-Myc9 (EY3400), Ste5(1–242)-GFP2 (EY3401), Ste5(1-242D49-66)-GFP2 (EY3402), ste5L482/485A-Myc9 (pYMW37/ aka ste5NES1mII/EYL2778/YMY533), ste5D522-527-Myc9 (pYMW5/ aka steNES2D/ EYL2779/YMY534), ste5L633/636A-Myc9 (pYMW15/ aka ste5NES3mII/ EYL2780/YMY535), ste5L610A/614A/634A/637A-Myc9 (pYMW109/ aka ste5NES4mIINES3mII/ EYL2781/ YMY536). Yeast and bacterial transformations were performed as described [84,85] except for temperature sensitive strains which had the following protocol modifications: strains were not heat shocked, more DNA was used for transformation, and strains were recovered for 20 minutes in rich medium before pelleting, and resuspending in selective medium for plating. Some *ssa*‾ mutant strains were generated by sporulation of strain MW63 (EYL349) *MATa/MATa ssa1::HIS3/SSA1 ssa2D::LEU2/SSA2 SSA3/ssa3::TRP1 SSA4/ssa4::URA3 ura3-52/ura3-52 his3/ his3 leu2-3,112/leu2-3,112, trp1-1/trp1-1 his3/his3* diploid, (kind gift of E. Craig, University of Wisconsin) followed by phenotypic analysis of ascospore segregants. Backcrossing was done to make strains more congenic. To permit introduction of plasmids harboring a *URA3* gene for selection we isolated spontaneous mutations in the *URA3* gene in *ssa4::URA3* and other *URA3* strains by resistance to 5-fluoro-orotic acid and subsequent complementation tests followed by tests of reversion to Ura+ and confirmation of mitochondrial function by growth on YPD-2% glycerol plates. *MATa bar1D:: KAN^R* derivatives of EY3136 and EY3141 (S1 Table) were constructed by first transforming them with pRS316 (*URA3-CEN4* plasmid), then mating Ura+ transformants with Research Genetics BY4742 *MATa bar1D0::KAN^R* (S288c strain background that is enriched in the Craig lab strains) at room temperature in patches on YPD plates, then sporulating and dissecting tetrads, then testing ascospores for all auxotrophies, mating type and resistance to hygromycin. *MATa bar1D0::KAN^R ssa1::HIS3 ssa2::LEU2* ascospores

were not recovered out of ~40 tetrads dissected. Based on deducing the genotype of dead asco-spores among tetrads, we suspect that the *MATa bar1D0::KANR ssa1::HIS3 ssa2::LEU2* asco-spores germinate poorly. Two *MATa bar1D:: KAN^R ssa1D::HIS3 ssa2D::LEU2* ascospores were identified and switched to *MATa* using a *GAL1p-HO* plasmid, then cured of the *URA3-CEN* plasmids and purified by streakouts (S1 Table, EYL1797 and EYL1798). We found that the EY3136 and EY3141 ascospore strains originating from the Craig lab diploid MW63 lacked Kss1 based on the absence of immunoblot signals with anti-Kss1 and anti-active MAPK anti-bodies. *KSS1+* derivatives of EY3136 and EY3141 were identified among the *bar1D::*K*AN^R* ascospores from crosses to S288c BY4742 which has a functional *KSS1* gene.

## Antibodies

The antibodies used in this study included: Mouse monoclonal 9E10 and mouse monoclonal 12CA5 (Harvard University antibody facility), anti-active phosphorylated p42p44 MAPK rab-bit polyclonal antibody (Cell Signaling Technology Phospho-p44/42 MAPK(Erk1/2) (Thr202/Tyr204) #9101), anti-Fus3 peptide rabbit polyclonal antibody [14,20], anti-Kss1 peptide rabbit polyclonal (6775-kss1-yc-19, Santa Cruz Biotechnology, Inc.), mouse monoclonal anti-Tcm1 antibody (gift of J. Warner, Albert Einstein College of Medicine). The anti-phospho-p42p44 antibody recognizes the TpEYpWYRAPE motif and was made against a peptide from ERK2 that overlaps residues 220–247 and is 92.95% identical to human ERK1 residues 237–264 (26/28 residues identical), 53% identical to Fus3 residues 167–192 (15/28 identical) and 57% iden-tical to Kss1 residues 170–195 (16/28 identical).

## Identification of 70 kd protein

Ste5-Myc9 was expressed from its own promoter on a multicopy *2*μ plasmid pSKM19 in EY1775 *MATa bar1D ste5D* cells in SC-uracil selective medium with 2% dextrose. Cells were grown and whole cell extracts prepared as described [86]. In Fig 1B, Ste5-Myc9 was immuno-precipitated (IP'd) with mouse 9E10 monoclonal antibody from 10 mg of yeast whole cell extracts prepared using liquid nitrogen and grinding with a mortar and pestle to break the cells as in [86]. The buffer had 20 mM Tris-HCl pH 7.2 (Fisher), 125 mM potassium acetate (Fisher), 0.5 mM EDTA (Sigma), 0.5 mM EGTA (Sigma), 0.1% Tween 20 (Sigma), 1 mM DTT (BioRad), 20 mM 12.5% glycerol (Fisher), 10 μg/ml pepstatin (Sigma), 1mM benzenesulfonyl-fluoride (Sigma), 1 mM sodium ortho-vanadate (Sigma), 25 mM β-glycerophosphate (Sigma). The 70kd Ssa1, Ssa2 proteins were isolated by electrophoresis on an SDS polyacrylamide gel followed by staining with colloidal coomassie blue (Invitrogen). The 70kd band was excised from the gel and subjected to reduction, carboxyamidomethylation and in-gel digestion with trypsin. Recovered peptides were separated by microcapillary reversed-phase high-pressure liquid chromatography (HPLC) and analyzed using a Finnegan LCQ qudadrupole ion trap mass Spectrometer by 3D-ion trap tandem mass spectrometry (LC-MS/MS)(by Eric Spooner and Daniel P. Kirby of Harvard Microchemistry). Two peptide sequences were identified: 1-I*I*NEPTAAAI*AYGL*DK*-16 (peptide 1 theoretical monoisotropic mass 1458.77, experi-mental monoisotropic mass 1458.49) and 1-(SQ)*VDEI*VL*VGGSTR-14 (peptide 2 theoreti-cal monoisotropic mass 1658, experimental monoisotropic mass 1658.69) (Datafile 0719yfse5-gt.dat) with qualifications indicated as an asterisk. Qualification in the sequence determination from MSMS spectra are necessary due to indistinguishable molecular mass of pairs of amino acids (the isobaric pairs are Leu/Ile MW 113, or Gln/ Lys MW 128 and Phe/oxyMet MW 148) which is indicated by an asterisk, and from areas of the MSMS spectra that could be solved only by pairs of amino acids, which is indicated by both a parenthesis and an asterisk.

## Whole cell extracts, co-immunoprecipitation and immunoblot analysis

Cells were grown to an $A_{600}$ of 0.8–1 and induced with α factor for 15 minutes where indicated, then pelleted by centrifugation, washed once with ice-cold water, then frozen in dry ice/ethanol. Pellets were thawed on ice and resuspended in 1.0 ml of modified H-buffer described in Elion et al 1993 [20] with varying amounts and types of inhibitors of phosphatases, proteases and sodium chloride. Cell extracts were prepared by glass-bead breakage as described [16,20,86]. Co-IPs were done in the same buffer adjusted to 200 mM NaCl, and immunoblot analyses were carried out as described [16,20,23,86]. The protein concentration of whole cell extracts was determined with the BioRad assay. For detection of Ste7-MYC, 1 mg of *CYC1prom-STE7-MYC* whole cell extracts was concentrated by 40% ammonium precipitation then dissolved in loading buffer prior to immunoblot analysis as described [83]. Co-IPs used 250 µg to 2 mg of whole cell protein extracts, 15 µg of 12CA5, 1–2 µg of 9E10, and 30 µl of protein A agarose beads (Sigma). Samples were subjected to SDS-PAGE and immunoblot analysis as described [16,20,23,82,86,87]. For analysis of Ste5-Myc9 phosphorylation, samples were run on native polyacrylamide gels that were 8% acrylamide, 0.19% bis-acrylamide, 0.75 mM gel thickness and run at 40 mAmps. Benchmark prestained protein ladder (Life Technologies Cat. No.10748-010) was used for many of the gels. We note that several of the original films were accidentally labeled incorrectly with wrong protein ladder sizes taken from the catalog rather than the Gibco Benchmark package insert. The correct package insert molecular weights are in the figures (S1 Appendix). Immunoreactivity in immunoblots was detected with horseradish peroxidase-conjugated secondary antibody with ECL (Amersham, Arlington Heights, IL). The apparent molecular weight of protein bands in gels was determined by measuring the distance of migration of molecular weight markers (Rf) in the same gel, creating a graph of the log10 of the molecular weight of marker proteins versus their Rfs, then extrapolating from the graph a log10 for an Rf of a specific band and then taking the antilog of this value for the band's apparent molecular weight.

The conditions used for each experiment were as follows: In Fig 1B (YF) the extracts were prepared as described under "Identification of 70 kDa protein". In Fig 2A–2H (YMW), Fig 3C–3E (YMW), and Fig 4A (PBM) the modified H buffer was as described in Elion et al, 1993 [20] with 200 mM NaCl and in Fig 2A–2H (YMW) the immunoprecipitation buffer had 150 mM NaCl. In Fig 3A and 3B (HS), the modified H buffer lacked NaCl and contained 0.05% PMSF, 50 mM NaN₃, 0.02 mM EDTA, 0.025 mM each meta- and ortho-vanadate (pH 7 stock solution), 0.05 mM NaF, a 1:100 dilution of phosphatase inhibitor cocktail 1 (Sigma P2850 microcystin LR, cantharidin and (-)-p-bromotetramisole), 50 mM benzamidine, and 50 µM PSI proteasome inhibitor [Z-Ile-Glu(OBut)-Ala-Leu-H (aldehyde, Peptide International, IAT-3169-v). The PSI proteasome inhibitor is known to increase the abundance of phosphorylated and ubiquitinylated proteins. In Fig 4B–4D (FWF) and Fig 5B–5F (FWF), the modified H buffer contained 1.5 mM NaCl, 10% glycerol, 25 mM Tris-HCl pH 7.4, 15 µM MgCl₂, 15 mM EGTA, 1 mM DTT, 0.1% Triton-X-100, 1 mM NaN₃, 0.25 mM each of meta- and ortho-vanadate (pH 7 stock solution), ~0.5 mg/ml phenylmethylsufonyl fluoride (PMSF, Sigma), 5 mg/ml pepstatin-A (Sigma), 5 mg/ml chymostatin (Sigma), 5 mg/ml leupeptin (Sigma), 5 mg/ml antipain (Sigma), 1 mM benzamidine (Sigma) and 50 µM PSI proteasome inhibitor [Z-Ile-Glu(OBut)-Ala-Leu-H (aldehyde, Peptide International, IAT-3169-v). In Fig 4E–4K (RRM) and Fig 8A–8F (RRM), the modified H buffer was as described in Elion et al 1993 [20] with 150 mM NaCl and 50 µM PSI proteasome inhibitor [Z-Ile-Glu(OBut)-Ala-Leu-H (aldehyde, Peptide International, IAT-3169-v). In Fig 6A–6F (RRM), the modified H buffer was as in Elion et al, 1993 [20] with 150 mM NaCl. In Fig 9A and 9B (FWF), the extraction buffer conditions were exactly as described in Farley et al 1999 [83]. In Fig 9C and 9D (RRM)

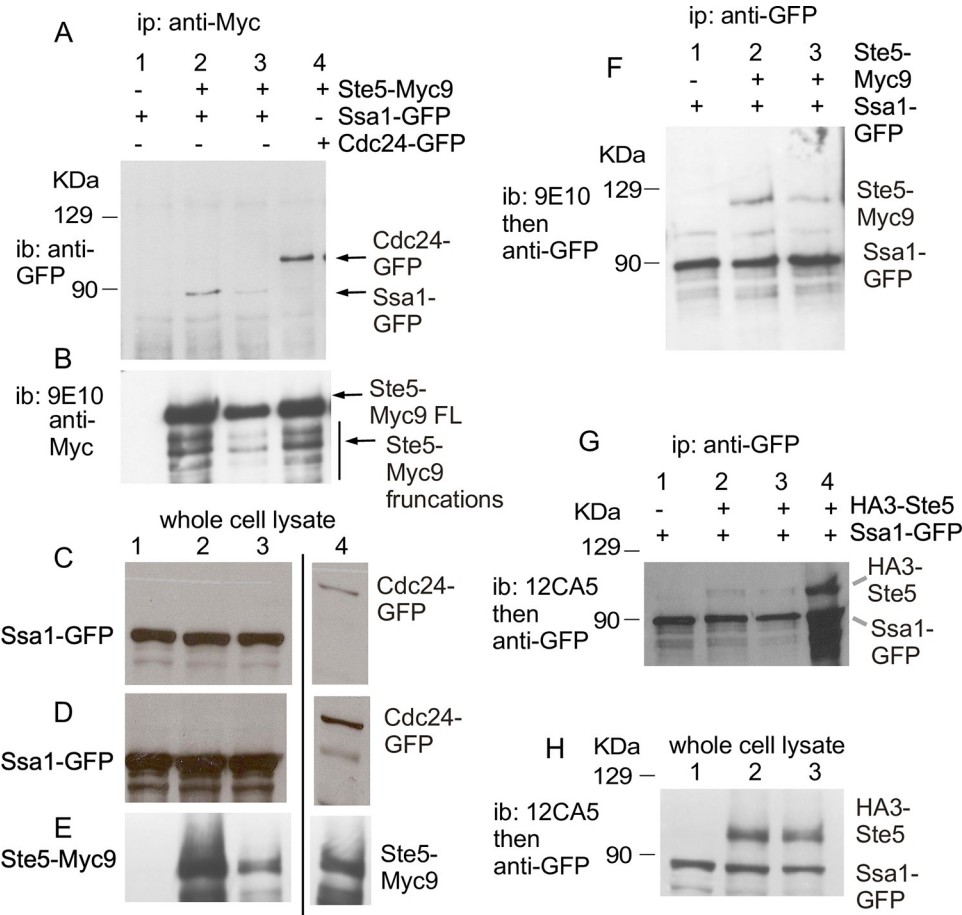

**Fig 2. Ste5 co-immunoprecipitates with Ssa1-GFP and vice versa.** A-B. Ste5-Myc9 co-IPs(co-IPs) Ssa1-GFP and *vice versa*. Co-IPs were done on whole cell lysates (extracts) prepared from EY1775 cells expressing STE5-MYC9 (pSKM90) with or without SSA1-GFP (pYMW122). Ste5-Myc9 was IP'd with 9E10 monoclonal antibody and Ssa1-GFP was IP'd with anti-GFP polyclonal antibodies. A control IP was done on extracts with Ste5-Myc9 co-expressed with GFP-Cdc24. lanes 2 and 3 are different plasmid transformants expressing different levels of Ste5-Myc9. C-E. Immunoblots of whole cell extracts of strains in A. F. IPs of Ssa1-GFP with Ste5-Myc9. Anti-GFP polyclonal antibodies were used to IP Ssa1-GFP and blots were probed with either anti-GFP polyclonal antibodies or 9E10 to detect Ste5-Myc9. G. Co-IPs of Ssa1-GFP with HA3-Ste5. HA3-Ste5 was detected with 12CA5 monoclonal antibody. H. Immunoblot of whole cell lysates from strains in D.

the modified H buffer was exactly as in Elon et al 1993 [20] with 250 mM NaCl in the breakage buffer and the kinase assay done as described in Elion et al 1993 [20].

## Densitometry

The band intensities of immunoblots were measured using either ImageJ software (ImageJ 1.50i, Wayne Rasband, National Institutes of Health, U.S.A. http://imagej.nih.gov/ij) or Adobe Photoshop CS5 (Version 12.0 x64). TIFFs of gels were prepared, all 300 resolution, 8 bit, all made without image contrast adjustment or tiff compression, and analyzed using ImageJ software from the N.I.H. and associated information in addition to European Union methods from Certus Technology. For each band, all lanes within boxes for area plots were cropped with the line tool and area was measured with the wand tool. Average background (calculated from several readings) was subtracted. Two to three readings were made for each

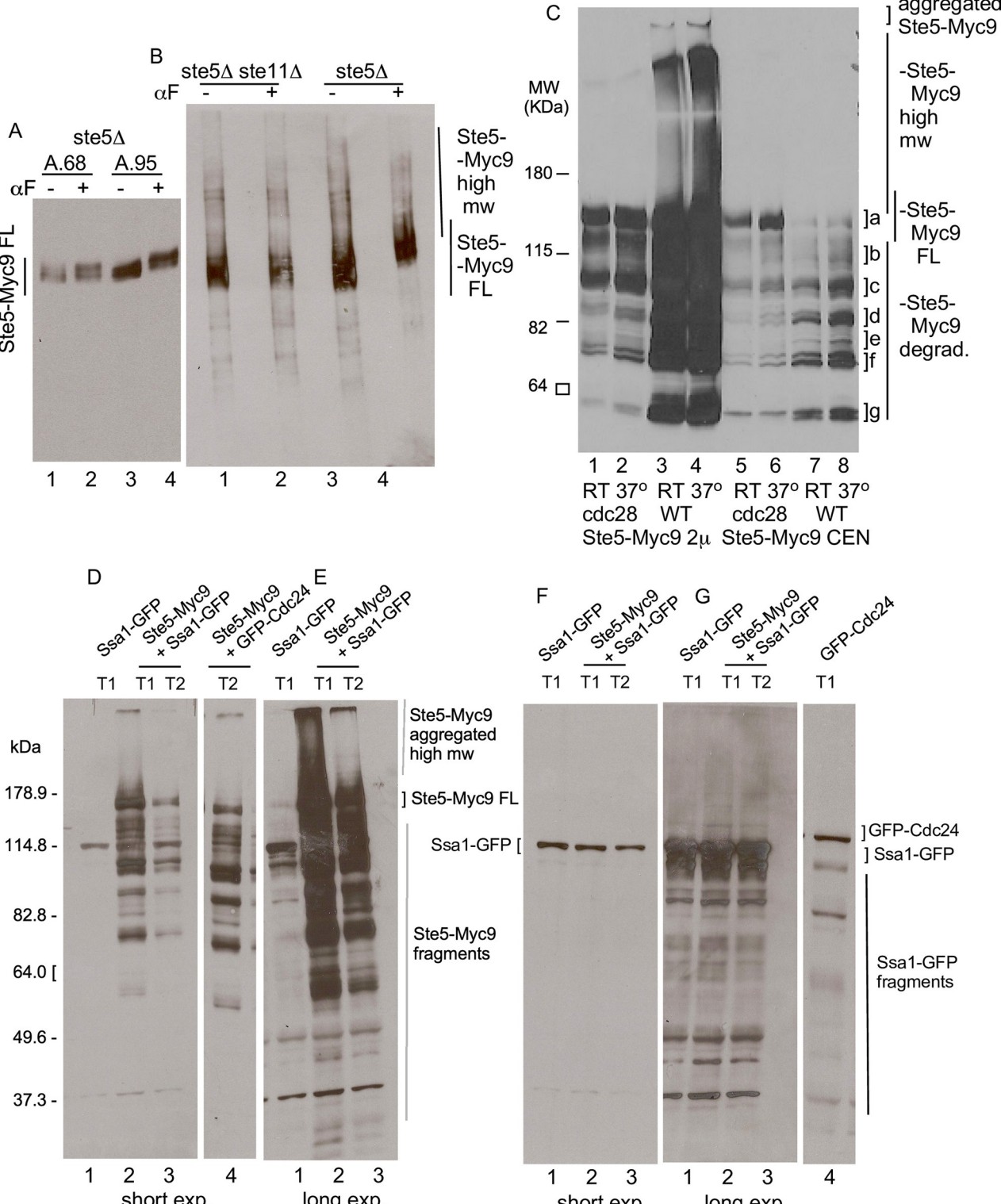

**Fig 3. Ste5-Myc9 migrates in polyacrylamide gels as full-length, high molecular weight, aggregated and truncated forms of varying proportions in *ste5D*, *ste5D ste11D*, *cdc28-4* and *SSA1-GFP*[OP] strains.** A. SDS-PAGE of whole cell extracts of Ste5-Myc9 (2m) in *bar1D ste5D* (EY1775) treated without and with a factor, using a modified H-buffer for extract preparation that has many protease and phosphatase inhibitors and no added NaCl. B. Native PAGE of Ste5-Myc9 (2m) in *bar1D ste5D* (EY1775) and *bar1D ste5D ste11D* without and with a factor. The cell extraction buffer is the same as in A but approximately twice as much extract was loaded on the gel. C. The *cdc28-4* mutation reduces high molecular weight and aggregated forms

of Ste5-Myc9. Ste5-Myc9 was expressed from *2μ* (pSKM19) and *CEN* (pSKM12) plasmids in W303a *CDC28 (*EY957) and *cdc28-4* (PY1236) strains grown at room temperature and after 3 hours at 37˚C and whole cell extracts prepared by standard procedures. Lanes 1,2: *cdc28-4 STE5-M9-2*μ RT, 37˚C. Lanes 3, 4: WT *STE5-MYC9-2*μ RT, 37˚C. Lanes 5,6: cdc*28-4 STE5-MYC9-CEN* RT, 37˚C. Lanes 7,8: WT *STE5-MYC9-CEN* RT, 37˚C. D-E. Ste5-Myc9 with excess Ssa1-GFP. The full SDS-PAGE gel of whole cell extracts used for co-immunoprecipitation in Fig 2 are shown at short (D) and long (E) exposure times. F-G. The immunoblot in D. was stripped and reprobed for Ssa1-GFP and GFP-Cdc24. with anti-GFP antibodies, F is a short exposure and G is a long exposure. Labels:] aggregated in loading well, |high mw forms, a] full-length,] b-g degraded fragments. The extraction buffers in C-E were as in Elion et al, 1993 [20] with 200 mM NaCl.

measurement. In some instances, when ImageJ software was no longer compatible with updated operating systems, the histogram function of Canvas Draw was used for densitometry on non-adjusted images.

## β-galactosidase assays

The FUS1::ubiYlacZ construct of the Nasmyth, Ammerer and Errede labs uses the backbone pDL1460 [88,89] which has ubiquitin with tyrosine fused to the N-terminus of b-galactosidase which reduces its half-life from >20 hours to 10 minutes and avoids confounding effects of accumulation. The promoter has the *FUS1* UAS fused to the *CYC1* TATA element and yields higher basal expression than the native promoter. Cells were grown overnight at room temperature to an $A_{600}$ of ~0.3, pelleted, and resuspended at an $A_{600}$ of 0.3 in 10 ml aliquots of fresh media. α-factor was added from a concentrated stock (6 mM in methanol) and samples were shaken for 2 hours, pelleted, washed once with ice-cold water and frozen at -80˚C. Pellets were thawed and extracts prepared and quantified exactly as described [83]. Units of beta-galactosidase were calculated by the following formula: Units = $[A_{420}$ x (1.7/0.0045)]/ [time (min) x extract volume (ml) x protein concentration (mg/ml)].

## Extract centrifugation

150 mls of cells were grown to $A_{600}$ of 0.3 at room temperature then pelleted in either a Sorvall RC5B or Sorvall RT 6000B centrifuge at 5,000 rpm for 5 minutes at 4˚C. Pellets were washed once with ice cold sterile doubly distilled water that had been chilled to 4˚C, then they were resuspended in fresh ddH$_2$0, transferred and pelleted again. Pellets were drained and frozen in a dry ice/ethanol bath and stored overnight at -80˚C. After being thawed on ice, 1 ml lysis buffer and glass beads were added to the pellets and the samples were vortexed 5 times for 30 seconds each, chilling on ice between each vortexing. Then 0.4 ml of lysis buffer were added to the samples on ice and they underwent a sixth vortex for 30 seconds. Samples were then spun in the swinging bucket Sorvall RT 6000B centrifuge at 5,000 rpm (3,000 x g) for 10 min at 4˚C. Supernatants were transferred equally to two chilled eppendorf tubes, saving one as a control for total input. The other tube of sample was centrifuged in a refrigerated Eppendorf 5415C centrifuge at 14,000 rpm (16,000 x g) for 15 minutes and then the supernatant was transferred to a new tube. An equal volume of lysis buffer was added to each pellet. Protein concentration was determined and 5x loading buffer added to samples for running on an SDS polyacrylamide gel. The buffer used for extract preparation had 25 mM Tris-Cl pH7.4 (Fisher), 200 mM NaCl (Sigma or Fisher), 15 mM MgCl$_2$ (Sigma or Fisher), 1 mM DTT (BioRad), 15 mM EGTA (Fisher), 0.1% triton-X 100 (Fisher), 10% glycerol (Fisher), 1 mg/ml phenylmethylsulfonyl (PMSF), 4 mM 1,10-phenanthroline (Sigma), 2 mM benzamidine (Sigma), 5 μg/ml leupeptin (Sigma), 5 μg/ml pepstatin A (Sigma), 5 μg/ml chymostatin (Sigma), 5 μg/ml aprotonin (Sigma), 0.25 mM sodium ortho-vanadate (Sigma), 0.25 mM sodium meta-vanadate (Sigma), 50 mM sodium fluoride (Sigma), 1 mM sodium azide (Sigma).

**Fus3 kinase assay.** The EY3136 and EY3141 strains were made *bar1D* and *KSS1+* by crossing them to S288c BY4742 (Research Genetics strain RG1104 *MATa bar1D0::KAN$^R$*). Strains

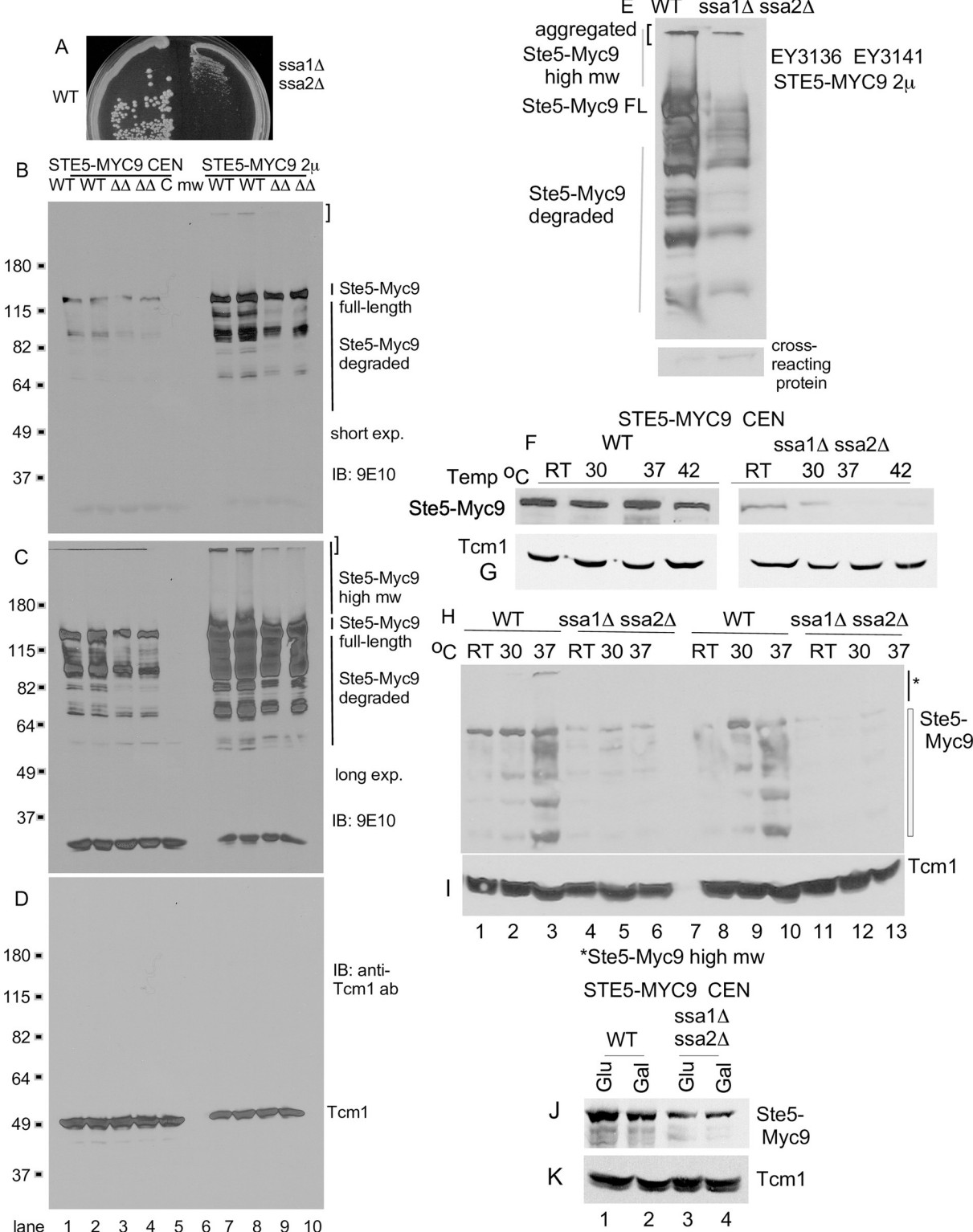

**Fig 4. The abundance of Ste5 is lower in a *ssa1D ssa2D* double mutant.** A. *ssa1D ssa2D* strains grow poorly. Photo of *SSA1 SSA2* (EY3141) and *ssa1D ssa2D* (EY3136) strains on a YPD plate that was incubated at 30°C for 3 days. B-C. Ste5-Myc9 abundance and high molecular weight forms are reduced in a *ssa1D ssa2D* strain. Whole cell extracts were prepared from EY3136 *MATa ssa1::HIS3 ssa2::LEU2* and EY3141 *MATa SSA1 SSA2* strains expressing *STE5-MYC9-CEN-TRP1* (pLSSte5Myc9TRP)(lanes 1–4), *CEN-TRP1* (EB406) (lane 5) or *STE5-MYC9-LEU2-2*μ (pLS40, lanes 7–10). Cells were grown in either SC-tryptophan or SC-leucine medium containing 2% dextrose and

incubated with α factor where indicated. W = WT, DD = *ssa1D ssa2D*. Aggregated, high molecular weight and truncated fragments of Ste5-Myc9 are indicated. B is a short exposure and C is a long exposure. D. Tcm1. The blot in B was stripped and reprobed with a monoclonal antibody to Tcm1 ribosomal protein. E. Retransformation of EY3141 and EY3136 with *STE5-MYC9-2m* again reveals reduced abundance in EY3136 *ssa1*::*HIS3 ssa2*::*LEU2* compared to wild type EY3141. A cross-reacting protein from same blot is shown below (*). F-I. Ste5-Myc9 *CEN* abundance is greatly reduced in an *ssa1D ssa2D* strain especially as temperature increases. Triplicate strains harboring either *STE5-MYC9-CEN* (pSKM12) or *STE5-MYC9-2*μ (pSKM19) were grown at room temperature (~25°C) in SC-uracil medium containing 2% dextrose then shifted to pre-warmed medium at either 30°C, 37°C or 42°C and shaken for 3 hours then pelleted and extracts were prepared. The immunoblots were first probed for Ste5-Myc9 (F, H) then stripped and probed for Tcm1 (G, I). J-K. Comparison of Ste5-Myc9 (J) and Tcm1 (K) levels in *SSA1 SSA2* and *ssa1D ssa2D* strains grown with either 2% dextrose (Dex) or 2% galactose (Gal).

were transformed with pYEE121 (*FUS3-HA5-CEN-URA3*) and pYEE128.30–1 (*fus3K42R-HA5-CEN-URA3*) for kinase assays. Approximately 150 mls of cells were grown to A600 of 0.7–0.8 and processed exactly as described in Elion et al 1993 [20] using an extraction buffer that contained 250 mM NaCl. Transformations and growth were done at room temperature.

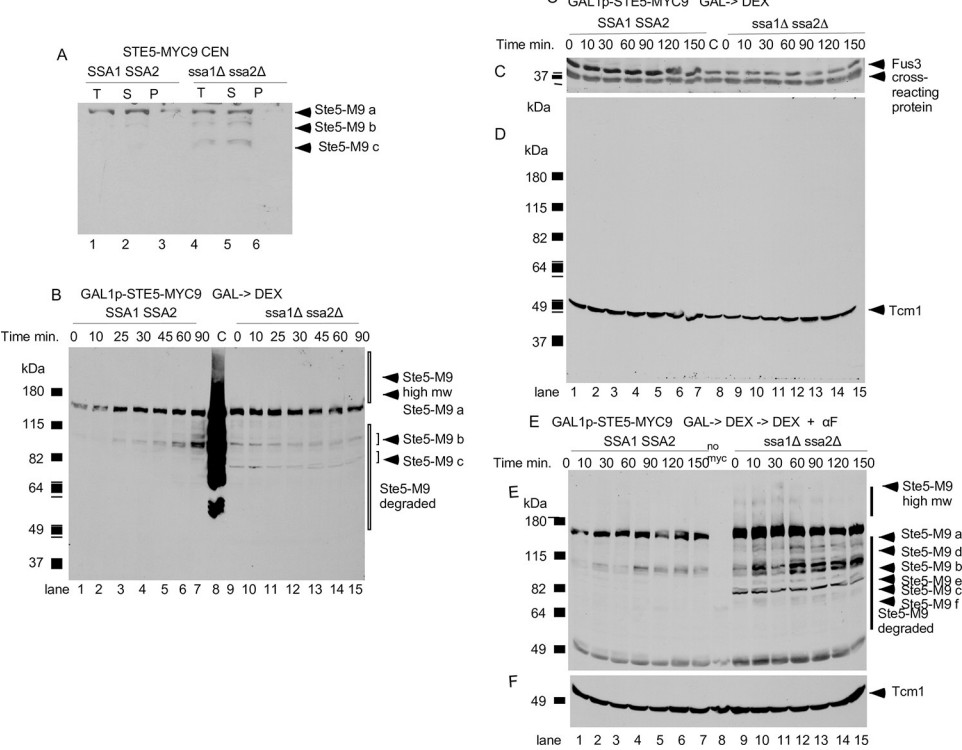

**Fig 5. N-terminal truncations of Ste5-Myc9 accumulate in the *ssa1D ssa2D* double mutant.** A. Comparison of Ste5-Myc9 in pellet and supernatant of wild-type and *ssa1D ssa2D* extracts subjected to 16,000 x g centrifugation. EY3407 *SSA1 SSA2* and EY3409 *ssa1D ssa2D* cells expressing pSKM12 (*STE5-MYC9 CEN URA3*) were grown to A600 of 0.3 and extracts prepared, then an aliquot was centrifuged at 16,000 x g into supernatant (S) and pellet (P) fractions. The immunoblot has 100 μg of total (T) WCE and supernatant and 20 μg of pellet. "a" is full length Ste5-Myc9 and "b" and "c" are truncation products. B-E. Ste5-Myc9 N-terminal truncation products accumulate in a *ssa1D ssa2D* double mutant. B. Ste5-Myc9 levels after *GAL1* promoter shut-off of *GAL1prom-STE5-Myc9* gene during vegetative growth. EY3136 and EY3141 cells harboring pSKM30 (*GAL1p-STE5-MYC9 URA3 CEN*) were pregrown in SC-uracil-2% raffinose medium, then in SC-uracil-2% galactose medium for 4 hours then pelleted, washed and resuspended in 2% dextrose medium. "C" is for control and is the MW standard co-run with overloaded EY1775 + *STE5-MYC9 CEN* (W303a background) grown in 2% dextrose medium. C-D. Abundance of Fus3 and Tcm1 during *GAL1prom-STE5-Myc9* shut-off. Cells were grown as in B. except they were 5 hours in SC-uracil-2%galactose medium. Blot is probed with anti-Fus3 antibodies (C) and anti-Tcm1 antibody (D). E-F. Ste5-Myc9 and Tcm1 during *GAL1-STE5-MYC9* shut off in WT and *ssa1D ssa2D* cells treated with α factor. Cells and extracts prepared as in C except that samples were treated with 5 μM α factor. The immunoblots were probed either with 9E10 antibody (E) or anti-Tcm1 antibody (F). The "a-f" indicate full-length and truncated forms of Ste5-Myc9.

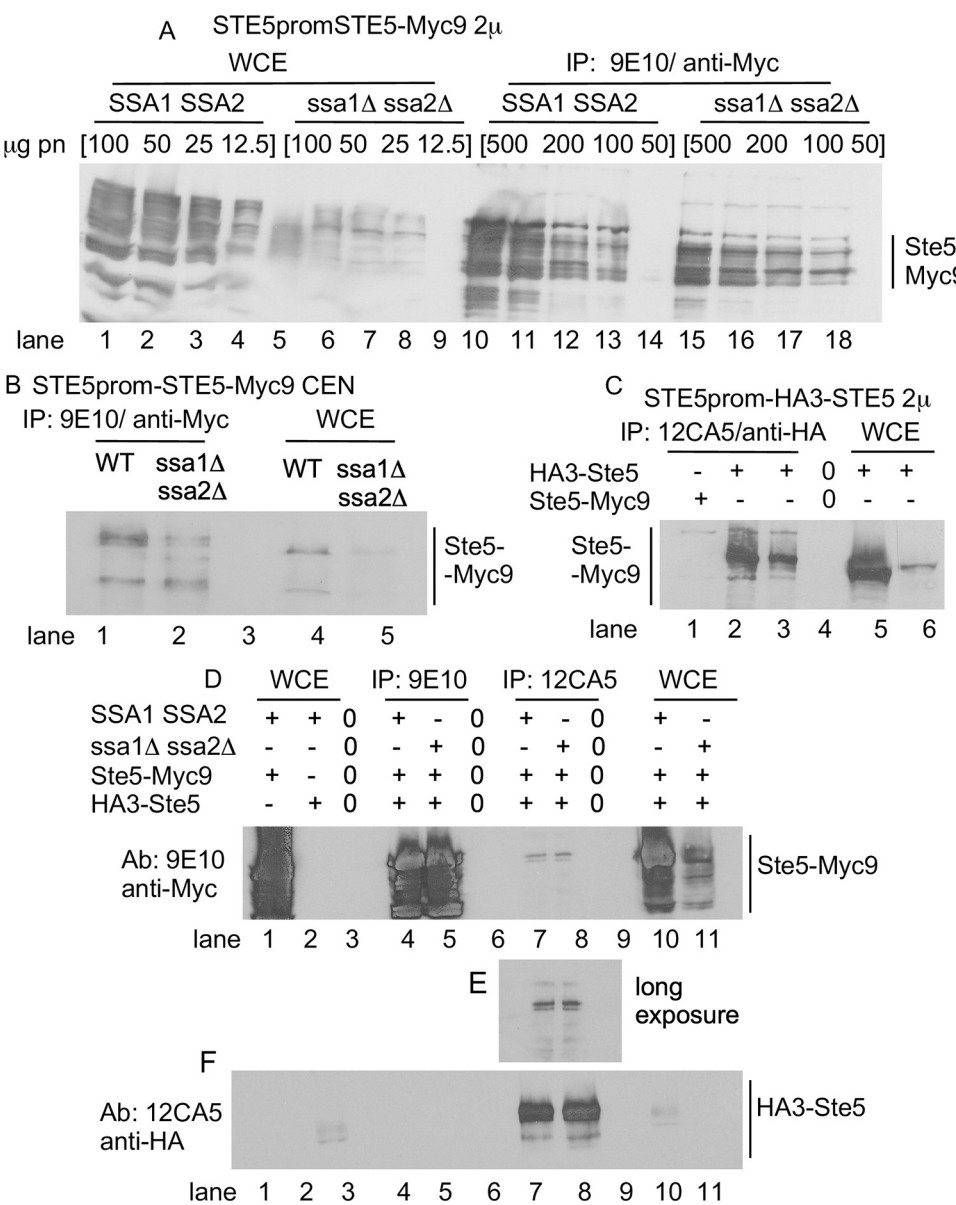

**Fig 6. N- and C-terminal tagged forms of Ste5 from *ssa1D ssa2D* strains have enhanced antibody accessibility.** A. *STE5-Myc9-2*μ WCE ips from wild type (WT) and *ssa1D ssa2D* extracts. Lanes 1–4: WT WCE 100 mg, 50 mg, 25 mg, 12.5 mg; lanes 5–8: *ssa1D ssa2D* WCE 100 mg, 50 mg, 25 mg, 12.5 mg. Lane 9 empty. Lanes 10–14: WT WCE IPs 500 mg, 200 mg and 100 mg, 50 mg. Lane 15 empty. Lanes 16–19: *ssa1D ssa2D* IP WCE 500 mg, 200 mg, 100 mg, 50 mg. Blot probed with 9E10 antibody. B. *STE5-MYC9-CEN* WCE ips from WT and *ssa1D ssa2D* extracts. 0.2 mg WCE was IP'd with 9E10 and compared to 50 μg WCE. Blot was probed with 9E10. Lane 1: WT *STE5-MYC9* WCE. Lane 2: *ssa1D ssa2D STE5-MYC9* WCE. Lane 3: WT Ste5-Myc9 IP. Lane 4: *ssa1D ssa2D* Ste5-Myc9 IP. C. *HA3-STE5-2*μ WCE ips from WT and *ssa1D ssa2D* extracts. 12CA5 IPs were done with 0.5 mg of WCE and run next to 0.1 mg WCE. Blot was probed with 12CA5. Lane 1: WT Ste5-Myc9 WCE IP with 12CA5. Lane 2: WT HA3-Ste5 WCE IP with 12CA5. Lane 3: *ssa1D ssa2D* HA3-Ste5 WCE IP with 12CA5. Lane 4 empty. Lane 5: WT HA3-Ste5 WCE. Lane 6: *ssa1D ssa2D* HA3-Ste5 WCE. For A-C 0.5mg whole cell extract was IP'd with either 12CA5 or 9E10 and then probed with 9E10 in the immunoblot. D. Oligomerization between Ste5-Myc9 (pRM01) and HA3-Ste5-Myc9 (pSKM26) in WT and *ssa1D ssa2D* whole cell extracts. Lane 1: WCE WT Ste5-Myc9. Lane 2: WCE WT HA3-Ste5. Lane 3: Empty. Lane 4: IP 9E10, WT Ste5-Myc9 + HA3-Ste5. Lane 5: IP 9E10, *ssa1D ssa2D* Ste5-Myc9 + HA3-Ste5. Lane 6: Empty. Lane 7: IP 12CA5 WT Ste5-Myc9 + HA3-Ste5. Lane 8: IP 12CA5 *ssa1D ssa2D* Ste5-Myc9 + HA3-Ste5. Lane 9: Empty. Lane 10: WCE WT Ste5-Myc9. Lane 11: WCE *ssa1D ssa2D* Ste5-Myc9. For A-D, extracts were prepared from WT and *ssa1D ssa2D* strains (EY3136, EY3141) expressing *STE5-MYC9 2*μ (pSKM19), *HA3-STE5* (pSKM87) or *STE5-MYC9 CEN* (pSKM12) grown at 30°C in SC-uracil medium containing 2% dextrose.

## Immunofluorescence microscopy

Indirect immunofluorescence and live cell imaging were performed exactly as described [35]. Quantification of nuclear and cortical localization were performed as described [16,35], counting at least 500 cells for each sample. Cells were examined under an Axioscope 2 microscope (Carl Zeiss, Thornwood, N.Y.) linked to a digital camera (C4742-95, Hamamatsu, Bridgewater, N.J.) with filters (set 41018 for GFP and set 51006 Texas Red filter for fluorescein isothiocyanate) from Chroma Technology (Brattleboro, Vermont, USA). Strains harboring pCUP1-GFP-STE5 (pSKM21; 35) were grown overnight at room temperature in selective SC medium to early logarithmic phase ($A_{600}$~0.15), then shifted to selective SC medium containing 0.5 mM $CuSO_4$ for 2 hours before treating cells with $\alpha$ factor as previously described [16,35]. Cells harboring *ADH1prom-SSA1-GFP*, *STE5prom-STE5-MYC9* or *GAL1prom-STE5-MYC9* were grown in SC-selective medium containing either 2% dextrose or 2% raffinose followed by 2% galactose. Cells were examined on a Nikon TE2000E epifluorescence microscope using 360/40, 457/17 (DAPI) filter set (Chroma Corp.). Digital images were collected with a Cooke 12-bit SensiCam CCD driven by IPLab 3.9 software. Whole-image adjustments were performed in Adobe Photoshop. Some cells were visualized with a Hamamatsu ORCA ER digital camera and MetaMorph 7 software at the Department of Cell Biology Microscopy Facility of the Harvard Medical School.

**Mating assays.**   Cells to be tested were first streaked and freshly grown up on YEPD or SC selective plates using autoclaved toothpicks. Then *MATa* or *MATα* cells were arrayed very thinly in square-shaped or pie-slice shaped patches with toothpicks on either SC selective or YEPD plates and grown overnight at either room temperature or 30˚C. In parallel, a fresh culture of *MATα lys9* or *MATa lys9* cells were grown overnight in a few mls of YEPD and lawns were made using 0.2 mls of equal numbers of cells from the fresh cultures. The next day the patches were mated to plates with lawns of either *MATa lys9* or *MATα lys9* cells on YEPD plates for various lengths of time up to 6 hours at room temperature or 30˚C. Then the mixtures of cells on the YNB plates were transferred to selective plates, typically YNB (supplemented as needed if the diploids were to harbor homozygous mutations in metabolic genes) to select for prototrophs that would only arise from formation of diploids. The selective plates were then incubated at room temperature or 30˚C and photographed every day for 4–5 days. It was surprising that the *ssa* mutants were able to mate so well. To test the possibility that we were selecting haploid suppressors rather than diploids, we tested numerous individual prototrophs from our matings for auxotrophies and ability to sporulate and found that they all had appropriate markers for diploids and were able to sporulate as expected. Velvets were used to transfer cells using a replica block. The velvets were prewashed in water and detergent, rinsed carefully and dried on a laundry rack and then packaged into aluminum foil and autoclaved prior to use.

**Data analysis.**   Mean (M), standard deviation (S.D.) and standard error (S.E.) were calculated manually, with excel or website calculators (i.e. socscistatistics (copyright Jeremy Stangroom 2023) and goodcalculators (copyright Good Calculators 2015–2023). The p-value significance tests were calculated with a Student's t-test of two independent means, two-tailed at the socscistatistics website. The Fisher's exact test of significance was calculated for populations of cells after first distinguishing two groups, the group in question versus all other cells not in that particular group using the socscistatistics and statology (copyright 2023, Statology) websites.

## Results

### Ste5 co-purifies with Ssa1

To identify proteins that associate most prominently with Ste5, we purified Ste5-Myc9 from logarithmically dividing cells grown at 30˚C before and after $\alpha$ factor treatment (Materials and

Methods). A number of proteins co-IP'd specifically with Ste5-Myc9, but not with untagged Ste5. Fig 1B labeling indicates six bands that are most prominent: Ste5-Myc9 (~115 kDa), band "a" also labeled as ~70 kDa, bands "b", "c" (which are ~68–58 kDa) and IgG2a (~50 kDa)(Fig 1B). Ste5-Myc9 is largely full-length and the +α factor sample migrates more slowly (Fig 1B, lane 3), presumably from hyper-phosphorylation [38]. Thus, the preparation of Ste5-Myc9 was intact and the α factor treatment induced a robust pheromone response. Band "a" is present in an amount similar to that of Ste5-Myc9, both in absence and presence of α factor treatment (Fig 1B, band "a" lane 2 –αF, lane 3 +αF). The band "a" protein is specifically associated with Ste5-Myc9 and not present in IPs of extracts expressing untagged Ste5 (Fig 1B, lane 1). Similar results were found in repeated immunoprecipitations and a scale-up for mass spectrometry. ImageJ densitometry of the coomassie stained gel in Fig 1B indicates a ratio of 1.45 of Ste5-Myc9:70 kDa Hsp70 –α factor and a ratio of 0.67 Ste5-Myc9: 70 kDa Hsp70 +α factor, suggesting less Hsp70 associated with Ste5-Myc9 in the +α factor extract (S2 Table). The two other bands, "b," "c," are present in sub-stoichiometric amounts compared to Ste5-Myc9. Focus was placed on band "a" due to its relatively high abundance. A large-scale IP of Ste5-Myc9 was performed, the ~70 kDa band "a" was excised from the gel, proteolyzed with trypsin, and microcapillary HPLC-ion trap tandem mass spectrometry was done on two trypsin-generated peptides. Analysis of tandem mass spectra of two major HPLC peaks identified residues 170–185 (IINEPTAAAIAYGLDK) of Hsp70 isoforms Ssa1, Ssa2, Ssa3 and Ssa4 (Fig 1C and 1D). The second peptide identified residues 325–338 of Ssa1 and Ssa2 (SQVDEIVLVGGSTR; Fig 1D). Thus, the majority of Ste5-Myc9 was complexed to Ssa1 and/or Ssa2 both in absence and presence of α factor.

With limits of detection in Coomassie blue being 50 ng and higher and potentially micromolar binding affinities between Ste5 and associated kinases, it is possible that other proteins that associate with Ste5-Myc9 were not visualized. Bands a, b and c are currently unknown in identity. Several signal transduction partners of Ste5 are close in size to bands "a", "b" and "c", including Ste7 (57,723.7 Da) and Ste11 (80,718.8 Da) whereas Fus3 (40,770.5 Da) and Kss1 (42,680.6 Da) might be obscured by the wide IgG2a band. Hsp70s Ssb1 (66,601 Da) and Ssb2 (66,594 Da) are predicted to interact with Ste5 (discussed later in results) and could also be one of the bands.

To confirm the presence of Ssa1, we determined whether Ste5-Myc9 and Ssa1-GFP co-IP using higher salt (200 mM NaCl), nondenaturing immunoprecipitation conditions we have used to reveal Ste5 binding to Ste4, Fus3, Ste7, Ste11, Cdc24, and Msn5/Ste21 [9,10,16,23,86]. We used western immunoblot analysis and antibody-ECL detection methods that are sensitive to the 0.1 ng protein range. Ste5-Myc9 associates with Ssa1-GFP when Ste5-Myc9 is IP'd with 9E10 anti-Myc monoclonal antibody (Fig 2A and 2B, whole cell extracts Fig 2C–2F) and Ste5-Myc9 associates with Ssa1-GFP when Ssa1-GFP is IP'd with anti-GFP polyclonal antiserum (Fig 2F). The interaction between Ste5-Myc9 and Ssa1-GFP is specific based on: 1) the absence of a signal when extracts lack Ste5-Myc9 (Fig 2A, lane 1), 2) more Ssa1-GFP in the co-IP when more Ste5-Myc9 is immunoprecipitated from greater abundance in the whole cell extract (Fig 2A and 2B compare lanes 2,3 in ip Fig 2A and 2B and wce Fig 2C–2E), and 3) the single band of correct migration position for Ssa1-GFP (Fig 2A lanes 2,3) which is distinct from that of GFP-Cdc24, a positive control (Fig 2A, lane 4). HA3-Ste5 also co-immunoprecipitated with Ssa1-GFP (Fig 2G, lanes 2–4, whole cell lysate Fig 2H) with a better signal when more whole cell lysate was used for the IP (Fig 2D Lane 4). These results confirm that Ste5 is complexed to Ssa1 under native conditions.

## Ste5 undergoes extensive posttranslational modification, aggregation and fragmentation that is influenced by Cdc28, α factor and temperature

The mobility of Ste5 in polyacrylamide gels reflects post-translational modifications from phosphorylation and ubiquitinylation [33,34,38] and a propensity to aggregate and degrade

into fragments (this study). This complex banding pattern is affected by expression level, genetic background and posttranslational modification by Cln/Cdk Cdc28, Fus3 and other protein kinases, ubiquitinylation by SCF$^{Cdc4}$ and growth conditions. In a denaturing SDS-PAGE gel, Ste5-Myc9 migrates as a fuzzy broad band near and higher than the expected molecular weight, with multiple phosphorylated sub-bands that shift upward with α factor treatment (Figs 1B and 3A, (3)). In a native gel, the broad diffuse banding pattern is more extensive and more obviously shifts to higher apparent molecular weight with α factor treatment (Fig 3B; note that Fig 3A and 3B buffers contain many phosphatase and protease inhibitors including the PSI proteasome inhibitor). The upward shift of Ste5-Myc9 in the presence of α factor requires signaling through the MAPK cascade and is blocked by a *ste11D* (MAPKKK) mutation (Fig 3B). High molecular weight forms of Ste5-Myc9 accumulate as a high molecular weight smear and as aggregates in the gel well when Ste5-Myc9 is overexpressed and are more pronounced in extracts from wild type (WT) cells grown at 37˚C compared to room temperature (Fig 3C). Aggregates in a gel well can occur when a protein is polyubiquitinylated or oligomerizes which occurs for Ste5 [34,37,38]. The *cdc28-4* mutation increases the relative amount of full-length Ste5-Myc9 at 37˚C and room temperature (Fig 3C, lanes 5,6,] a) and reduces the spread of the full-length band, presumably from loss of phosphates from Cln/Cdc28 (Fig 3C,] a, lanes 2, 7, 9). The *cdc28-4* mutation also abolishes aggregated forms of Ste5-Myc9 in the gel well and high molecular weight forms in the smear, suggesting these species include phosphorylated and/or ubiquitinylated versions of Ste5-Myc9 (Fig 3C, compare lanes 1,2 with 3,4) [37]. The *cdc28-4* mutation also reduces the relative abundance of proteolyzed fragments of Ste5-Myc9 (Fig 3C, compare a] to] b-]g), presumably from loss of ubiquitin-mediated degradation (Fig 3C, lanes 6–9). The prominent high molecular weight smear and aggregates in the gel well are specific to Ste5-Myc9 and are not apparent for Tcm1, Fus3, Kss1 and Ste7 (see later figures). They were not apparent for Ste5-Myc9 in the Coomassie-stained gel (Fig 1B), perhaps because Coomassie Blue is a less sensitive means of detection compared to the 9E10 monoclonal antibody directed against the Myc epitope which can detect ~0.1 ng protein. Strikingly, long exposures of films of whole cell extracts from the Ssa1-GFP and Ste5-Myc9 co-immunoprecipitations in Fig 2 also reveal phosphorylated Ste5-Myc9, high molecular weight forms and aggregation that are all more prominent with excess Ssa1-GFP than with GFP-Cdc24 (Fig 3D–3E).

## Ssa1 and Ssa2 positively regulate Ste5 abundance especially at higher temperatures

We determined the effect of *ssa10Δ ssa2Δ* double mutations on the abundance and gel migration pattern of Ste5-Myc9 expressed from *CEN* (~1 copy/cell) and *2μ* (multiple copies/cell) plasmids using haploid strains generously provided by the Craig lab. The *ssa1Δ ssa2Δ* double mutant strain MW142 (EYL339) grew poorly compared to the isogenic *SSA1 SSA2* strain (WT; T211 (EYL341)) at all temperatures examined (Figs 4A and S2A, note that strain MW142 was originally designated as *MATa* but was found to be *MATa* as indicated in S1 Table). The *ssa1Δ ssa2Δ* double mutant strain MW142 (EYL339) generated colonies that were only 19.4% the diameter of wild type (Fig 4A; YPD plate at 30˚C, day 3, the mean colony size in arbitrary units for *ssa1Δ ssa2Δ* is 0.788 SE +/- 0.0256 and WT is 4.06, SE +/- 0.148, p-value < 0.00001).

Ste5-Myc9 protein levels were reduced in the *ssa1D ssa2D* double mutant MW142/EYL339 at room temperature during vegetative growth when it was expressed from its own promoter on a *CEN* plasmid (Fig 4B, lanes 1–2). Densitometry showed that the level of Ste5-Myc9 was ~20% that of wild type T211/EYL341 after normalization with ribosomal protein Tcm1 (i.e.

0.20 +/- 0.02 S.E. (N = 2), S2 Table). This observation was striking given that Ste5 has a long half-life of ~90 minutes by immunoblot analysis [37,38] and 3 hours by heavy lysine isotope labeling [90] and the abundance of Ste5 protein, but not its mRNA, increases somewhat during α factor stimulation, peaking after 1 hour of stimulation [38]. When Ste5-Myc9 was overexpressed with a 2-micron multicopy plasmid, the abundance of Ste5-Myc9 was ~57% that of the wild type control (i.e. 0.57 +/- 0.034 (N = 2) Fig 4B, lanes 4,5; S2 Table). Thus, overexpression bypasses some of the reduction caused by loss of *ssa1D ssa2D*. Longer exposure of the immunoblot revealed that Ste5-Myc expressed in the wild type Craig strain formed aggregated species in the well of the gel and a somewhat higher molecular weight smear (Fig 4C, lanes 7,8). In addition, degradation fragments of Ste5-Myc9 were present with both 2 micron and *CEN* expression levels (Fig 4B and 4C, lanes 1,2,7,8). By contrast, Ste5-Myc9 from the *ssa1D ssa2D* strain had less obvious aggregated forms, less obvious higher molecular weight smear, a narrower full-length Ste5-Myc9 band (Ste5-Myc9FL) and an altered pattern of Ste5-Myc9 fragments (Fig 4C, lanes 2,3,9,10). These effects were specific to Ste5-Myc9, because there were no alterations in SDS-PAGE banding pattern detected for Tcm1 (Fig 4D, i.e. ribosomal protein L3) or other proteins examined in this study.

To test the generality of the requirement of Ssa1 and Ssa2 and the effect of temperature, we created isogenic *MATa ssa1D ssa2D* and *MATa* wild type (WT, i.e. *SSA1 SSA2)* strains by sporulating and dissecting ascospores from MW63, a *MATa/MATα ssa1::HIS3/+ ssa2::LEU2/ + ssa3::TRP1/+ ssa4::URA3/+* diploid from the Craig lab (S1 Table). The *ssa1D ssa2D* haploid ascospores grew poorly at room temperature and 30˚C and were inviable at 37˚C in streakouts on YPD plates (S2A–S2E Fig). During vegetative growth, the *ssa1::HIS3*, *ssa2::LEU2* single mutants and *ssa1::HIS3 ssa3::TRP1 ssa4::URA3* and *ssa2::LEU2 ssa3::TRP1 ssa4::URA3* triple mutants had only slightly slower growth than wild-type at room temperature (~25˚C) and 30˚C (S2A–S2D Fig, *ssa1*: 53% wt colony size N = 22, *ssa2*: 77% N = 20). By contrast, the *ssa1D ssa2D* double mutant grew poorly at room temperature and 30˚C and barely grew at all at 37˚C (S2E Fig, Table 7; *ssa1D ssa2D* was 19% wt colony size at room temperature, arbitrary mean colony diameter 0.79 +/- 0.026 S.E. (N = 50) for *ssa1D ssa2D* and 4.06 +/- 0.15 S.E. (N = 50) for *SSA1 SSA2*, p-value 0.00001). The small and big colonies grew at similar rates in liquid culture, suggesting varying delays in resuming growth on solid support. Spontaneous suppressors that permitted faster growth arose after 5 days incubation at 37˚C at a rate of $10^{-7}$ similar to what has been found [84,85,91,92]. The *ssa1D ssa2D ssa4D* triple and *ssa1D ssa2D ssa3D ssa4D* quadruple mutants were inviable and died as unbudded haplospores as previously found.

Importantly, transformation of ascospores EY3136 *MATa SSA1 SSA2* and EY3141 *MATa ssa1D ssa2D* with the same *STE5-Myc9-2m* and *-CEN* plasmids recapitulated lower abundance of Ste5-Myc9 in the *ssa1D ssa2D* strain (Fig 4E, Mean +/- S.E. from separate sets of 2m experiments: 0.37 +/- 0.035 (N = 4) and 0.29 +/- 0.022 SE (N = 4) and even lower abundance with multiple *CEN* plasmid experiments, S2 Table).

We examined Ste5-Myc9 at different temperatures when expressed from the *CEN* plasmid. In the wild type control, more Ste5-Myc9 was in the gel well and fragmented at 37˚C compared to at RT (Fig 4F–4I, lanes 1–3,8–10) suggesting elevated temperature leads to more Ste5 aggregation and degradation. Remarkably, Ste5-Myc9 was nearly undetectable at 30˚C, 37˚C, and 42˚C compared to wild type, although Tcm1 remained unchanged (Fig 4F–4I). Densitometry and averaging of data from three independent transformants for WT and *ssa1D ssa2D* strains revealed that Ste5-Myc9 abundance was 21% that of wild type at room temperature, further reduced at higher temperatures to 14% at 30˚C, 7% at 37˚C and undetectable at 42˚C (i.e. for RT: 0.19 +/- 0.08 S.E. (N = 3), p-value 0.003154; for 30˚C: 0.13 +/- 0.13 S.E. (N = 3), p-value 0.005405; and for 37˚C: 0.12 +/- 0.12 (N = 3), p-value 0.00032; Fig 4F and 4G, S2 Table). Thus,

temperature increase appears to increase degradation of Ste5-Myc9 in wild type and *ssa1D ssa2D* strains (Fig 4F–4I, S2 Table). The effects are most likely posttranscriptional since the Ste5 promoter is not regulated by Ssa1, Ssa2, Hsf1 or Msn2/Msn4 (SGD *STE5* expression annotation data). The truncated species are unlikely to be premature translational stop peptides since the Myc tags are at the carboxyl-terminus.

The level of Ste5-Myc9 was also reduced when *ssa1D ssa2D* cells were grown in 2% glucose and 2% galactose (Fig 4J and 4K; note that Ste5-Myc9 abundance is slightly lower in 2% galactose compared to 2% glucose) further substantiating a general requirement for Ssa1 and Ssa2 for Ste5 abundance under a different carbon source. Thus, Ssa1 and Ssa2 are important for Ste5 protein abundance and integrity at room temperature and crucial at elevated temperatures.

## Loss of Ssa1 and Ssa2 alters the integrity of Ste5

We examined the effect of *ssa1D ssa2D* mutations on the integrity of Ste5, by examining a number of its characteristics that we have previously defined. Ste5 exists in a high molecular weight complex that sediments in a particulate fraction of >500 kDa with Ste11, Ste7 and Fus3 in glycerol gradients and is also associated with cytoskeletal proteins including Cdc24, Bem1 and Bni1 [1,4,10,22]. We examined the distribution of Ste5 in supernatant and pellet fractions of our whole cell lysate preparations from WT *SSA1 SSA2* (EY3407 from EY3141) and *ssa1D ssa2D* (EY3409 from EY3136) strains, using an initial brief 3,000 x g centrifugation followed by a 16,000 x g centrifugation that is known to pellet nuclei, cell debris, contractile/cytoskeletal apparatus, cell wall and some aggregated proteins. The lanes in the immunoblot in Fig 5A have 100 μg protein for total and supernatant samples and 20 μg protein for pellet samples. Strikingly, densitometry revealed ~4-fold more full length (FL) Ste5-Myc9 in *WT* compared to in *ssa1D ssa2D* and different pellet to supernatant distributions: ~10% of Ste5-Myc9 FL was in the *ssa1D ssa2D* pellet compared to ~76% in pellet of WT extracts. Furthermore, more Ste5-Myc9 was fragmented in the supernatant of the *ssa1D ssa2D* lysate compared to the WT lysate (Fig 5A, note black marks at top are indelible marker ink on the film). These results further substantiate that the *ssa1D ssa2D* mutations alter abundance and integrity of Ste5 and suggest the mutations may change subcellular distribution.

To examine a previously synthesized, post-transcriptional pool of Ste5-Myc9, we induced its overexpression with the *GAL1* promoter for 5 hours in galactose medium, then stopped expression with dextrose medium and followed abundance over time in the absence and presence of α factor as done previously [38]. The addition of a factor is expected to inhibit, over time, the pool of Cln/Cdc28 kinases that phosphorylate Ste5 and target it for ubiquitylation and degradation by the proteasome. The PSI proteasome inhibitor was included in the extraction buffer to inhibit ubiquitin-mediated degradation in extracts. In a W303a *ste5D* null strain, the abundance of Ste5-Myc9 declined to 24% its initial level by 150 minutes [38] and accumulated as high mw species when overexpressed sufficiently in W303a *ste5D* (Fig 5B, lane 8 labeled C, 2-micron plasmid, selective medium with 2% dextrose). The abundance of *GAL1* promoter-induced full-length Ste5-Myc9 was significantly lower in the wild type *SSA1 SSA2* Craig lab background compared to W303a and did not decline noticeably over 90 minutes, but more fragments of Ste5-Myc9 accumulated by 90 minutes ("Ste5-Myc9 a", Fig 5B, lanes 1–7 compared to lane 8, note Ste5-Myc9 fragments labeled "b"). Some reduction in abundance of full-length Ste5-Myc9 (labeled as "Ste5-Myc9 a") could be discerned in the *ssa1D ssa2D* strain (Fig 5B, lanes 9–10). We did not detect high molecular weight species of Ste5-Myc9. A proteolytic fragment indicated as "Ste5-Myc9 b" was present in both the Craig background wild-type (EY3136) and *ssa1D ssa2D* (EY3141) whole cell extracts, but the pattern of

accumulation was distinct. The wild type "Ste5-Myc9 b" fragments are visible at 25 min to 90 min and increase in abundance. By contrast, in the *ssa1D ssa2D* extracts, the "Ste5-Myc9 b" fragments are present at all time points with no increase in abundance. Notably, a second set of proteolytic fragments, "Ste5-Myc9 c" were readily visible in the *ssa1D ssa2D* extract but not the wild type extract. By contrast, Tcm1 remained constant in these experiments and was intact and not aggregated (Fig 5D). Thus, the integrity of Ste5-Myc9 is compromised in the *ssa1D ssa2D* double mutant compared to wild type with accumulation of additional Ste-Myc9 fragments with distinct proteolytic sites.

Further changes in the Ste5-Myc9 fragment profile occurred with α-factor (Fig 5E and 5F). The "Ste5-Myc9 a" full-length band became 1.7+/-0.22-fold broader in the *ssa1D ssa2D* strain compared to in wild type together with a low amount of high molecular weight Ste5-Myc9 species at the 0 to 90 minute time points, particularly from time 0 to 60 minutes and barely detectable at 90, 120 and 150 minute time points when the pool of Cln/Cdc28 should be most inactive. In addition, the *ssa1D ssa2D* extracts had a broader "Ste5-Myc9 b" set of fragments, the "Ste5-Myc9-b" and "Ste5-Myc9 c" fragments were more abundant, and there were additional sets of fragments (labeled "d,e,f,g" in Fig 5E). By comparison, in the wild type extracts with α factor, the Ste5-Myc9 fragments at position "b" were only weakly detected at the 10 minute time point and there were no obvious "Ste5-Myc9 c, d,e,f,g" fragments (Fig 5B and 5E). Thus, Ste5-Myc9 is more vulnerable to proteolysis without Ssa1 and Ssa2, especially during α-factor signaling. The strong protective function of Ssa1 and Ssa2 for Ste5-Myc9 was quite specific, because the *ssa1D ssa2D* mutations did not obviously alter the integrity of the protein band profiles of Fus3, Tcm1 (Fig 5C, 5D and 5F) or Fus3-HA, Ste7-Myc or Kss1 (Figs 8A, 9B and 9D). Thus, Ssa1 and Ssa2 are crucial for maintaining steady state levels of Ste5-Myc9 and protecting it from degradation during vegetative growth, with greater impact during a factor signaling and temperature increases.

We examined whether Ssa1 and Ssa2 influence the accessibility of antibodies to bind to C-terminal Myc9 and N-terminal HA3 tags on Ste5. Our rationale was based on a working hypothesis that the ends of Ste5 may be more sensitive to conformational changes, because we previously found mutations that change the availability of the N- and C- termini of Ste5 to antibody [16] and the N-terminal ~1–161 and C-terminal ~760–877 regions of Ste5 are predicted to have high intrinsic disorder (S1B–S1E Fig). Antibody accessibility is a well-established method to monitor changes in peptide flexibility, conformation, and binding to partners [e.g. 93,94]. Strikingly, although wild type whole cell extracts had >3-fold more Ste5-Myc9 than *ssa1D ssa2D* whole cell extracts, 3-fold more Ste5-Myc9 and 7-fold more HA3-Ste5 were IP'd from the *ssa1D ssa2D* whole cell extracts than from the wild type whole cell extracts (Fig 6A–6C). ImageJ densitometry (S2 Table) revealed 1.9 +/- 0.13 S.E.-fold more Ste5-Myc9 was IP'd from *ssa1D ssa2D* extracts expressing the *STE5-MYC9 2*μ plasmid, although the abundance of Ste5-Myc9 was 3.4 +/- 0.076 S.E.-fold. higher in the isogenic wild type strain (Fig 6A) and 1.7-fold more Ste5-Myc9 was IP'd from the *ssa1D ssa2D* extracts expressing the *STE5-MYC9 CEN* plasmid compared to wild type extracts, although the abundance of Ste5-Myc9 was 2.6-fold higher in the wild type extracts (Fig 6B). Moreover, 3.0-fold more HA3-Ste5 was IP'd from the *ssa1D ssa2D* strain expressing *HA3-STE5 CEN* plasmid than from the isogenic wild type strain, although the wild type strain had ~7.3-fold more HA3-Ste5 than the *ssa1D ssa2D* strain (Fig 6C, S2 Table). A second experiment revealed a 5.6-fold more HA3-Ste5 IP'd from the *ssa1D ssa2D* strain compared to WT (Fig 6D, S2 Table; Mean 4.3 +/- 1.3 S.E.). Therefore, both N- and C-terminal tags on Ste5 are more available to antibodies in the *ssa1D ssa2D* extracts.

The active form of Ste5 may be a dimer that forms through internal RING-H2::RING-H2 and other intermolecular interactions (S1A, S3D and S3E Figs; [38]). Less than 1% of the total

pool of Ste5 is detected as oligomers during vegetative growth using Ste5-Myc9 [38], presumably because of auto-inhibition within the monomer [38,95]. We examined oligomerization between HA3-Ste5 and Ste5-Myc9 overexpressed with 2 micron plasmids in wild type and *ssa1D ssa2D* whole cell lysates (Fig 6D). The abundance of Ste5-Myc9 was several-fold less in the *ssa1D ssa2D* extracts compared to wild type (Fig 6D lanes 9,10) and the weakly detected HA3-Ste5 was also reduced in the *ssa1D ssa2D* extract (Fig 6D, lanes 10,11, top panel). Nevertheless, equivalent amounts of Ste5-Myc9 (Fig 6D, lanes 4,5 top panel) and HA3-Ste5 (Fig 6D, lanes 7,8, bottom panel) immunoprecipitated and equivalent or slightly more Ste5-Myc9:: HA3-Ste5 oligomers co-immunoprecipitated with HA3-Ste5 from wild type and *ssa1D ssa2D* extracts (Fig 6D lanes 7,8). Therefore, Ste5-Myc9 dimerizes equivalently or slightly more in the *ssa1D ssa2D* extracts than the wild type extracts.

## Loss of Ssa1 and Ssa2 interferes with cortical localization of Ste5

We examined the effect of *ssa1D ssa2D* mutations on the localization of Ste5 at the cell cortex. Localization of Ste5 to the cell periphery is associated with activation of signaling and shmoo formation and is thought to involve an induced conformational change co-incident with binding to Gβγ that correlates with RING-H2 domain oligomerization at the plasma membrane. GFP-Ste5 localized throughout the cytoplasm in the *ssa1D ssa2D* double mutant before and after α factor treatment, but did not efficiently accumulate at the growth site of the emerging projection tip. The % rim staining for GFP-Ste5 in *ssa1D ssa2D* (EY3141) was 12.4 +/- 4.1 S.E (N = 3) versus 75.2 +/-7.3 S.E. (N = 6) for wild type (EY3136) after 90 minutes in 50 nM α factor (Table 1; Fig 7A and 7B shows a representative experiment). The *ssa1D ssa2D* cells were more enlarged and misshapen compared to wild type which is a sign of reduced α factor signaling. The fluorescence signal of GFP-Ste5 was lower in the *ssa1D ssa2D* strain than the wild-type strain leading to fewer and less intensely fluorescing GFP positive cells, consistent with lower abundance. The cells that exhibited enrichment at the growth site were the most shmoo-like in morphology. These findings were corroborated by indirect immunofluorescence with two additional Ste5 constructs, Ste5-Myc9 and TAgNLS$^{K128T}$-Ste5-Myc9, the latter of which has enhanced ability to be localized to the nucleus as well as be stably recruited to the cell cortex (Table 1) [35]. Thus, Ssa1 and Ssa2 are required for efficient recruitment of Ste5 to the cell periphery even when the level of Ste5 is increased or when the Ste5-Myc9 derivative has enhanced ability to localize to the nucleus and be recruited. Collectively, these findings suggest the integrity of Ste5 is not optimal for being recruited to the plasma membrane.

## Ssa1 and Ssa2 enhance nuclear accumulation of Ste5

Cytoplasmic Hsp70/Hsc70 proteins in yeast and mammals including Ssa1 and Ssa4 shuttle through the nucleus and are implicated in facilitating nuclear import of associated proteins [70,96], although excess Ssa1 and Ssa2 inhibits nuclear accumulation of its substrate Gts1 [97]. We examined the effect of *ssa1D ssa2D* mutations on nuclear accumulation of Ste5-Myc9. We compared the localization of Ste5-Myc9 expressed at native levels in wild-type and *ssa1D ssa2D* strains and found that Ste5-Myc9 accumulated in fewer *ssa1D ssa2D* nuclei compared to wild-type before and during α factor treatment at room temperature (Table 1; % Nuclear accumulation of Ste5-Myc9 (2m) at room temperature is 25 +/- 0.8 S.E. for WT versus 1.6 +/- 0.092 for *ssa1D ssa2D*, p-value 0.00260132 and after 2 hrs a factor treatment WT is 4.0 +/- 0.33 S.E. versus *ssa1D ssa2D* 0.7 +/- 0.38 S.E., p-value 0.01101528). Ste5(1–242)-GFP2 which has the NLS and RING-H2 domain also accumulated in fewer nuclei in *ssa1D ssa2D* cells (Table 1; % Nuclear accumulation for WT is 69.3 +/- 23.9 S.E. (N = 3) versus *ssa1D ssa2D* is 0.2 +/- 0.2 S. E. (N = 3), p-value 0.04476). By contrast, TAgNLS-GFP2, which has an NLS from SV40 T-

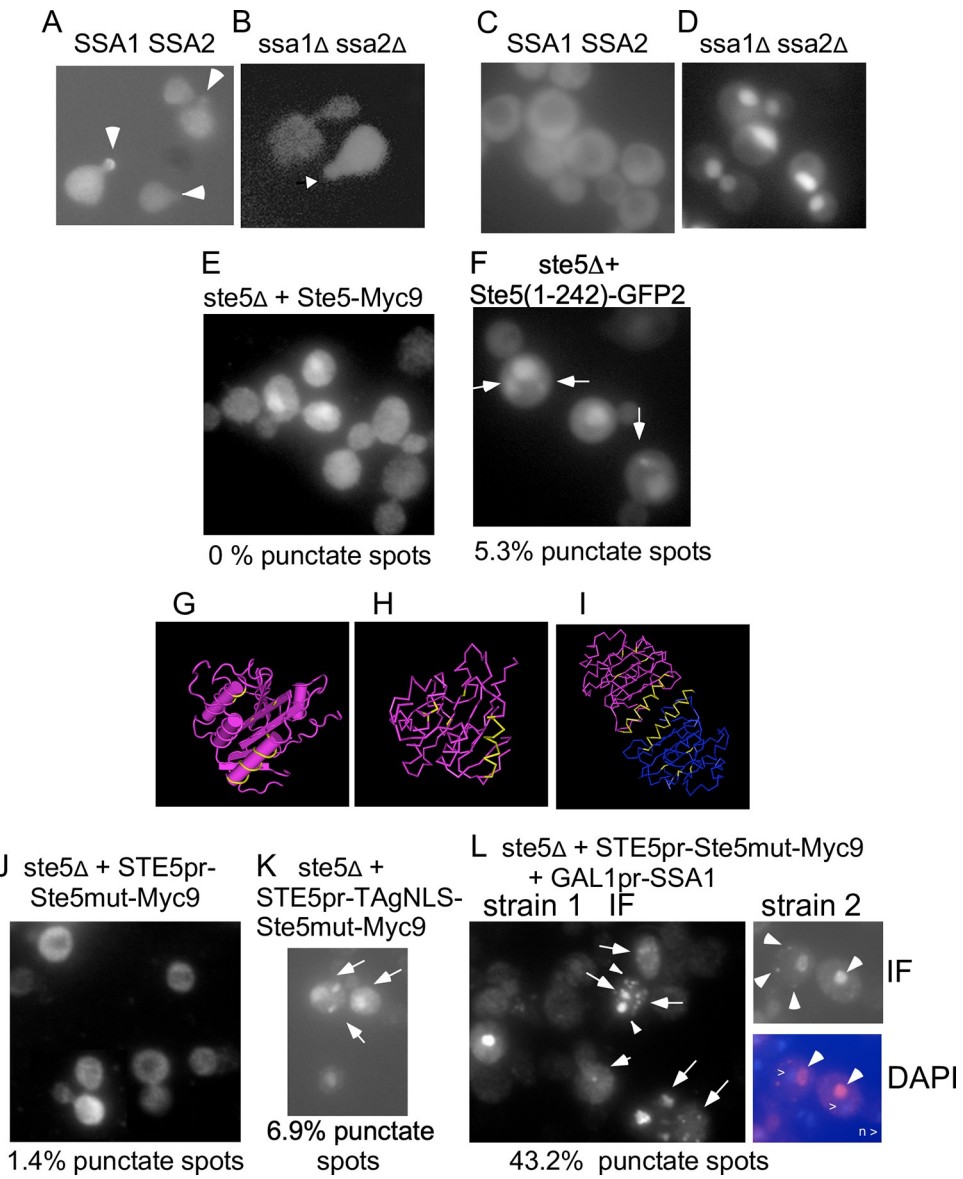

**Fig 7. Ssa1 and Ssa2 alter the localization of wild-type and mutant Ste5.** A-B. Live cell image of GFP-Ste5 expressed from a *CEN* plasmid in α factor-induced WT and *ssa1D ssa2D* cells. The GFP-Ste5 (pSKM21) signal is weaker and found in fewer *ssa1D ssa2D* nuclei and cells in the population compared to in WT cells (See Table 1). Arrows indicate GFP-Ste5 accumulation at shmoo tips. C-D. Live cell images of Ssa1-GFP in WT and *ssa1D ssa2D* strains. E. Ste5(1–242)-GFP2. *ste5D (EY1775)* + Ste5(1–242)-GFP2 grown at room temperature. Arrows indicate punctate spots (foci). F. Ste5-Myc9. *ste5Δ* (EY1775) + Ste5-Myc9. G-H Ste5-Ms 3FZE monomer crystal structure. I. Ste5 4F2H dimer crystal structure [17,27]. The positions of the Ste5L610A/L614A/L634A/L637A mutations within the VWA-like domain and the helix that dimerizes are colored yellow. See S1A Fig for larger images. J. ste5L610/614/634/637A-Myc9 in *ste5D* (indicated as Ste5mut-Myc9), K. GAL1pr-TAgNLS-ste5L610/614/634/637A-Myc9 in *ste5D*. L-O. Overexpression of Ssa1 increases ste5L610/614/634/637A-Myc9 localization in punctate foci. M. *GAL1pr-SSA1* (EBL531) + *ste5mut-Myc9* (pSKM19) clone 1 (YMY550) stained with 9E10. N. *GAL1pr-SSA1* (EBL531) + *ste5mut-Myc9* clone 2 (YMY549) stained with 9E10. O. DAPI stain of the cell in N. Arrows indicate punctate foci. The *ste5D* strains co-expressing *ste5L610/614/634/637A-Myc9* and *SSA1* were grown at 30˚C in SC-2% dextrose selective medium then shifted to SC-2% galactose selective medium to induce Ssa1.

**Table 1. Localization of Ste5 to cell periphery and nucleus in wild type and *ssa1D ssa2D* cells.**

| Plasmid/Strain | %N>C (SE)[1] | %Rim (SE)[2] | %GFP+ [3] (SE)[4] |
|---|---|---|---|
| 1A. *CUP1prom-GFP-STE5, CEN, RT, 50 nM aF 90 min* | | | |
| WT | | 75.2 (7.28) N = 6 | 87.7 (6.4) N = 3 |
| *ssa1D ssa2D* | | 12.4 (4.1) N = 3 | 35.2 (7.6) N = 3 |
| p-value[5] | | [.000727] | [.006161] |
| 1B. *STE5prom-STE5-MYC9, 2m*, RT, 50 nM aF 45 min | | | |
| WT | | 29.3 (~150 cells) | |
| *ssa1D ssa2D* | | 16.8 (~150 cells) | |
| 1C. *STE5prom-TAgNLSK128T-STE5-MYC9, 2m*, RT, 50 nM aF 20 min | | | |
| WT | | 40.7 (~250 cells) N = 1 | |
| *ssa1D ssa2D* | | 4.2 (~250 cells) N = 1 | |
| 2. *STE5prom-STE5-MYC9, 2m*, RT, 50 nM aF 45 min | | | |
| WT | 25 (0.83) N = 8 | | |
| *ssa1D ssa2D* | 1.6 (0.092) N = 8 | | |
| p-value | [.00260132] | | |
| WT aF 1hr | 2.7 (0.84) N = 3 | | |
| *ssa1D ssa2D* aF 1hr | 1.4 (0.92) N = 3 | | |
| p-value | [.3156809] | | |
| WT aF 2hrs | 4.0 (0.33) N = 3 | | |
| *ssa1D ssa2D* aF 2hrs | 0.7 (0.38) N = 3 | | |
| p-value | [.01101528] | | |
| 3. *STE5prom-STE5-MYC9, 2m* | | | |
| *ste5D::TRP1* (EY1775), RT | | | |
| 2% dextrose | 4.6 (1.9) N = 7 | | |
| 2% raffinose | 73.7 (14.9) N = 8 | | |
| p-value | [< .00001] | | |
| 2% galactose | 75.4 (6.9) N = 8 | | |
| p-value | [< .00001] | | |
| 4. *STE5prom-STE5-MYC9, 2m* | | | |
| *ste5D::ADE2* (EY3427) | | | |
| 2% dextrose | 9.3 (3.7) N = 4 | | |
| 2% raffinose | 69.5 (9.0) N = 4 | | |
| p-value | [.000816] | | |
| 2% galactose | 74.0 (11.7) N = 4 | | |
| p-value | [.001908] | | |
| 5. *CUP1prom-STE5-MYC9, CEN* | | | |
| WT (EY957), RT | | | |
| 2% dextrose | 13.5 (5.5) N = 2 | | |
| 2% raffinose | 66.0 (19.0) N = 2 | | |
| p-value | [.11746] | | |
| 2% galactose | 61.0 (21.0) N = 2 | | |
| p-value | [.160147] | | |
| 6. *STE5prom-STE5(1–242)-GFP2*, RT | | | |
| WT | 69.3 (23.9) N = 3 | | |
| *ssa1D ssa2D* | 0.2 (0.2) N = 3 | | |
| p-value | [.04476] | | |
| 7. *ADH2prom-TAgNLS-GFP2*, RT | | | |
| WT | 98.33 (1.67) N = 3 | | |

*(Continued)*

**Table 1.** (Continued)

| Plasmid/Strain | %N>C (SE)[1] | %Rim (SE)[2] | %GFP+ [3] (SE)[4] |
|---|---|---|---|
| *ssa1D ssa2D* | 93 (3) N = 2 | | |
| p-value | [.184167] | | |
| 8. *ADH2prom-TAgNLS-NES-GFP2*, RT | | | |
| WT | 8.13 (6.6) N = 3 | | |
| *ssa1D ssa2D* | 0 (0) N = 3 | | |
| p-value | [.292909] | | |

[1]Nuclear accumulation: The %N>C is the percentage of total 9E10 positive cells with more intense staining in the nucleus compared to the cytoplasm. EY3136 (wild type *SSA1 SSA2*) and EY3141 (*ssa1D ssa2D*) cells harboring pSKM19 (*STE5prom-STE5-MYC9 2m*), pSKM17 (*CUP1prom-STE5-MYC9 CEN*) and pSKM98 (*TAgNLSK128T-STE5-MYC9 2m)* in SC-uracil 2% dextrose (with copper sulfate for *CUP1* promoter). Cells were grown overnight at room temperature to A600 0.3–0.6 then adjusted to equal density, centrifuged and resuspended in fresh medium without or with 5 mM a factor with shaking for the indicated time. EY1775 (*ste5D::TRP1*) and EY3427 (*ste5D::ADE2*) harboring pSKM19 and EY957 (wild type for *STE5*) harboring pSKM17 were grown overnight to logarithmic phase of A600 0.5 in SC-uracil medium containing either 2% dextrose, 2% galactose or 2% raffinose. Cells harboring Ste5-Myc9 were then fixed and processed for indirect immunofluorescence with DAPI and 9E10 monoclonal antibody for the Myc epitope.

Live cell imaging was done for GFP-Ste5, pSKM113 (Ste5(1–242)-GFP-GFP), pYMY27.1 (TAgNLS-GFP-GFP) and pYMW25 (TAgNLS-PKI(NES)-GFP-GFP). Approximately 150–379 cells were tallied for each sample.

[2]Rim localization: Enriched 9E10 or GFP signal at cell cortex and projection tip.

[3]%GFP+ is the percentage of cells that are GFP positive. The GFP-Ste5 signal was weaker and less prevalent in the *ssa1D ssa2D* cells compared to wild type control.

[4]S.E. is standard error which is standard deviation divided by the square root of the number of samples.

[5]p-value significance tests were calculated with a Student's t-test of two independent means, two-tailed using the social science statistics calculator at the "socscistatistics" website (copyright Jeremy Stangroom 2023).

antigen and two copies of GFP to prevent diffusion into the nucleus, efficiently accumulated in nuclei of *ssa1D ssa2D* cells (Table 7;% Nuclear accumulation in WT is 98.33 +/- 1.67 S.E. (N = 3) versus *ssa1D ssa1D* is 93 +/- 3 S.E. (N = 2)). The TAgNLS-NES-GFP2 fusion protein (which has a nuclear export sequence from PKI and is able to undergo nucleocytoplasmic shuttling) localized in fewer nuclei in the *ssa1D ssa2D* strain compared to wild type, but it was not within a statistically significant p-value of .01 to .1 (% nuclear accumulation was 8.13 +/- 6.6 S.E. (N = 3) for WT versus 0 +/- 0 S.E. (N = 3) for *ssa1D ssa2D*, p-value .292909). These results imply that Ssa1 and Ssa2 may stimulate either nuclear accumulation of Ste5 via import, retention Ste5 in the nucleus or reduce export of Ste5 and may do so in part through the first 242 residues of Ste5.

Yeast cells grow most optimally at temperatures of 25˚C (room temperature) to 30˚C. The *S. cerevisiae* heat shock response is activated at temperatures of 37˚C and higher with cell viability declining after long exposure to temperatures higher than 42˚C [55]. A heat shock at 55˚C induced cell death with no nuclear accumulation of Ste5-Myc9 in either the *SSA1 SSA2* or *ssa1D ssa2D* strains (Table 2), suggesting that the high temperature heat shock effect does not require Ssa1 and Ssa2. The reduction in the nuclear pool of Ste5-Myc9 could arise through reduced import, greater degradation in the nucleus, reduced anchoring, increased export or cell death.

We serendipitously found that elevated temperatures induce more nuclear accumulation of Ste5 in two strains. A W303a *MATa cdc24-1* (*CDC24-CEN-URA3)* strain from Rob Arkowitz accumulated more Ste5-Myc9 in nuclei at 37˚C and 55˚C compared to at room temperature (Table 2, p-value .000147 comparing RT with 55˚C). More nuclear accumulation of Ste5(1–242)-GFP2 occurred in FY23 S288c cells grown at 37˚C compared to at room temperature, although it was not statistically significant for p <0.01–0.1 (% nuclear accumulation at RT was 53.8 +/- 15.9 S.E. versus 85.8 +/- 5.4 S.E. at 37˚C, p-value .197142). Thus, Ssa1 and Ssa2

**Table 2. Effect of temperature shift on accumulation of Ste5-Myc9 in nuclei.**

| Strain/Plasmid[1] | Temp | % N>C tallies[2] | %N>C M +/- S.E[4] | p-value[5] |
|---|---|---|---|---|
| *SSA1 SSA2 + STE5-MYC9 2m* | | | | |
| | RT 1–2 hr | 14, 17, 23 | 18 +/- 2.6 | |
| | 55˚C[3] 1 hr | 0, 0, 0[3] | 0 +/- 0 | 0.002439 |
| *ssa1D ssa2D + STE5-MYC9 2m* | | | | |
| | RT 1–2 hr | 1,1,3 | 1.7 +/- 0.7 | |
| | 55˚C[3] 1 hr | 0, 0, 0[3] | 0 +/- 0 | 0.066767 |
| *cdc24-1+ CDC24 + STE5-MYC9 2m* | | | | |
| | RT 2–3 hr | .2, 2,0.25, 2,1.8 | 1.2 +/- 0.4 | |
| | 37˚C 2–3 hr | 1,6.8,6,1,4 | 3.8 +/- 1.2 | 0.086776 |
| | 55˚C 1–3 hr | 14,24,27,17,31 | 23 +/- 3.2 | 0.000147 |
| FY23/S288c | RT 3 hr | 37.9, 69.7 | 53.8 +/- 15.9 | |
| | 37˚C 3 hr | 80.4, 91.1 | 85.75 +/- 5.4 | 0.197142 |

[1]Strains: EY3376 is a spontaneous, ura3- derivative of EY3136 that reverts at low frequency transformed with pSKM19 (*STE5-MYC9 URA3 2m).*

EYL2712 YMY467 (1,2,3) is EYL457 = YMY467 = [RAY914 *MATa cdc24-1::LoxPHIS5SpLoxP his3-200 leu3-3, 112 ura3-52 trp1-901 lys2-801 gal2 suc2-9* (pRS414 *TRP1-CEN-CDC24*)] harboring pSKM19 (*STE5-MYC9 URA3 2m).* Cells were grown in units of cells were pelleted and resuspended in pre-warmed YEPD medium and incubated with shaking at the indicated temperature. Cells were collected and prepared for indirect immunofluorescence using 9E10 monoclonal antibody and DAPI.

[2]N>C means the 9E10 signal in the nucleus is stronger than in the cytoplasm.

~133–400 cells were counted for data points. The Ste5-Myc9 signal was weaker in the *ssa1D ssa2D* cells than in the wild type strains.

[3]At 55˚C (for 1–3 hours) cells of this strain background appear enlarged with a large vacuole devoid of Ste5-Myc9 signal and some Ste5-Myc9 signal beneathcell periphery. The cells appeared to be dying.

[4]S.E. is standard error.

[5]p-value significance tests were calculated with a Student's t-test of two independent means, two-tailed with the social science statistics calculator at socscistatistics.com/tests/studenttest/default.aspx (copyright Jeremy Stangroom 2023).

together promote nuclear accumulation of Ste5 at room temperature and 30˚C in the Craig strain background, but increases in nuclear accumulation at elevated temperatures appear to still occur independently of Ssa1 and Ssa2.

## Overexpression of Ssa1 promotes punctate foci localization of a Ste5 VWA domain mutant

Given that protein quality control can occur in supramolecular foci and the tendency of Ste5 to aggregate, we examined whether Ste5 localizes in punctate structures that are affected by loss of Ssa1 and Ssa2 or by overexpression of Ssa1. Proteins that are either soluble or aggregated with or without ubiquitinylation are corralled into distinct foci by Ssa1, Ssa2 and co-chaperones where they are either refolded or routed for degradation in the vacuole or through autophagy [65–67]. We first examined the distribution of our Ssa1-GFP protein in cytoplasm, nucleus and punctate foci in wild type and *ssa1D ssa2D* cells as well as that of Ssa1-GFP and Ssa2-GFP fusion proteins in the database images of O'Shea and Weissman labs that are hosted by SGD [98]. We found Ssa1-GFP to localize throughout the cytoplasm and nucleus as shown by absence of nuclear exclusion (i.e. no areas are devoid of GFP signal; Fig 7A, Table 3). Ssa2-GFP is thought to localize in the cytoplasm and the nucleus whereas Ssa1-GFP is thought to localize only in the cytoplasm [98]. However, we observed no obvious nuclear accumulation for Ssa1-GFP and Ssa2-GFP in the O'Shea and Weissman GFP database images hosted by SGD which showed localization patterns quite similar to what we observe with our Ssa1-GFP

**Table 3. Prevalence of Ste5 in punctate foci.**

| Strain/Gene/Protein[1] | %Punctate spots[2] | %Nuc. Accum[3] | %Cortical[4] |
|---|---|---|---|
| A. *ADH1prom-SSA1-GFP* | | | |
| EY957 *bar1D* RT (N = 2) | 0 +/- 0 | 0.8 +/- 0.4 | 0 +/- 0 |
| EY3141 *ssa1D ssa2D* RT (N = 3) | 0 +/- 0 | 82.7 +/- 5.5 | 0 +/- 0 |
| B. GFP database of Huh *et al* [98].[5] | | | |
| yeastgfp.yeastgenome.org/display | | | |
| LocImage.php?loc = 15528): | | | |
| S288c + Ssa1-GFP (32 cells) | 3 (1 cell) | 0 | 0 |
| yeastgfp.yeastgenome.org/display | | | |
| ocImage.php? loc = 13306: | | | |
| S288c + Ssa2-GFP (71 cells) | 5.6 (4 cells) | 1.4 (1 cell) | 0 |
| C. *ste5D + STE5promSTE5-MYC9* | | | |
| (indirect immunofluorescence) | | | |
| W303a *ste5D*, 30˚C | 0 +/- 0 | 23 +/- 5.3 | 0.3 +/- 0.3 |
| p-value, N = 4 | 1 | | |
| W303a *ste5D*, a factor, 30˚C | 0.2+/-0.2 | 8.9+/-3.3 | 20.3+/-3.4 |
| p-value, N = 7 | 0.424114 | | |
| W203a *ste5D*, a factor, RT | 0 | 0 | 29.4 |
| no p-value, N = 1 | | | |
| D. *STE5 + CUP1prom-STE5-MYC9* | | | |
| (indirect immunofluorescence) | | | |
| W303a,S288c, 30˚C | 0 +/- 0 | 7.7 +/- 7.7 | 0 +/- 0 |
| p-value, N = 2 | 1 | | |
| E. *STE5 + STE5-MYC9-2m* | | | |
| (indirect immunofluorescence) | | | |
| W303a, 30˚C | 0 +/- 0 | 19.5 +/- 7.6 | 0.7 +/- 0.7 |
| p-value, N = 4 | 1 | | |
| F. *ste5D + Ste5(1–242)-GFP-GFP* | | | |
| RING-H2 (live)[6] | | | |
| Ste5(1–242)-GFP2, RT | 11.4 +/- 3.9[6] | 84.1 +/ -6.7 | 37.5 +/- 19.2 |
| p-value, N = 7[7] | 0.040835 | | |
| Ste5(1–242)-GFP2, 30˚C | 5.3 +/- 3.4 | 80.1 +/- 4.8 | 4.3 +/- 4.3 |
| p-value, N = 8 | 0.30671 | | |
| Ste5(1–242)-GFP2, 37˚C | 16.3 +/- 5.2 | 85.7+/-5.4 | 0 +/- 0 |
| p-value, N = 2 | 0.001903 | | |
| G. *ste5D + ADH2prom-TAgNLS-* | | | |
| *-GFP-GFP* (live) | | | |
| W303a *ste5D*, 30˚C | 0 +/- 0 | 100 +/- 0 | 0 +/- 0 |
| p-value, N = 3 | 1 | | |
| H. *ste5D + ADH2prom-TAgNLS-* | | | |
| *NES(PKI)-GFP-GFP* (live) | | | |
| W303a *ste5D* 30˚C | 0 +/- 0 | 0 +/- 0 | 0 +/- 0 |
| p-value, N = 2 | 1 | | |
| I. *ste5D + STE5prom- STE5-GST* | | | |
| (indirect immunofluorescence) | | | |
| Ste5-GST, RT | 1.4 +/- 1 | 0.9 +/- 0.9 | 32.3+/-11.6 |

(*Continued*)

**Table 3.** (Continued)

| Strain/Gene/Protein[1] | %Punctate spots[2] | %Nuc. Accum[3] | %Cortical[4] |
|---|---|---|---|
| p-value, N = 5 | 0.236212 | | |
| Ste5-GST, RT, a factor | 1.8 +/- 1.8 | 19.0+/-19.0 | 28.6+/-4.8 |
| p-value, N = 3 | 0.219944 | | |
| Ste5-GST, 37oC (N = 1) | 0 | 5 | 5 |
| Ste5-GST, 37˚C, a factor (N = 1) | 0 | 5 | 20 |
| J. *ste5D + STE5prom-TAgNLS-* *-STE5-GST* | | | |
| W303a *ste5D*, 30˚C | 0 +/- 0 | 98.3 +/- 1.7 | 0 +/- 0 |
| p-value, N = 2 | 1 | | |
| K. *ste5D + STE5prom-ste5-L610/614/* *634/637A-MYC9-2m* | | | |
| W303a *ste5D*, 30˚C | 1.4 +/- 1.4 | 0 +/- 0 | 0 +/- 0 |
| p-value, N = 3 | 0.014993 | | |
| L. *ste5D + STE5prom-TAgNLS-ste5-* *-L610/614/634/637A-MYC9-CEN* | | | |
| W303a *ste5D*, 30˚C | 2 +/- 2 | 74 | 0 |
| p-value, N = 2 | 0.078141 | | |
| M. *ste5D + STE5prom-ste5-* *-L610/614/634/637A-GST-2m* | | | |
| W303a *ste5D* 30˚C | 8.7 | 78.9 | 0 |
| W303a *ste5D*, 30˚C, + aF | 10 | 87 | 0 |
| no p-value, N = 1 | | | |
| N. *ste5D + STE5prom-ste5-* *-L610/614/634/637A-MYC9-2m* | | | |
| W303a *ste5D*, 30˚C | 6.9+/-3.8 | 52.0+/-27 | 0 +/- 0 |
| p-value, N = 3 | 0.048718 | | |
| O. *ste5D + STE5prom-Ste5-* *-L610/614/634-/637A-Myc9-2m* *+ GAL1prom-SSA1-GFP* | | | |
| W303a *ste5D*, 30˚C | 43.2+/- 4.6 | 9.6+/-6.3 | 0 +/- 0 |
| p-value, N = 4 | 0.000082 | | |
| P. *ste5D + STE5prom-Ste5-* *L582/585A-MYC9-2m* *+ GAL1prom-SSA1-GFP* | | | |
| W303a *ste5D* 30˚C | 0 | 24.2 | 0 |

*(Continued)*

**Table 3.** (Continued)

| Strain/Gene/Protein[1] | %Punctate spots[2] | %Nuc. Accum[3] | %Cortical[4] |
|---|---|---|---|
| no p-value, N = 1 | | | |

[1]Cells were grown in SC selective medium containing 2% dextrose, either lacking uracil, tryptophan and /or leucine at 30°C or at the indicated temperature. Cells were grown to logarithmic phase at an $A_{600}$ of 0.2–0.5 adjusted for equal cell density, ~$A_{600}$ 0.5 then treated with α factor. Where indicated, cells were transferred to pre-warmed medium at 37°C and incubated for 3 hours. For the *GAL1* promoter, cells were pre-grown in pre-warmed medium containing 2% raffinose prior to being transferred to pre-warmed medium containing 2% galactose and then grown for 5 hours.

[2]Cells were fixed with 10% formaldehyde and processed for indirect immunofluorescence to detect Myc tagged proteins and DAPI stained DNA or were visualized live for GFP tagged proteins as described (S3, S4).

The analysis used captured images from published and unpublished work.

[2]The % punctate spots is the percentage of cells that harbored 1 or more spots.

[3]Nuclear accumulation means that the 9E10 signal in the nucleus is equal to or greater than the 9E10 signal in the cytoplasm. In some instances, nuclear localization is an estimation done without DAPI co-staining or localization of a protein known to be nuclear. Nuclear exclusion means that the 9E10 signal is less than the 9E10 signal in the cytoplasm.

[4]The cortical accumulation in all cases was asymmetric and if a shmoo was present, it was located at the shmoo tip.

[5]The Yeast GFP fusion database of Erin O'Shea and Jonathan Weissman laboratories [98] is hosted by SGD at yeastgfp.yeastgenome.org. The localization of Ssa1-GFP and Ssa2-GFP appears to have an uneven lacy pattern without obvious nuclear accumulation. Note that nuclear accumulation is an estimate because there is no DAPI.

[6]Ste5(1–242)-GFP2 was visualized live in cells grown at RT, 37°C and by indirect immune-fluorescence on fixed cells that were grown at 30°C. The GFP$^+$ cytoplasmic pool is more prominent than the nuclear pool in fixed cells than in live cells.

[7]Values shown are the mean +/- standard error.

[8]p-values were calculated with a student's T-test (two independent means, two-tailed). The p-values were calculated by comparing the punctate foci values of the various Ste5 derivatives with those of *ste5D + STE5-MYC9-2m* grown at 30°C (i.e. 0,0,0,0,0 (N = 5)).

on the Axioscope 2 (Table 3). Strikingly, Ssa1-GFP became clearly enriched in what we presume to be nuclei in the EY3141 *ssa1D ssa2D* cells which lack endogenous pools of Ssa1 and Ssa2 that might compete with Ssa1-GFP for nuclear anchoring sites (Fig 7B, Table 3). Collectively, these observations suggest that, Ssa1 and Ssa2 are likely to have substrates in the cytoplasm and the nucleus and could potentially regulate Ste5 in either compartment. Although Ssa1 has been found to co-localize with aggregated proteins, we did not find it to localize in punctate foci in our images, although several cells in the O'Shea and Weissman public database images had a punctate spot (Table 3).

By live cell microscopy, Ste5-YFP and GFP-Ste5 localize in numerous punctate foci using a Nikon TE2000E [24], whereas Ste5-Myc9 localizes in an irregular pattern of localization throughout the cytoplasm and nucleus by indirect immunofluorescence using a Zeiss Axioscope 2 microscope and appears mottled with punctae using a BioRad 1024 multiphoton laser confocal microscope [35]. We reexamined Ste5-Myc9 and several derivatives for evidence of punctate foci using the Zeiss Axioscope 2 (see Materials and Methods for list of mutants). By indirect immunofluorescence, Ste5-Myc9 was not detected in punctate foci during vegetative growth at 30°C in either W303a or S288c strain backgrounds (Fig 7E, Table 3 % of total cells with punctate foci: W303a *ste5D* at 30°C: 0 +/-0 S.E. (N = 4), W303a 0 +/- 0 S.E. (N = 4), S288c 0 +/- 0 S.E. (N = 2). Punctate foci were detected at a very low level after α factor treatment, although this was not considered a statistically significant difference by p-value (Table 3, 0.2 +/- 0.2 S.E. (N = 7), p-value 0.424224). We could not quantify punctate foci for Ste5 in the in the *ssa1D ssa2D* strain because of low abundance (i.e. GFP-Ste5, Ste5-Myc9, and TAgNLSK128-Ste5-Myc9).

By contrast, it was possible to visualize punctate foci in several mutant derivatives of Ste5 using the Axioscope 2, suggesting these derivatives have a greater ability to aggregate or oligomerize. Nearly 100% of cells expressing Ste5C180A-Myc9 mutant that can not dimerize

through the RING-H2 domain exhibited punctate foci (see Fig 3 in reference 35). In addition, a Ste5(1–242)-GFP2 fusion that oligomerizes through the RING-H2 domain localized in punctate foci in the cytoplasm by live cell imaging (Fig 7F; at 30°C the % cells with punctate foci was 5.3 +/- 3.4 S.E. (N = 8), Table 3). These punctate foci were due to the RING-H2 fragment because none were detected for TAgNLS-NES-GFP-GFP and TAgNLS-GFP-GFP fusion proteins that localize to the nucleus and cytoplasm (Table 3). A Ste5-GST fusion that forms more dimers from GST also formed readily detectable punctate foci during vegetative growth (Table 3, 1.4% +/-1.0 S.E. (N = 5)) and after α factor stimulation (1.8 +/- 1.8 S.E. (N = 3)) by indirect immunofluorescence with anti-GST antibodies (Table 3). Strikingly, we also found that a VWA domain quadruple point mutant, ste5L610/614/634/637A-Myc9 (Fig 7G–7I) formed more punctate foci in the cytoplasm (Fig 7J, Table 3, punctate foci in 1.4 +/- 1.4 S.E. percent of cells (N = 3), p-value .014993, Table 3). A nucleus-enriched derivative, TAgNLS-ste5L610/614/634/637A-Myc9 formed punctate foci in 2 +/-2 S.E.% of cells (N = 2) (Fig 7K, p-value .078141, Table 3) and a GST-ste5L610/614/634/637A derivative formed punctate foci in 8.7% of cells (N = 1) (Table 3). Thus, the ste5L610/614/634/637A mutations in the VWA-like domain increases the propensity of Ste5 to localize in punctate foci.

Notably, when *STE5prom-ste5L610/614/634/637A*-was co-expressed with *GAL1prom-SSA1*, the ste5L610/614/634/637A-Myc9 protein accumulated in a greater number of punctate foci and larger patches of varying sizes in 43.2 +/- 4.6 S.E.% of cells (Fig 7L, Table 3, p-value .000082). Co-staining with DAPI revealed that the punctate foci did not overlap the nuclear DNA (S4 Fig; Table 3). The effect of Ssa1 inducing the localization of ste5L610/614/634/637A-Myc9 into punctate foci is a hallmark of Hsp70 chaperones moving misfolded proteins into inclusion bodies for refolding or degradation [40,65–67]. These data provide additional evidence of a role for Ssa1 in Ste5 homeostasis. We can not distinguish whether the areas of greater intensity designated as punctate foci or patches involve multiple foci or a single large fluorescing structure.

Ssa1 and Ssa2 bind misfolded proteins on hydrophobic patches of certain characteristics that have flipped to the aqueous surface of a protein. We therefore used bioinformatics to examine whether the L610A, L614A, L634A and L637A mutations are predicted to influence either the folding of Ste5 or the number of potential Ssa1 binding sites. The four mutations fall in evolutionarily conserved residues in Ste5 within the VWA-like domain and nearby oligomerization domain (S3K Fig), suggesting their effects on Ste5 have general implications. Atomic interaction predictions suggest all four alanine substitutions reduce intramolecular interactions that might normally stabilize the protein (S3F–S3I Fig). Alanine substitutions, which are less hydrophobic and less bulky than leucines (S5A–S5C Fig), are predicted to be less buried (S5D and S5E Fig). The region overlapping the Ste5L610/614/634/637A mutations is predicted to have slightly different crystallization properties than wild type Ste5 based on XtalPred analysis (Table 4, (S24)) and somewhat higher tendency for disorder based on IUPRED and IUPRED2/ANCHOR analysis; S5F–S5J Fig).

We asked whether there could be a difference in the predicted ability of Ste5L610/614/634/637A to bind Hsp70 proteins. Hsp70 proteins, including Ssa1, recognize client proteins through short hydrophobic segments of 5–7 amino acids flanked by positively charged amino acids [53] which typically occur every 30–40 amino acids [99]. BiPPred analysis has successfully predicted Hsp70 Kar2 binding sites [99] and a Ssa1 binding site in a peptide [99,100]. Interestingly, BiPPred predicted 13 fewer highly optimal recognition sequences for Hsp70 on the Ste5L610/614/634/637A polypeptide compared to wild type Ste5 (i.e 107 sites with highly optimal recognition scores of 0.9–1.0 for Ste5 versus 94 for Ste5L610/614/634/637A; (S4L Fig). These observations are certainly consistent with a role for Ssa1 in Ste5 homeostasis and kindle speculation that the mutated Ste5L610/614/627/634 VWA domain has less stable regions that are not as well recognized by Hsp70 chaperones for folding.

**Table 4. XtalPred analysis of Ste5 and ste5-610A/614A/634A/637A fragments.**

| Protein[1] | EP-C | RF-C | Gravy | Instability | pI | % Coil | LDR | Length |
|---|---|---|---|---|---|---|---|---|
| >gi|94969271| | 1 | 3 | -0.05 | 37.48 | 5.74 | 35 | 6 | 135 |
| >gi|117928536| | 5 | 11 | -0.1 | 44.74 | 10.69 | 46 | 33 | 173 |
| Ste5(1–917) | 5 | 1 | -0.42 | 48.04 | 5.26 | 72 | 169 | 917 |
| Ste5Ms(593–786)[2016] | 3 | 11 | -0.27 | 32.34 | 5.02 | 44 | 18 | 194 |
| Ste5Ms(593-786wt)[2021] | 2 | 11 | -0.3 | 32.34 | 5.02 | 45 | 16 | 194 |
| Ste5VWA(583–787) | 3 | 11 | -0.31 | 32.12 | 4.88 | 44 | 18 | 196 |
| Ste5VWA(583–786) | 1 | 1 | -0.26 | 35.76 | 4.93 | 43 | 16 | 204 |
| Ste5RING(177–229) | 5 | 10 | -0.04 | 66.68 | 7.58 | 60 | 1 | 52 |
| Ste5RING(170–230) | 5 | 10 | 0 | 69.99 | 8.28 | 68 | 0 | 60 |
| Ste5PH(318–588) | 3 | 9 | -0.19 | 28.73 | 4.26 | 41 | 4 | 131 |
| Ste5(551–700) | 3 | 1 | -0.27 | 40.5 | 7.08 | 41 | 16 | 150 |
| Ste5(551-700mutant[2]) | 3 | 11 | -0.21 | 43.39 | 7.08 | 41 | 16 | 150 |
| Ste5(581–786) | 4 | 11 | -0.29 | 37.38 | 5.03 | 44 | 24 | 206 |
| Ste5(581-786mutant) | 2 | 11 | -0.33 | 35.28 | 5.03 | 44 | 17 | 206 |
| Ste5(583–786) | 1 | 1 | -0.26 | 35.76 | 4.93 | 43 | 16 | 204 |
| Ste5(583-786mutant) | 1 | 1 | -0.3 | 33.64 | 4.93 | 46 | 17 | 204 |
| Ste5(593–786) | 2 | 11 | -0.3 | 32.34 | 5.02 | 45 | 16 | 194 |
| Ste5(593-786mutant) | 1 | 1 | -0.32 | 30.11 | 5.02 | 45 | 17 | 194 |

[1]XTalPred-RF analysis (S1) of Ste5 AAA35115.1 predicts crystallization probability and has two crystallization class indexes, EP-C and RF-C (Random Forest)(S16) that are based on nine physiochemical properties and secondary structure predictions. For EP-C, 1 is optimal, 2 is suboptimal, 3 average, 4 difficult and 5 is very difficult. For RF-C, 1 is best, 11 is worst. Gravy is a hydropathy index; positive value indicates hydrophobic, negative value indicates hydrophilic. % Coil is the predicted percentage of coiled coil structure. LDR is longest disordered region. The Instability index suggests stability if it is less than 40. The XtalPred positive control is >gi|94969271| and the negative control is >gi|117928536|. Ste5Ms (593–786) and Ste5(583–786) have been crystallized (S7, 3FZE, 4F2H). The RING-H2 domain is residues 177–229, the PH domain is residues 388–518, VWA domain is residues 583–787.

[2]Ste5mutant is Ste5-610A/614A/634A/637A.

## Strains derived from Craig lab diploid MW63 lack Kss1 by western analysis

During experiments to assess Fus3 and Kss1 activation, we discovered that EY3136 wild type and EY3141 *ssa1D ssa2D* ascospores from the Craig lab diploid MW63 are kss1- based on absence of detection of Kss1 protein with anti-Kss1 antibody and anti-phospho-p42p44 antibody that recognizes a conserved peptide in human ERK2 that has homology to Fus3 and Kss1 (S2D Fig). With the anti-Kss1 antibody, we did not detect Kss1 in EY3136 and EY3141 or some of their progeny (Fig 8A, e.g. WT-A ascospore, compare to *kss1D* and *fus3D* controls). However, Kss1 was detected in the W303a positive control strains EY957 *bar1D* and EY940 *bar1D fus3D* and several other *MATa bar1D* ascospores from the crosses to BY4742 (Fig 8A, e,g, ascospores *bar1D*-B and *ssa1D ssa2D*-B;). We also failed to detect Kss1-P in EY3136 and EY3141 and several ascospore progeny with the anti-phosphop42p44 antibody, although active Fus3-P and Kss1-P were detected in control strains (Fig 8B lanes 6,7,13 are WT-B and *ssa1D ssa2D*-B; +aF strains were treated with 50 nM a factor for 1 hour, 100 mg of WCE overloaded on minigel). It has previously been established by E. Elion that some laboratory strains lack Kss1 [101].

## Loss of Ssa1 and Ssa2 reduces the ability of Fus3 and Kss1 to be activated

We assayed activation of Fus3 and Kss1 in a time course experiment in the *bar1D Kss1*+ strains WT-B and *ssa1D ssa2D*-B. Basal levels of active Fus3 and Kss1 were not detected in either WT-B or *ssa1D ssa2D*-B (Fig 8C and 8D, compare to control lanes with extracts from W303a

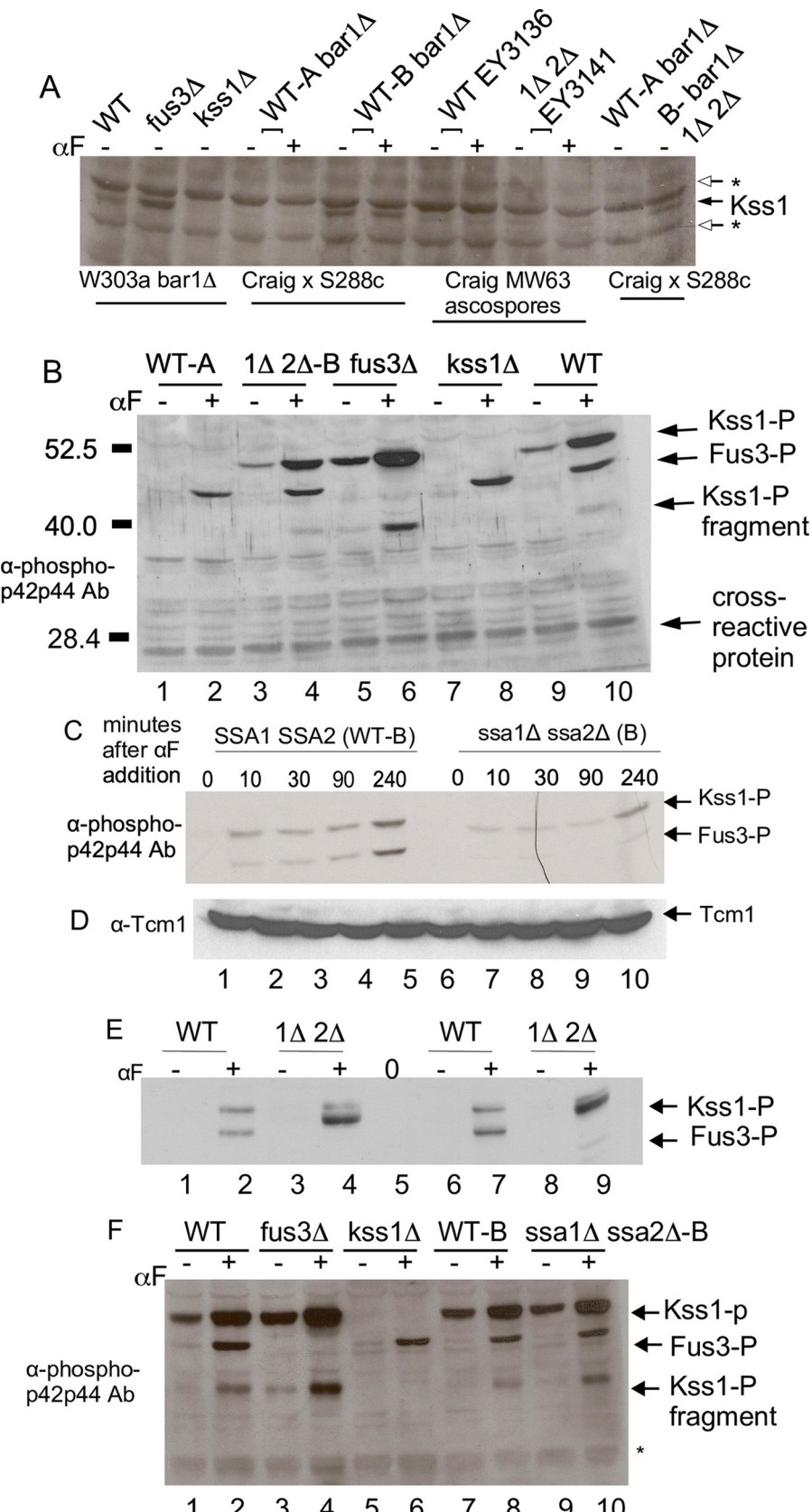

**Fig 8. Effect of *ssa1D ssa2D* mutations on activation of Kss1 and Fus3.** A. Kss1 protein is not detected in EY3136 WT and EY3141 *ssa1D ssa2D* strains derived from MW63. Immunoblot of WCE from logarithmically dividing cells grown at room temperature. Kss1 was detected with anti-Kss1 polyclonal 6775-kss1-yc-19a. Kss1 full-length and a fragment of Kss1 are indicated with arrows. The asterisks indicate cross-reacting proteins. A shorter exposure of the blot is in S6B Fig. Strains: W303a strains EY957 *bar1D*, EY940 *bar1D fus3D* and EY1119 *bar1D kss1D*, MW63 (EY3136 and EY3141) X Research Genetics 11408 (S288c) ascospores: A-WT *bar1D*, B-WT *bar1D*, MW63 progeny parents: EY3136 WT, EY3141 *ssa1D ssa2D*, ascospore A-WT *bar1D*, ascospore B- *bar1D ssa1D ssa2*. B. immunoblot of active Fus3 and Kss1 in *bar1D KSS1* and *bar1D kss1-* derivatives of EY3136 WT and EY3141 *ssa1D ssa2D*. Cells were grown logarithmically at room temperature before and after 1 hour exposure to 50 nM a factor. Active Fus3 and Kss1 detected with anti-phospho-p42p44 antibody with 100 mg of WCE loaded on a minigel. Strains: ascospore A-WT *bar1D*, ascospore B-*bar1D ssa1D ssa2D*, W303a strains: EY957 *bar1D*, EY940 *bar1D fus3D* and EY1119 *bar1D kss1D*. C-D. Time course of a factor activation of *bar1D* and *bar1D ssa1D ssa2D* strains. Ascospore B-WT *bar1D* and ascospore B-*bar1D ssa1D ssa2D* with detectable Kss1 protein were prepared after exposure to 150 nM α factor for 0,10, 30, 90 and 240 minutes and 50 mg of WCE samples were run on the minigel. The immunoblot was probed anti-phospho-p42p44 antibody (C) then stripped and reprobed with Tcm1 monoclonal antibody (D). E. Immunoblot of large amounts of WCE to detect basal Fus3 and Kss1 activation. Lanes 1–4 have 150 μg of WCE, lanes 5–8 have 300 μg WCE. The WCEs were precipitated with trichloroacetic acid to concentrate. E. Activation of Fus3 and Kss1 in *ssa1D ssa2D* with high a factor. Cells were grown as in B and exposed to 150 nM a factor for 10 minutes then processed as in B with 100 mg WCE loaded on minigel. Strains: W303a EY957 *bar1D*, EY940 *bar1D fus3D* and EY1119 *bar1D kss1D*, ascospore B-WT *bar1D* and ascospore B- *bar1D ssa1D ssa2D*. F. Immunoblot of active Fus3 and Kss1 after higher a factor treatment. Cells were grown as in B using 150 nM a factor for 10 minutes. 100 mg of total protein of WCE was run on the minigel. Samples were probed as in B. Strains are as in D.

*FUS3 KSS1*, *fus3D* and *kss1D*). Strikingly, the levels of α factor-induced active Fus3 and Kss1 were reduced in the *ssa1Δ ssa2Δ* ascospore B mutant during vegetative growth and at all time points from 10 minutes to 2 hours after addition of 50 nM α factor (Fig 8B, S2 Table). The decrease in the band signal was not from less protein loaded based on ribosomal protein Tcm1 (Fig 8B). Analysis of more *ssa1Δ ssa2Δ* cell extract did not reveal basal Fus3-P, whereas basal Kss1-P was detected in wild type and *ssa1Δ ssa2Δ* whole cell extracts from the *KSS1* strains (Fig 8D). Densitometry confirmed that the *ssa1Δ ssa2Δ* strain had undetectable basal levels of active Fus3 and less active Fus3 during α factor stimulation compared to wild type (S2 Table). Kss1 activation during α factor stimulation was reduced, although less impaired than that of Fus3 in the *ssa1Δ ssa2Δ* strain (Fig 8B and 8C), but the abundance of Kss1 was not reduced (Fig 8A, S2 Table). Active Fus3 and Kss1 were more readily detected in the *ssa1D ssa2D* strain at the later time point, suggesting the loss of Ssa1 and Ssa2 reduces the efficiency of signaling. We were able to detect Fus3 and Kss1 activation after a 10 minute exposure time by increasing the level of a factor three-fold to 150 nM and overloading 100 mg WCE on the minigel (Fig 8E). Collectively, these results suggest that the *ssa1D ssa2D* defect reduces the activation of Fus3 and Kss1 and may raise the threshold at which cells efficiently activate the pathway.

## Ssa1 and Ssa2 are not required for feedback phosphorylation of Ste7 but are required for efficient Fus3-HA kinase activity

Ste7-Myc is hyperphosphorylated in the presence of α factor through feedback phosphorylation by Fus3 and Kss1. We looked at basal feedback phosphorylation of Ste7-Myc by Fus3 using a *CYC1prom-STE7-MYC* gene that is not regulated by either the mating pathway or heat shock proteins Ssa1, Ssa2, Hsf1. The ratio of phosphorylated Ste7-Myc to hypophosphorylated Ste7-Myc was nearly identical in wild-type and *ssa1D ssa2D* strains with or without cycloheximide to reveal primary signaling events (Fig 9A, S2 Table, slower mobility protein band is compared to faster mobility phosphorylated protein bands). Kss1 is not likely to be responsible for the basal feedback phosphorylation of Ste7-Myc in the wild type EY3136 and EY3141 *ssa1D ssa2D* strains because both strains are kss1- by immunoblot analysis with anti-Kss1 antibodies and anti-phospho-p42p44 antibodies (Fig 8). Since Ste7 feedback phosphorylation does not occur in a *fus3D kss1D* double mutant, basally active Fus3 must be responsible for the

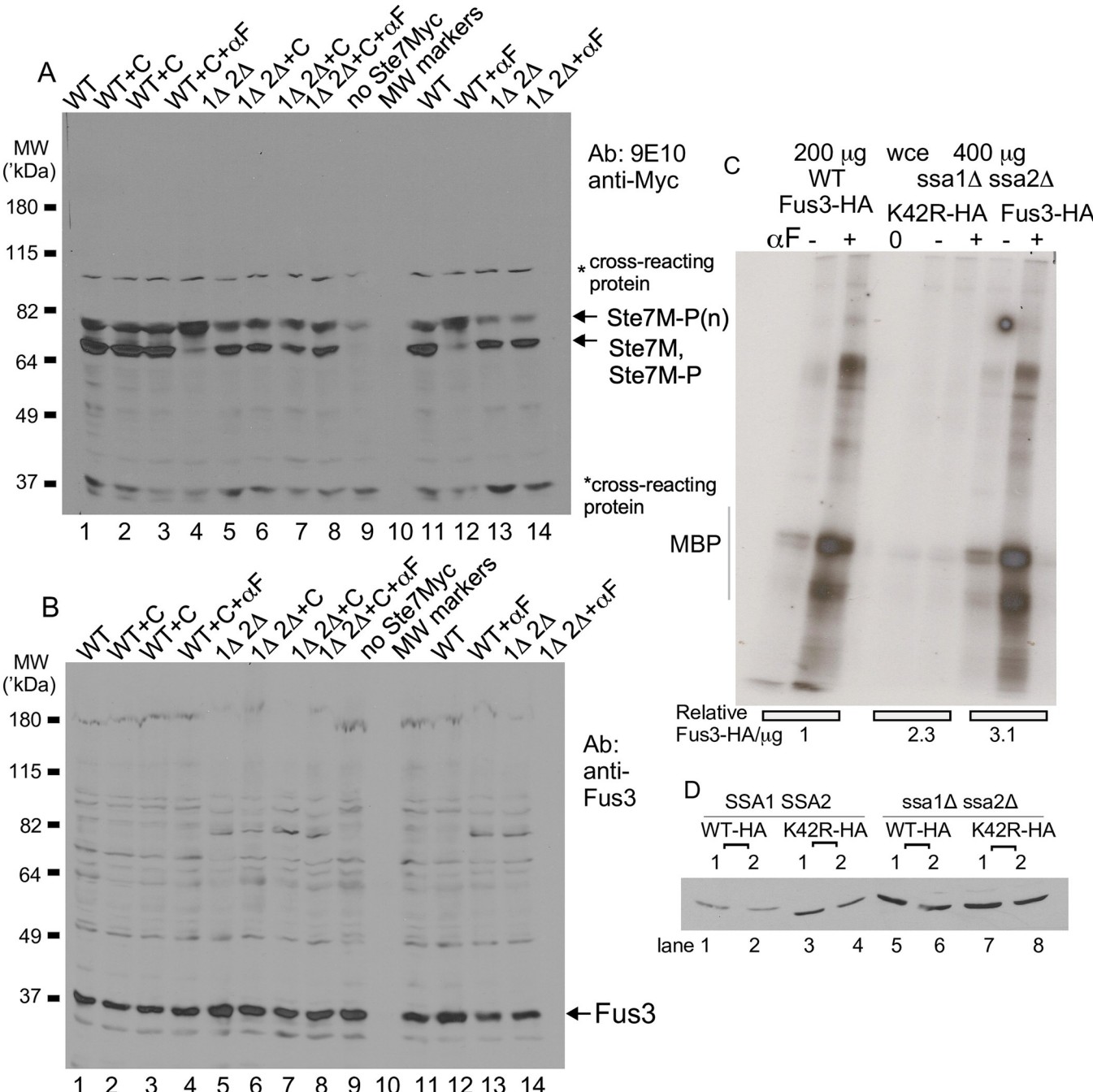

**Fig 9. Effect of *ssa1D ssa2D* mutations on Fus3 abundance, Ste7 feedback phosphorylation and Fus3 kinase activity.** A. Abundance of hypo- and hyper-phosphorylated Ste7. Wild-type and *ssa1D ssa2D* cells harboring *CYC1-STE7-MYC-CEN* were grown to mid-logarithmic phase then treated with 10 µg/ml cycloheximide for 10 minutes where indicated (+). 1 mg of WCEs were concentrated by 40% ammonium precipitation for analysis. The blot was probed with 9E10 antibodies. B. Abundance of Fus3. The same extracts prepared in A. were subjected to immunoblot analysis exactly as in A. and probed with anti-Fus3 polyclonal antibodies. C. Fus3 kinase assays with Fus3-HA#5 and fus3K42R-HA#5. Ascospore B-*MATa bar1D* and ascospore B-*MATa bar1D ssa1D ssa2D* strains expressing *FUS3-HA* (pYEE121) and *fus3K42R-HA* (pYEE128.30–1) were grown logarithmically at room temperature with or without 50 nM α factor for 60 minutes. WCEs were assayed for Fus3 kinase activity in an immunoprecipitation-coupled reaction using 200 µg of WCE wild type and 400 µg WCE for *ssa1D ssa2D*, with myelin basic protein (MBP) added to the kinase reaction. MBP is an *in vitro* substrate for human ERK and Fus3. Fus3 kinase assay. Relative levels of Fus3-HA#5 and fus3K42R-HA#5 are shown below the kinase assay. D. Abundance of Fus3-HA#5 and fus3K42R-HA#5. The immunoblot samples are two transformants each for ascospore B-*MATa bar1D* and ascospore B-*MATa bar1D ssa1D ssa2D* harboring *FUS3-HA#5-CEN* (pYEE102) and *fus3K42R-HA#5-CEN* (pYEE106). 12CA5 was used for detection. Equal total protein was loaded on the gel.

feedback phosphorylation of Ste7 (Fig 9A). The abundance of endogenous Fus3 was slightly higher in the EY3141 *ssa1D ssa2D kss1-* strain compared to wild type EY3136 *kss1-* which could enhance the pool able to feedback phosphorylate Ste7-Myc (Fig 9B, S2 Table). Thus, Ssa1 and Ssa2 are essential for basal activation of Fus3 and full basal activation of Kss1 and essential for α factor-induced activation of both Fus3 and Kss1 but have little effect on Ste7.

We assessed the ability of Fus3 to phosphorylate associated substrates in Fus3-HA co-immunoprecipitation kinase assays of extracts from WT-B *bar1D* and *bar1D* and *ssa1D ssa2D*-B cells that had been exposed to 50 nM α factor for one hour. The profile of Fus3-HA phosphorylated co-immunoprecipitated substrates and exogenously added myelin basic protein (MBP) was similar for both strains (Fig 9C and 9D) and was dependent on Fus3 catalytic activity (Fig 9C and 9D fus3K42R-HA, lanes 4,5). The level of Fus3-HA activity in the *ssa1D ssa2D* WCE was ~16.7% that of the wild type control (twice as much WCE was immunoprecipitated for the *ssa1D ssa2D* strain which had 3-fold higher abundance of the Fus3-HA protein; Fig 9F; S2 Table). Thus, a smaller pool of Fus3-HA enzyme is active in the *ssa1D ssa2D* strain.

## Ssa1 and Ssa2 have either inhibitory or negligible effects on abundance of Ste7, Fus3, and Kss1

We determined whether loss of Ssa1 and Ssa2 has obvious effects on the abundance of MAPKK Ste7 and MAPKs Fus3 and Kss1. We did not examine MAPKKK Ste11 whose abundance declines during prolonged a factor stimulation and may be negatively regulated by Ste5, Ubc4 and Ubc5 [102] and is known to be positively regulated by Hsc82, Hsp82, Sti1, Sse1, and Ydj1 [103–105]. The large inhibitory effect of the *ssa1D* and *ssa2D* mutations on Ste5 abundance was specific to Ste5, based on comparative analysis of Fus3, Ste7, Kss1 and Tcm1. The abundance of Fus3 was not obviously reduced in the *ssa1D ssa2D* double mutant, even when Ste5 had been overexpressed (Fig 5C, S2 Table). Fus3 abundance was approximately the same or slightly higher in a *ssa1D ssa2D* double mutant compared to wild-type (S2 Table). The level of Fus3 appeared elevated in a *ssa1D ssa2D* double mutant when Fus3 was active or catalytically inactive (i.e. Fus3K42R-HA and Fus3-HA; Fig 9D). Fus3 was slightly elevated when the *ssa1D ssa2D* strain co-expressed the *CYC1prom-STE7-MYC* gene, either before and after blocking translation with cycloheximide (Fig 9B, S2 Table). The increase in endogenous Fus3 ranged from 1.2 +/- 0.12 S.E. to 1.5 +/- 0.65 S.E. for nontagged Fus3 to several-fold for HA-tagged Fus3 (S2 Table; 3.06 +/- 0.28 S.E. Fus3-HA versus 2.26 +/- 0.62 for fus3K42R-HA). Thus, Ssa1 and Ssa2 have an inhibitory effect on Fus3 abundance that is independent of Fus3 catalytic activity. This is consistent with unconfirmed high throughput data suggesting Ssa1 may associate with Fus3 and suggests the Hsp70 proteins may promote its degradation. An increase in abundance of Fus3 in the *ssa1D ssa2D* double mutant might counteract somewhat a reduction in the level of Ste5 protein, as would an increase in Ste11, Ste7 and Ste12 resulting from reduced signaling through Ste5 [102].

The abundance of Kss1 was also slightly elevated in the *ssa1D ssa2D* double mutant to ~1.24 the level in the isogenic wild type strain. However, a caveat of this interpretation is high background from the anti-Kss1 antibodies, which interfered with densitometry on replicates. A *fus3D* control strain had a larger 2.59-fold increase in Kss1, consistent with the known repressive effect of Fus3 on Kss1 expression (Fig 8A, S2 Table). By contrast, Ste7-Myc abundance was not obviously altered in the *ssa1D ssa2D* mutant, (Fig 9A, S2 Table). The relative abundance of phosphorylated and unphosphorylated species of Ste7-Myc was the same in wild type and *ssa1D ssa2D* strains (Fig 9A, S2 Table, *ssa1D ssa2D*/WT Ste7-Myc 1.02 +/- 0.1 S.E., Ste7-Myc-P 1.0 +/- 0 S.E.). Collectively, these findings suggest that Ssa1 and Ssa2 negatively

regulate the abundance of Fus3 and potentially Kss1. By contrast, Ssa1 and Ssa2 did not appear to regulate Ste7-Myc in an obvious way. We note that minor changes in abundance might not be obvious by immunoblot analysis.

## Loss of Ssa1 and Ssa2 reduces basal and α factor-induced expression of a *FUS1::UbiY-lacZ* reporter gene and causes high standard error

We determined the ability of α factor to induce mating pathway responses in the *ssa1D ssa2D* double mutant and several other *ssa* mutant strains. Since the EY3141 *ssa1D ssa2D* and wild type EY3136 WT strains derived from MW63 are kss1- (Materials and Methods and Fig 8), all, or most, of the MAPK signaling responses in these strains are through Fus3. We do not yet know the status of Kss1 in *ssa1D*, *ssa2D*, or *ssa2D ssa3D ssa4D* strains but they are likely to be kss1- since MW63 is a diploid made from isogenic parents. A lack of viability prevented analysis of *ssa2D ssa3D ssa4D* and *ssa1D ssa2D ssa3D ssa4D* strains. We monitored expression with a *FUS1:UbiYlacZ* gene that has a short half-life from a UbiY moiety and is a well-established readout for a factor-dependent activation of the mating MAPKs and transcription [88,89]. The *FUS1* UAS fragment is fused to the *CYC1* TATA element. This construct has higher levels of basal expression than the native *FUS1* promoter, permitting robust dection of changes in basal expression.

FUS1::ubiY-lacZ was measured in *ssa1D*, *ssa2D*, *ssa1D ssa2D* and *ssa2D ssa3D ssa4D* strains grown at room temperature without and with a factor induction (Fig 10A and 10B). The wild type basal FUS1::ubiY-lacZ was ~20% the levels induced by a factor, a higher basal level than what is detected in W303a and S288c strains (Figs 10A, 10B and S6A). Ssa1 and Ssa2 provided additive contributions to the basal levels of FUS1:UbiYlacZ leading to 3.6 +/-0.12 S.E. % WT levels in the *ssa1D* mutant, 5 +/- 0.12 S.E.% in the *ssa2D* mutant and 2 +/- 0.17 S.E. % WT levels in the *ssa1D ssa2D* double mutant (Fig 10A). In the presence of α factor for 2 hours, FUS1::ubiYlacZ β-galactosidase activity was reduced. At 5 mM a factor, the *ssa1D* single mutant had 47 +/- 1.4 S.E.% of wild type, *ssa2D* had 52 +/- 0.87% and *ssa1D ssa2D* double mutant had 19 +/- 6.9% of wild type (Fig 10A and 10B). The reduction in FUS1::UbiY-lacZ in the *ssa1D ssa2D* double mutant was statistically significant (Fig 10B). A *ssa1D ssa3D ssa4D* triple mutant solely dependent on Ssa2 had 2.7 +/- 0.16% of wild type basal LacZ activity and, after 5 mM α factor stimulation, had 46.1 +/- 0.58 S.E.% wild type lacZ activity similar to the *ssa1D* and *ssa2D* single mutants (Fig 10A and 10B), indicating Ssa3 and Ssa4 contribute negligibly to *FUS1::UbiY-lacZ* expression. Therefore, Ssa1 and Ssa2 are equivalently required for full expression of FUS1:UbiYlacZ, with little contribution from Ssa3 and Ssa4.

The HSP70 and HSP90 chaperones are known to buffer cells from phenotypic variability. The induction of *FUS1::ubiYacZ* by a factor (Fig 10A and 10B) revealed high standard errors as a characteristic of the *ssa1D ssa2D* double mutant (S6 Fig). Large fluctuations for basal expression are known to occur from stochastic loss of inhibition by negative regulators such as the RGS protein Sst2. Ssa1 and Ssa2 are predicted to regulate Sst2 in addition to the Dig1 and Dig2 repressors [106] which could potentially impact basal signaling and transcription.

## Ssa1 and Ssa2 are required for G1 arrest and α factor-induced shmoo formation but not mating

Increasing levels of mating MAPK cascade activation are needed for transcriptional activation, G1 arrest and shmoo formation [85]. We assayed the contribution of Ssa1 and Ssa2 to G1 arrest and shmoo formation in EY3136 and EY3141 (S1 Table). The *ssa1D ssa2D* strain was less efficient at being inhibited by α factor and formed smaller more turbid halos of growth inhibition compared to wild type (Fig 10G and 10H). Taken together, the reduced FUS1::

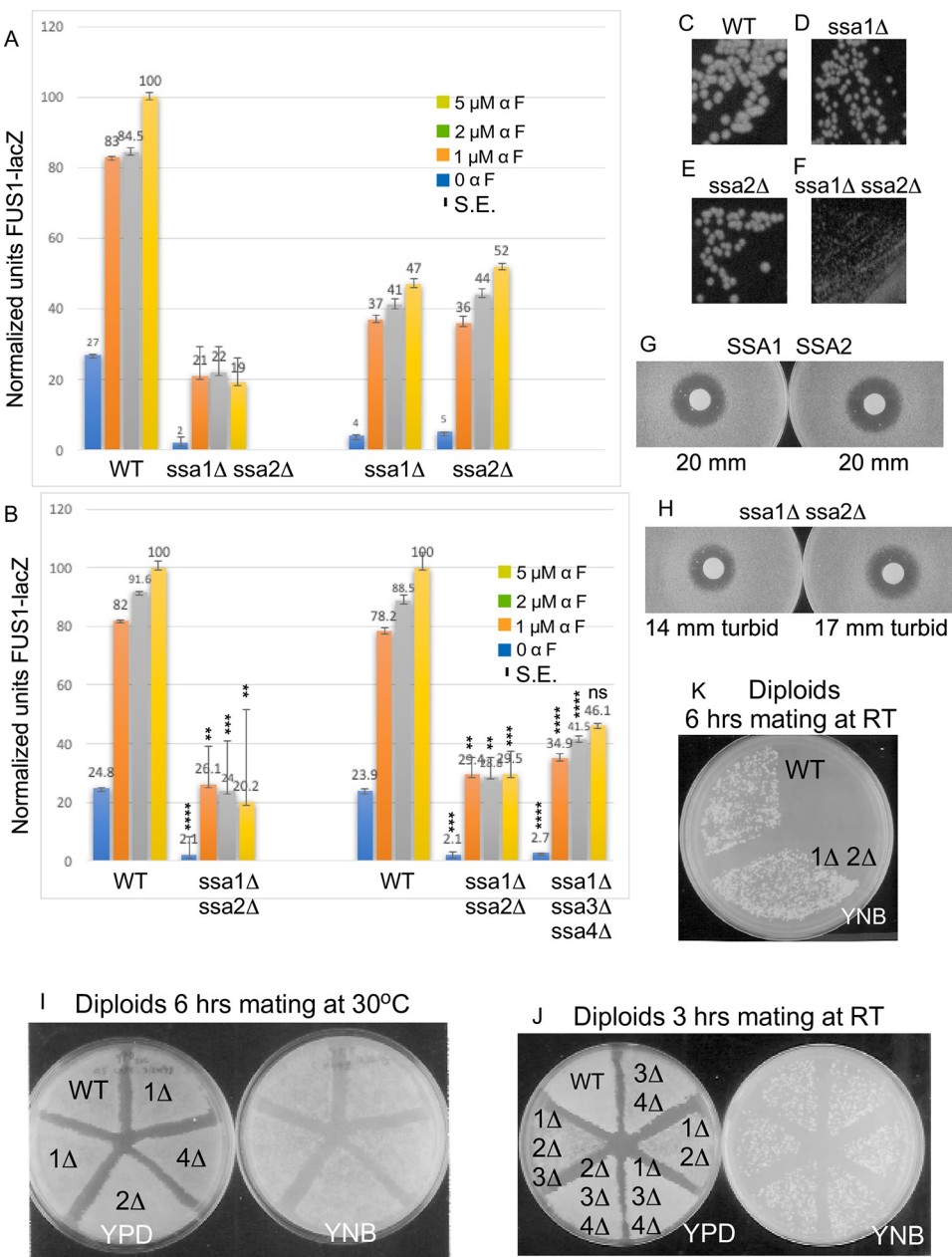

**Fig 10. *ssa1D ssa2D* double mutants have reduced FUS1::ubiYlacZ activity and G1 arrest but can still mate.** A-B. Quantification of β-galactosidase activity in wild-type and *ssa1D*, *ssa2D*, *ssa1D ssa2D* and *ssa1D ssa3D ssa4D* strains expressing FUS1::ubiYlacZ. EY3136, EY3137, EY3138, EY3141 and EY3148 (non-reverting *ura3⁻* derivatives) harboring the FUS1::ubiYlacZ plasmid pDL1460 were grown at room temperature and exposed to the indicated amount of α factor for 2 hours. Extracts were prepared as described in Materials and Methods. Mean +/- S.E. is shown. p-value asterisk key: ns (not significant) >0.05, * = <0.05, ** <0.01, ***,0.001, **** <0.0001. C-F. Streakouts of strains on YPD plates grown at room temperature. Strains: WT EY3136 (C), *ssa1D* EY3137 (D), *ssa2D* EY3138 (E), *ssa1D ssa2D* EY3141 (F). G-H. Duplicate halo assays for WT (G) and *ssa1D ssa2D* (H) strains grown at RT. K. Patch mating of *ssa1D*, *ssa2D*, *ssa3D* and *ssa4D* single, double and triple deletion mutants. Patches of wild type and mutant strains were grown on YEP-2% dextrose agar plates and then mated against lawns of *MATa lys9* wild-type cells for the indicated time and then transferred to YNB 2% dextrose petri plates to select for prototrophs. F. RT, 6 hour mating for *WT*, *ssa2D ssa2D*. I. 30˚C, 6 hour mating for WT, *ssa1D*, *ssa2D*, *ssa4D*. J. RT, 3 hour mating for WT, *ssa1D ssa2D*, *ssa3D ssa4D*, *ssa1D ssa3D ssa4D*, *ssa2D ssa3D ssa4D*, *ssa1D ssa2D ssa3D*.

ubiYlacZ activity and growth arrest reveal that mating pathway activation is impaired in the *ssa1D ssa2D* double mutant.

The morphologies of vegetatively dividing *ssa1D ssa2D* cells in vegetative growth were similar to wild-type but the *ssa1D ssa2D* cells were slightly larger (~10% wider) and more were unbudded (i.e. 41.4% unbudded EY3136 *SSA1 SSA2* cells versus 56.2% unbudded EY3141 *ssa1D ssa2D* cells (Fig 11A and 11C, S4 Table). Interestingly, the *ssa1D ssa2D* cells had a weaker morphological response to α factor than wild type *SSA1 SSA2* cells. Fewer *ssa1D ssa2D* cells formed shmoos compared to wild type, and fewer became crumpled from lysing after high a factor signaling (Fig 11B and 11D–11F, Table 5, S4 Table). Most noteworthy was that many more *ssa1D ssa2D* cells were round enlarged rather than the classic pear shape with tapered projection. The cells that had some shmoo morphology had shorter less emerged projections that were broader with 29.2% round unbudded *ssa1D ssa2D* cells versus 6.8% *SSA1 SSA2* cells (e.g. Fig 11A–11F, Tables 5 and S4). In Fig 11, the cell width for *SSA1 SSA2* in arbitrary units is 1 (size range 0.89–1.2, N = 10) versus 1.43 for *ssa1D ssa2D* (size range 0.97–1.9, N = 17). Therefore, Ssa1 and Ssa2 are needed for efficient α factor-induced polarized morphogenesis in addition to vegetative growth. Moreover, in the absence of Ssa1 and Ssa2, more cells expand growth in all directions rather than towards a single polarization site. The reduced shmoo formation of the *ssa1D ssa2D* double mutant is consistent with reduced a factor signaling through Fus3.

We also assessed G1 arrest and shmoo formation in cells solely dependent on Ssa1 for survival. Ssa1 was overexpressed in a *ssa1D ssa2D ssa4D* strain with a *GAL1prom-SSA1-CEN* (EYL342/MW331 from the Craig laboratory) by growth in medium containing 2% galactose, then Ssa1 was depleted by shifting cells to medium containing 2% dextrose to repress the *GAL1* promoter, and then cells were exposed to α factor or mock buffer and monitored over time for cell morphology (Table 6) and viability (S2B and S2C Fig). Viability was unchanged during the first 6 hours of glucose repression of *GAL1prom-SSA1* and then it started to overtly decline by 8 hours. The cells were mainly unbudded at onset of experiment (60.9 +/- .25 S,E, % unbudded) and after 6 hours of Ssa1 depletion were 81% unbudded without a factor and 77.2 +/- 1.2 S.E. % unbudded with a factor. Shmoo formation was greatly reduced. After 2 hours of repression of the *GAL1-SSA1* gene, 39.4 +/- 4 S.E.% of the cells formed shmoos which is less than expected compared to wild type cells. Strikingly, after 6 hours of depleting Ssa1, only 3.2 +/- 0.25% formed shmoos, a 10-fold reduction (Table 6, p-value .023095). Thus, Ssa1 is needed to transit from G1 to S phase, to bud and to form shmoos during mating.

Surprisingly, we found that *ssa1-ssa4* single, double and triple mutants could mate, despite the defects in signaling through the mating MAPK cascade. The mating of wild-type, *ssa1*, *ssa2*, *ssa3*, *ssa4* single, double and triple mutants were compared in a standard plate assay in which patches of cells are mated to a lawn of wild-type cells. There was no obvious decrease in mating ability for any of the mutants compared to wild-type at room temperature or 30˚C in *BAR1* or *bar1* backgrounds (Fig 10I–10K, Table 7). Given the poor growth, reduced Ste5 abundance, MAPK activation, FUS1::ubiYlacZ levels and morphogenesis in the *ssa1D ssa2D* double mutant,it is surprising that there was no obvious mating defect. These results lead to speculation that the *ssa1 and ssa2* mutations induce additional changes that compensate for the signaling defects. For example, Ssb1 and Ssb2 are predicted to interact with Ste5 [106] and may generate a pool of Ste5 that is folded and functional in the *ssa1D ssa2D* strain along with other proteins. Extremely low levels of GST-Ste5 undetectable by immunoblot analysis are sufficient to drive signal transduction, whereas over production of GST-Ste5 can be slightly inhibitory [9]. Genetic analysis of interspecies chimeras suggests that weaker binding of Ste5 to kinases may enhance mating [107]. In addition, counteractive effects of increased Fus3 abundance and

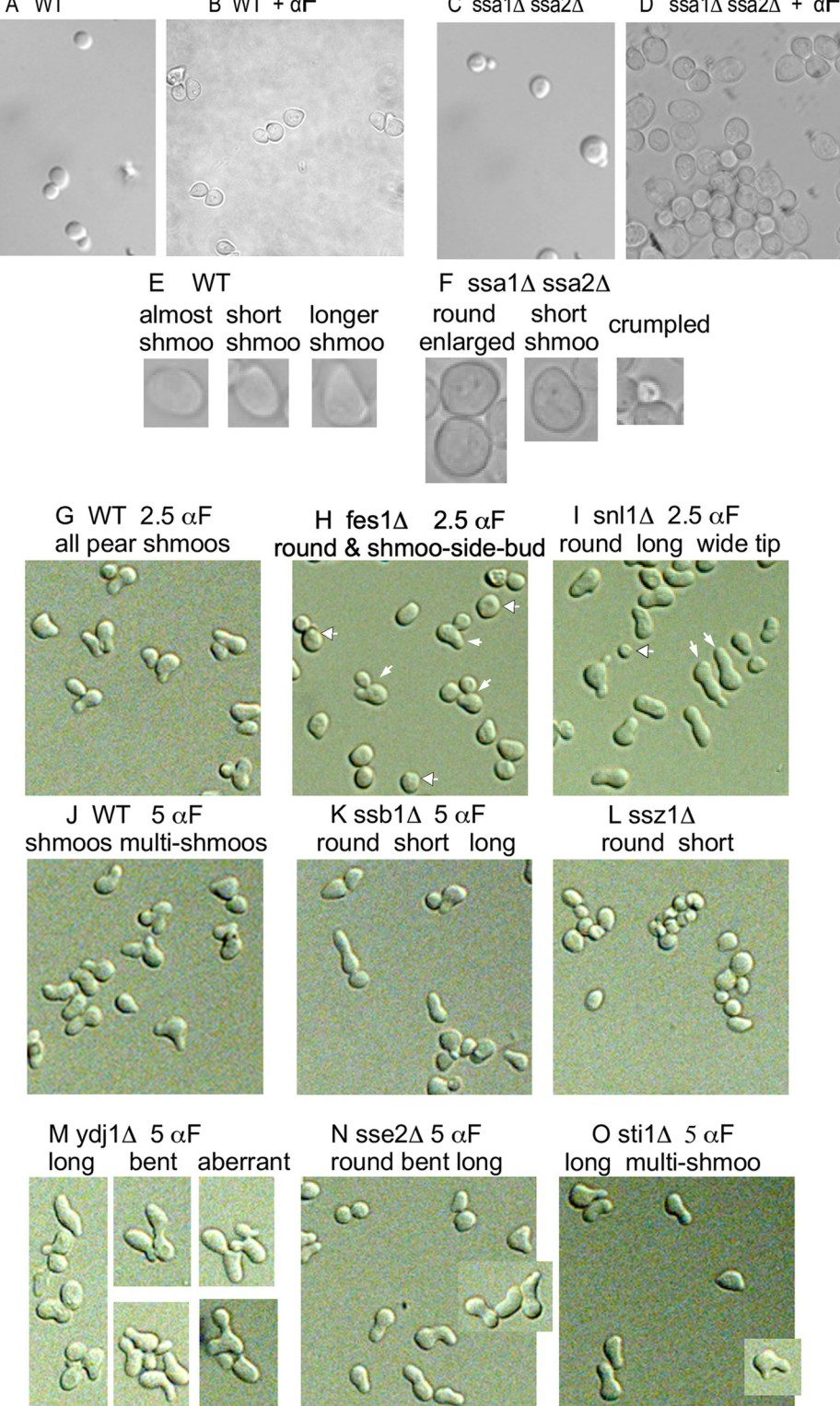

**Fig 11. *ssa1D ssa2D* double mutants and several HSP70-HSP90 network mutants are defective in shmoo formation.** A-D. Images of wild-type EY3136 and EY3141 *ssa1D ssa2D* cells during logarithmic growth in YEPD before and after treatment with 5 mM α factor for 90 minutes at room temperature. The field of α factor treated *ssa1D ssa2D* cells has only 6 shmoos out of a total of 80 cells. E-F. Examples of wild-type and *ssa1D ssa2D* shmoo morphologies. G-O. S288c WT, HSP70 and HSP70-HSP90 network mutants. Images of fields of cells of BY4741 S288c

WT (KSS1+) and chaperone mutants derived from BY4741 grown logarithmically in YEPD at room temperature and treated with a factor for 2 hours at room temperature. G. WT 2.5 mM aF; H. *fes1D* 2.5 mM aF; I. *snl1D* 2.5 mM aF; J. WT 5 mM aF; K. *ssb1D* 5 mM aF; L. *ssz1D* 5 mM aF; M. *ydj1D* 5 mM aF; *sse2D* 5 mM aF; *sti1D* 5 mM aF.

any other signaling protein that is downregulated by Ste5 may relieve the reduction in signaling that results from reduced Ste5.

## Interaction data suggest Ssa1 is the chaperone most dedicated to the mating pathway, followed by Ssa2 and the HSP70-HSP90 network

To investigate compensatory protein quality control and relative importance of different HSP70 chaperones to the mating and invasive growth pathways, we made an inventory of published predicted physical interactions between Hsp70 and Hsp90 protein networks and 212 mating pathway and 121 invasive growth pathway proteins (S3 Table). The list includes eleven Hsp70 proteins Ssa1-Ssa4, Ssb1, Ssb2, Ssz1 (cytosolic), Ecm10, Ssc1 (mitochondrial) and Kar2, Lhs1 (ER), two Hsp40/DnaJ family proteins (Ydj1, Sis1/BAG-1), two nucleotide exchange factors (Fes1/HspBP1 homolog, Snl1/Bag-1 homolog), 2 Hsp110 proteins (Sse1 and Sse2), two Hsp70-associated disaggregation chaperones (Hsp104, Hsp42), two Hsp90 proteins (Hsc82, Hsp82), and three Hsp90 co-chaperones in the HSP70-HSP90 network, Sti1 (which also interacts with Ssa1), Cdc37 and Sba1. Most data came from unconfirmed high throughput experiments. All cells in the published experiments had been grown at either RT or 30°C in logarithmic phase in either YEPD, SCD or SILAC medium and were not subjected to heat shock (S4 Table).

The co-translational chaperones Ssb1 and Ssb2 had the most predicted total interactions (64% and 46% respectively) followed by the posttranslational chaperones Ssa1 (53%) and Ssa2 (39%). Hsp82 is predicted to interact with only 12% of the mating and IG proteins and Hsc82

**Table 5. Morphology of *SSA1 SSA2* and *ssa1D ssa2D* strains.**

| Strain, time point[1] | Round | % Unbudded[2] | | | %Budded[3] | %Cr | %AS+S |
| | | AS | S | MS | | | |
| --- | --- | --- | --- | --- | --- | --- | --- |
| 1. *STE5-GFP CEN* | | | | | | | |
| WT + aF 2 hr | 7.3 (3) | 29.2 (16.9) | 48.1 (16.1) | 0 (0) | 0.62 (0.019) | 7.9 (2.5) | 77.3 |
| *ssa1D ssa2D* +aF 2 hr | 35.4 (0.46) | 49.8 (8.6) | 32.5 (14.7) | 0 (0) | 0 (0) | 3.2 (0) | 82.3 |
| p-value[6] | 0.0113 | | | | | | |
| 2. *STE5-MYC9 2m* | | | | | | | |
| WT + aF 2 hr | | | 72.3 (15.4) | | | | 86.5 (10.1) |
| *ssa1D ssa2D* + aF 2 hr | | | 47.3 (11.4) | | | | 78.9 (12.7) |
| p-value | | | 0.2815 | | | | 0.6576 |

[1]Strains in experiments and 2 were grown overnight in SC selective medium with 2% dextrose pH 5.0 at room temperature to A600 ~0.2–0.4, pelleted, adjusted for equal density in fresh medium at A600 of 0.6 and treated with 2.5 mM a factor for 2 hours with shaking then scored live for GFP containing cells and after fixation with 1/10 volume 37% formaldehyde for Ste5-Myc9 cells. In some cases, tallies were done on photographs of samples.

[2,3] Cells were scored for whether they were unbudded (Ub), budded (B), shmoo (S), enlarged irregular almost shmoo shaped (AS), multi-shmoo (MS), and crumpled dying (Cr). Greater than 250 cells were scored for each characteristic in experiments 1 and 2. The mean (M) of the percentage of total cells is shown for each category with the standard error (S.E.). For the *STE5-GFP CEN* experiments, N = 2 or N = 3.

For the *STE5-MYC9 2m* experiments, N = 3 or N = 4.

[4]M(SE) is the mean of values. S.E. is the standard error. M (S.E.) was calculated for the total number of almost shmoo and shmoo cells.

[5]p-values were calculated using a Student's t-test of two independent means, two-tailed. They were calculated for the total number of shmoos and the total number of almost shmoo and shmoo cells at each time point compared to the 2 hour time point of a factor treatment.

**Table 6. Morphology of *ssa1D ssa2D ssa4D + GAL1prom-SSA1* cells after repression of *SSA1* expression.**

| Strain, time point | Round | %Unbudded cells[2] | | | | % Budded cells | | |
| | | AS | S | MS | AS+S | Round | AS | S |
| --- | --- | --- | --- | --- | --- | --- | --- | --- |
| EYL342 *ssa1D ssa2D ssa4D + GAL1prom-SSA1*[1] | | | | | | | | |
| galactose to dextrose shift | | | | | | | | |
| 1 t = 0 hr "true" | 61.1 | 0 | 0 | 0 | | 38.9 | 0 | 0 |
| 2 t = 0 hr "true" | 60.6 | 0 | 0 | 0 | | 39.4 | 0 | 0 |
| Mean (S.E.)[2] | 60.85 (.25) | | | | 0 (0) | | | |
| 1 t = 0 hr | 50.9 | 0.6 | 0 | 0 | | 43.6 | 0 | 0 |
| 1 t = 0 hr + af 2 hr | 35.4 | 16 | 39.4 | 0 | | 4.6 | 2.3 | 1.7 |
| 2 t = 0 hr + af 2 hr | 43.4 | 17.7 | 29.7 | 0 | | 5.7 | 0.6 | 2.3 |
| Mean (S.E.)[3] | 39.4 (4) | | 34.55 (4.85) | | 55.2 (5.52) | | | |
| p-value[4] | 0.033183 | | | | | | | |
| 1 t = 3 hr | 57.7 | 0 | 0 | 0 | | 42.3 | 0 | 0 |
| 1 t = 3 hr + af 2 hr | 27.4 | 26.3 | 42.9 | 0 | | 2.3 | 0.6 | 0.6 |
| 2 t = 3 hr + af 2 hr | 31.4 | 26.3 | 38.3 | 0 | | 3.4 | 0.6 | 0 |
| | 29.4 (2) | | 40.6 (2.3) | | 66.9 (2.3) | | | |
| p-value | 0.004082 | | 0.376744 | | 0.286837 | | | |
| 1 t = 4.5 hr | 78.3 | | | | | 21.7 | | |
| 1 t = 4.5 hr + af 2 hr | 48.3 | 19.9 | 27.3 | | | 3.4 | 0.6 | |
| 2 t = 4.5 hr + af 2 hr | 44 | 29.7 | 21.1 | | | 4.6 | 0.6 | |
| | 46.15 (2.15) | | 24.2 (3.1) | | 48.9 (1.90) | | | |
| p-value | 0.021 | | 213984 | | 0.68701 | | | |
| 1 t = 6 hr | 81.1 | | | | | | | 18.9 |
| 1 t = 6 hr + af 2 hr | 78.3 | 8.6 | 2.9 | | | | | 9.7 |
| 2 t = 6 hr + af 2 hr | 76 | 6.9 | 3.4 | | | | | 13.7 |
| | 77.15 (1.15) | | 3.15 (.25) | | 10.9 (6.0) | | | |
| p-value | 0.005172 | | 0.023095 | | 0.029798 | | | |
| 1 t = 8.12 hr | 83 | | | | | 17 | | |

[1]EYL342 (*MATa ssa1::HIS3 ssa2::LEU2 ssa4::LYS2 +pGAL1prom-SSA1 CEN*), was grown in YEP-2% galactose overnight to logarithmic phase, washed once in YEP-2% dextrose (pH 5.0), resuspended in the same medium and a 1 ml aliquot was taken for a time zero time point (t = 0 "true"). At the indicated intervals, 1 ml aliquots of cells were removed for growth rate determination and for incubation without or with either 10 ml DMSO or 10 ml 500 mM a factor in DMSO (5 mM final concentration) with shaking at 30˚C. Cells were fixed with addition of 1/10 volume of 37% formaldehyde, then chilled on ice and sonicated before examining under the microscope. Cell viability was tallied at each time point by plating cells onto YPD plates.

[2] Cells were scored for whether they were unbudded (Ub),budded (B), shmoo (S), enlarged irrecgular almost shmoo shaped (AS), multi-shmoo (MS), and crumpled dying (Cr). Greater than 250 cells were scored for each characteristic in experiments 1 and 2, and 175 cells were scored for each characteristic in experiment 3. The percentage of total cells is shown for each category with the standard error (S.E.) in parentheses with the number of experiments (N).

[3]M (S.E.) is the mean of values. SE is the standard error, which is the standard deviation divided by the square root of the number of samples. M (S.E.) was calculated for the total number of almost shmoo and shmoo cells.

[4]p-values were calculated using a Student's t-test of two independent means, two-tailed. They were calculated for the total number of almost shmoo and shmoo cells at each time point compared to the 2 hour time point of a factor treatment. Cell viability was tallied at each time point by plating cells onto YPD plates.

with 25%, followed by Ydj1 (22%), a J domain co-chaperone for both Hsp70 and Hsp90 (S4 Table, Fig 12A). The Hsp110 chaperone Sse1 potentially interacts with 37% of pathway proteins compared to only 5% for Sse2. There were fewer predicted interactions for Ssa3 (9%), Ssa4 (14%), Sse1 (5%), Ssc1 (11%), Ssq1 (1.4%), Ssz1 (15%), Ecm10 (2.3%) and Kar2 (6%), Sis1/BAG1 (15%) (Fig 12A). Ecm10, Kar2, Lhs1, Fes1, Snl1, and Sba1 had no predicted interactions with the 41 core signal transduction and transcription proteins of mating and invasive growth (Fig 12A, S4 Table).

**Table 7. Summary of growth and mating of *ssa1D*, *ssa2D*, *ssa3D*, and *ssa4D* single, double and triple mutants.**

| Strain[1] | Growth | | Mating | |
|---|---|---|---|---|
| | RT | 30˚C | RT | 30˚C |
| EY3136 WT | ++ | ++++ | +++ | +++ |
| EY3137 *ssa1D* | + | +++ | | +++ |
| EY3138 *ssa2D* | + | +++ | | +++ |
| EY3139 *ssa3D* | + | +++ | | +++ |
| EY3140 *ssa4D* | + | +++ | | +++ |
| EY3141 *ssa1D ssa2D* | +/- | +/- | +++ | +++ |
| EY3143 *ssa1D ssa4D* | + | | | |
| EY3144 *ssa2D ssa3D* | + | | | |
| EY3145 *ssa2D ssa4D* | + | | | |
| EY3146 *ssa3D ssa4D* | + | | +++ | |
| EY3147 *ssa1D ssa2D ssa3D* | +/— | | +++ | |
| EY3148 *ssa1D ssa3 D ssa4D* | + to ++ | | +++ | |
| EY3149 *ssa2D ssa3D ssa4D* | + | | +++ | |

[1]Streakouts and patch mating of WT and *ssa1D-ssa4D* single, double and triple mutant strains. WT and *ssa1D-ssa4D* single mutant and *ssa1D ssa2D* double mutant strains were tested for mating at 30˚C after mating for 6 hours on YPD followed by recovery of diploid prototrophs on YNB 2% dextrose at 30˚C. The. WT, *ssa1D-ssa4D* double mutant and triple mutant strains were quite sick and were mated at room temperature for 3 hours on YPD. In these experiments extra cells of the two slowest growing strains, *ssa1D ssa2D* and *ssa1D ssa2D ssa3D*, so that these patches would be of equal density to wild type patches after overnight growth at either 30˚C or room temperature. WT and *ssa1D ssa2D* strains were tested in mating assays four times and the *ssa1D-ssa4D* single mutants and all other double and triple mutants were tested twice. Strains tested in these assays were: EY3136 *MATa* wild type (*kss1-*), EY3137 *MATa ssa1::HIS3*,EY3138 *MATa ssa2::LEU2*, EY3139 *MATa ssa3::TRP1*, EY3140 *MATa ssa4::URA3*, EY3141 *MATa ssa1::HIS3 ssa2::LEU2* (*kss1-*), EY3143 *MATa ssa1::HIS3 ssa4::URA3*, EY3144 *MATa ssa2::LEU2 ssa3::TRP1*, EY3145 *MATa ssa2::LEU2 ssa4::URA3*, EY3146 *MATa ssa3::TRP1 ssa4::URA3*, EY3147 *MATa ssa1::HIS3, ssa2::LEU2 ssa3:: TRP1*, EY3148 *MATa ssa1::HIS3 ssa3::TRP1 ssa4::URA3*, EY3149 *MATa ssa2::LEU2 ssa3::TRP1 ssa4::URA3*. Plates were photographed every day for 4 days.

We compared the percent distribution of Ssa1, Ssa2, Ssa3, Ssa4, Sse1 and Sse2 interactions with mating pathway and invasive growth pathway proteins and the extent of overlap redundancy between chaperones. Ssa1 had the greatest number of predicted mating pathway and invasive growth pathway clients (i.e. 79% and 77% respectively) followed by Ssa2 (55%, 52% respectively) (Fig 12B–12E). In contrast, Ssa3 was 13% for mating and 18% for invasive growth and Ssa4 was 14% for mating and 24% for invasive growth (Fig 12B–12E, S4 Table). Interestingly, 100% of mating pathway proteins predicted to be regulated by Ssa2 are also predicted to be regulated by Ssa1, and likewise for Ssa3 (100%) and Ssa4 (100%) (Fig 12B–12E), with similar overlap for the invasive growth pathway (S8B Fig, 94% for Ssa2, 100% for Ssa3, 100% for Ssa4). The bias towards Ssa1 is pronounced among 29 core mating pathway signaling components (95%) and 25 core invasive growth pathway signaling components (94%) (Fig 12B–12E).

The literature suggests a balance between Hsp70 promotion of degradation of misfolded proteins and Hsp90 promotion of folding to full maturation [60,64,104,108]. Current models suggest that Hsp90 uses both Ssa-dependent and Ssa-independent routes to fold proteins [40,41,44,103–105,109]. Hsp70-Hsp90 complexes are thought to act downstream of the Ssa chaperones to mature proteins [105,109]. The Hsp70-Hsp90 complexes include Ssa1, Ssa2, Hsp82 and Hsc82, and one or more of Ydj1, Sse1, Sse2, Sti1/HOP, Sba1, Cpr and Cdc37 [61,62,104,105,109]. Hsp90 has been classified as mainly regulating protein kinases and has

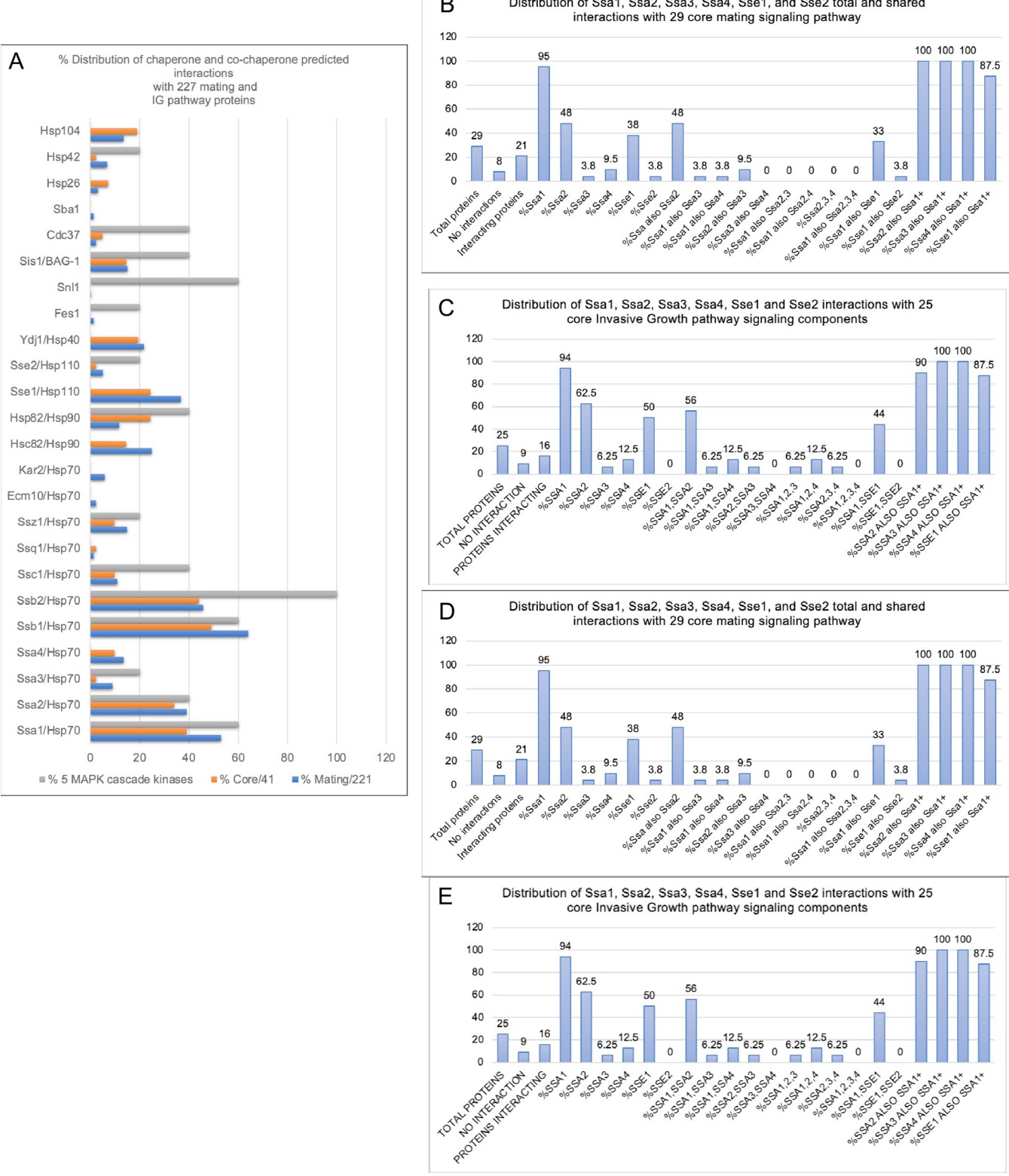

**Fig 12. Analysis of published putative interactions between Hsp70, Hsp90 and other HSP network proteins and 212 mating pathway proteins and 121 invasive growth pathway proteins.** A. Percent distribution of predicted interactions between Hsp70 chaperones Ssa1, Ssa2, Ssa3, Ssa4, Ssb1, Ssb2, Ssc1, Sse1, Sse2, Ssq1, Ssz1, Ecm10, Kar2, Hsp90 chaperones Hsp82, Hsc82, small heat shock proteins Hsp26, Hsp42, disaggregase Hsp104 and co-chaperones Ydj1, Fes1, Snl1, Sis1/BAG-1, Cdc37, Sba1 and 221 mating and invasive growth pathway proteins, 41 core signal transduction and transcription factor proteins for mating

and invasive growth, and the 5 kinases that comprise the MAPK cascade (Ste20, Ste11, Ste7, Fus3, Kss1). See S3 Table for the list of mating and invasive growth proteins, S4 Table for the list of proteins potentially interacting with each chaperone and co-chaperone and database references. The majority of the reported interactions are from unconfirmed high-throughput experiments done at either room temperature or 30˚C. B-E. Comparisons of unique and shared predicted interactions among Ssa1, Ssa2, Ssa3, Ssa4, Sse1 and Sse2 for mating pathway and invasive growth pathway proteins. A-B. Percent distribution and overlap among 212 mating and 121 invasive growth pathway proteins. C-D. Percent distribution and overlap among core signaling and transcription factor components of mating (29 proteins) and invasive growth (25 proteins). Of 212 mating pathway proteins investigated, only 150 have predicted interactions with the HSP network and percentages are based on 150 proteins. Of the 121 invasive growth pathway proteins, 38 had no interactors and percentages are based on 83 proteins.

some function in the mating pathway and is required for full stability of Ste11ΔN, a constitutively active mutant kinase that does not bind Ste5 [103]. Ste11DN has reduced ability to induce *PRE-lacZ* in a *Δhsp82::LEU2 Δhsc82::LEU2 hsp82-G313N* triple mutant strain, however, shmoo formation and mating have not been documented for this mutant [103,104].

We found considerable redundancy for the kinases of the MAPK cascade, with Ste20 predicted to be regulated by Ssb1, Ssb2, Ssa1, Hsp82, Hsc82; Ste11 by Ssb1, Ssb2, Ssa1, Ssa2, Ssa3, Ssc1, Hsp82, Hsc82; Ste7 by Ssb1, Ssb2, Ssc1; Fus3 by Ssb1, Ssb2, Ssa1, Ssa2, Hsc82; and Kss1 by Ssb2, Ssz1, Hsp82 (S4 Table, Fig 12A). That Ste7 may be regulated by Hsp82 but not by Ssa1 or Ssa2 (S4 Table, 102) is consistent with our finding no obvious differences in abundance of Ste7 in the *ssa1D ssa2D* strain, whereas the changes in abundance for Ste5 and Fus3 correlate with co-immunoprecipitation and/or predicted interaction data.

The inventory of Hsp70 and Hsp90 interactions (S4 Table, Fig 12) indicates that Ssa1 and Ssa2 are the major regulators for mating and invasive growth pathways with some overlap from Hsp82 and Hsc82 that could be either independent of or dependent upon Hsp70-Hsp90 complexes. The Hsp110 protein Sse1 which cooperates with Hsp70 and Hsp90 [46] is predicted to interact with 50% of mating and 57% of invasive growth proteins (S4 Table, Fig 12), suggesting these predicted clients could be involved in tripartite complexes with these chaperones.

## Hsp70 and Hsp90 regulators Fes1, Ydj1, Snl1, Sse2, Sti1 and Ssz1 are also required for shmoo morphogenesis

To further explore compensatory protein quality control that could occur in the absence of Ssa1 and Ssa2, we determined whether mutations in other Hsp70 and Hsp70-Hsp90 network proteins affect the ability of cells to follow through on the multiple Gbg (Ste4/Ste18) and Gg (Gpa1) -mediated responses that induce cell cycle arrest in G1 phase and formation of shmoos. Fes1 (Factor Exchange For Ssa1) and Snl1 (homolog of human BAG-1) are nucleotide exchange factors for Ssa1 that have few predicted interactions with mating pathway proteins (i.e. Fbp1, Lsg1, Mpk1/Slt2 for Fes1 and Mpt5 for Snl1, S4 Table). Fes1 is required for the release of Ssa1-bound misfolded protein to the proteasome and is not thought to function with Hsp90. The *fes1D* null mutant [39,40] was defective in shmoo formation with many round cells and the few shmoos that formed often had a novel additional morphology of a small bud on the side (Fig 11G and 11H; Tables 8, S4, S5 and S6, S7 and S8 Figs, 38.9% +/- 9.7 S.E. for *fes1D* versus 87.4% +/- 0.4 S.E. for wt). In terms of forming a "long shmoo", the *fes1D* mutant formed 1.3% +/- 0.7 S.E. compared to 14.1% +/- 0.6 S.E. by wild type (Tables 8, S4, S5 and S6). Thus, blocking processing of Ssa1-client complexes in the *fes1D* mutant resulted in as strong a mutant phenotype as a *ssa1D ssa2D* double mutant, with an additional novel phenotype of either concommittant bud formation at a default polarization site or formation of a shmoo on a dividing cell. Given that the major target of Fes1 function is Ssa1 and Ssa2, these findings provide further support for a role for Ssa1 and Ssa2 in regulating the mating pathway.

By contrast, loss of the Snl1/BAG-1 nucleotide exchange factor that is a co-chaperone for Ssb1 and Ssa1 Hsp70s had little effect on shmoo formation. The *snl1D* mutant formed 78% the

**Table 8. Summary of defects in shmoo formation among HSP70 and HSP90 network mutants.**

| Strain[1] | aF mM | %Shmoo | # shmoo[2] | #total-shmoo[2] | p-value[3] |
|---|---|---|---|---|---|
| WT EY3136 | 5 | 54 (100)[4] | 169 | 146 | |
| *ssa1D ssa2D* EY3141 | 5 | 15 (28) | 60 | 342 | <0.00001 |
| *1D2D4D GAL1p-SSA1* | 5 | | | | |
| t = 0 hr repress *SSA1* | | 34.6 (100) | 121 | 229 | |
| t = 6 hr repress *SSA1* | | 3.1 (9) | 11 | 339 | <0.00001 |
| WT EYL1740 | 2.5 | 36 (100)$_{2.5}$ | 174 | 309 | |
| WT EYL1740 | 5 | 54 (100)$_5$ | 512 | 432 | |
| *snl1D* | 2.5 | 24 (67) | 238 | 542 | 0.0481 |
| *fes1D* | 2.5 | 27 (75) | 273 | 726 | 0.1539 |
| *fes1D* | 2.5 | 34 (94) | 386 | 761 | 0.3613 |
| *fes1D* | 2.5 | 31 (86) | 659 | 1487 | 0.0264 |
| *fes1D* total tally | 2.5 | 28.7 (80) | 1318 | 2974 | 0.0197 |
| *sti1D* | 5 | 82 (152) | 378 | 84 | <0.00001 |
| *ydj1D* | 2.5 | 24 (67) | 133 | 416 | 0 |
| *ydj1D* | 5 | 23 (45) | 110 | 172 | <0.00001 |
| *ssb1D* | 5 | 46 (85) | 241 | 245 | 0.1049 |
| *ssb1D*[5] | 5 | 53 (98) | 336 | 633 | <0.00001 |
| *sse1D* | 5 | 42 (78) | 262 | 368 | <0.00001 |
| *sse2D* | 5 | 0 (0) | 0 | 202 | <0.00001 |
| *ssz1D* | 5 | 8 (15) | 54 | 614 | <0.00001 |

[1]Strains and data are taken from Table 5, S1, S7 and S8.

[2]#Shmoo is the number of cells that had a shmoo morphology. #Total-shmoo is the total number of cells scored minus the number of shmoo cells.

[3]p-values were calculated using the Fisher's Exact Test at the Social Science Statistics website socistatistics.com, copyright 2023 Jeremy Stangroom.

[4]The numbers in parentheses (#) are % wild type shmoo formation.

[5]The sample was counted a second time with more cells in the tally.

wild type number of shmoos, although some shmoos were wider with irregular peanut shapes (Fig 11L, Tables 8, S4 and S6, S6 and S7 Figs). Thus, Fes1 has a more important role in regulating folding of clients in the mating pathway.

Ssb1 is predicted to interact with 142 mating and invasive growth proteins (Fig 12, S4 Table). Surprisingly, the *ssb1D* mutant underwent nearly normal G1 arrest and shmoo formation and made 88.7% the wild type number of shmoos, although some shmoos were longer than wild-type (Fig 11K; S4 Table, S6 Table, Table 8, S7C Fig). Perhaps paralog Ssb2 compensates for loss of Ssb1. Ssz1 is an Hsp70 that associates with the ribosome via contact with Zuo1, a ribosome-associated J protein, and is predicted to interact with 4 core signaling proteins (Far1, Kss1, Sst2, Tec1) and 29 other mating and invasive growth pathway proteins (S4 Table). Mutation of Ssz1 caused defects in both G1 arrest and shmoo formation (Figs 11L and S7C, S6 Table, Table 8). Therefore, Ssz1 may provide unique functions for mating morphogenesis that are linked to translation.

We also assessed mutations in 4 chaperones with dual functions in the Hsp70 and Hsp90 pathways: *YDJ1*, *STI1*, *SSE1* and *SSE2*. Ydj1 is the J domain/Hsp40 adapter that helps Hsp70/Hsp90 complexes be recognized by Rsp5 NEDD E3 ubiquitin ligase for ubiquitinylation and subsequent degradation of associated client protein by the proteasome [108]. Of all of the HSP proteins we investigated, Ydj1 is the only one predicted to regulate Gpa1 (Gα) which negatively regulates the mating signaling pathway (Table 8). Notably, the *ydj1D* cells were more enlarged and misshapen during vegetative growth with some cells appearing shmoo shaped (Fig 11M,

S8F Fig, S6 Table), a phenotype that is seen with reduced Gpa1 function. The *ydj1D* mutant did not efficiently arrest in G1 phase and became more severely misshapen (both unbudded and budded cells), enlarged and elongated after a factor treatment and formed fewer shmoos that were often enlarged and aberrant, with projections that were longer, wider, curved or bent (Fig 11M, S7B, S8E and S8F Figs, Tables 8, S4 and S6). The retention of the ability of the *ydj1D* mutant to enlarge in presence of a factor suggests the cells are not fully blocked in signal transduction, in agreement with reduced *FIG1* expression and Axl1 biogenesis [91,92,104].

Sse1 and Sse2 are ATPase components for Hsp90 and nuclear exchange factors for Hsp70 [40,110]. Sse1 has 10 predicted interactions with core mating and invasive growth signaling components (i.e. Far1, Kss1, Msg5, Msn5/Ste21, Ras2, Rga1, Rga2, Ste20, Ste21/Msn5, Ste50) and 71 predicted interactions with other mating and invasive growth pathway proteins (Table 8), whereas Sse2 is predicted to interact with 1 core signaling protein (i.e. Far1) and 10 other pathway proteins (Table 8). Surprisingly, the *sse1D* mutant had a nearly normal pheromone response for G1 arrest and shmoo morphogenesis and formed 74.4% wild type number of shmoos (S6 Table, Figs 8 and S5). By contrast, the *sse2D* mutant only formed 16.9% wild type level of shmoos, although it could undergo G1 arrest and cell enlargement (Fig 11N, Tables 6 and S6, S7E Fig). Thus, Sse2 must regulate proteins essential for shmoo formation and perhaps compensates in the *sse1D* mutant, but not *vice versa*. This finding is striking because Sse2 is not thought to be important for the regulation of Ste11, suggesting the defect is linked to Hsp70 rather than Hsp90 [105].

Sti1 is a multifunctional HSP co-chaperone that interacts with yeast Hsp90s and Hsp70s Ssa1 and Ssa2 and regulates transfer of clients from Hsp90 to Hsp70. Sti1 inhibits Hsp90 ATPase activity and activates Ssa1 ATPase activity [110–113]. Currently, Sti1 has no known interactions with mating proteins (S4 Table), although a *sti1D* mutation reduced *FIG2* expression and Ste11DN association with an undefined isoform of Hsp90 [104]. Interestingly, the *sti1D* mutation increased the number of shmoos by 53% as well as the length of shmoo projections compared to wild type (Figs 11O and S7G, S4 and S6 Tables, Table 8), indicating either enhanced shmoo formation or reduced downregulation of shmoo formation.

In summary, Fes1, Ydj1, Sse2, and Ssz1 are needed for proper shmoo morphogenesis with their loss causing distinct changes in morphology, whereas Sti1 negatively regulates shmoo formation, consistent with a biochemical role in inhibiting the ATPase activity of Hsp90 [110–113]. Sti1 is not essential for signal transduction linked to shmoo morphogenesis when signaling is through the native pathway that is dependent on Ste5 and contrasts prior predictions based on a mutant Ste11DN allele that signals independently of Ste5 [104]. The severity of defects was greater for the *ydj1D*, *sse2D* and *sti1D* mutations that impact both Hsp70 and Hsp90 networks. The distinctly different morphologies of these mutants underscores compensatory protein quality control mechanisms that operate together with Ssa1 and Ssa2.

## Discussion

The removal of damaged proteins that arise from misfolding, chemical insults, aggregation and translational truncations disrupts cell function and promotes aging and disease. We have found Ssa1 and Ssa2 to have key roles in regulating Ste5 integrity. Ssa1 co-purifies with Ste5 (Fig 1) and this association can be confirmed using tags on either end of Ste5 and Ssa1 in co-immunoprecipitations (Fig 2). The large amount of Ssa1 associating with Ste5 is likely due to its large size and many potential Ssa1 binding sites based on BIPPRed analysis (S3K Fig). Ssa1 is in high abundance compared to Ste5 and MAPK pathway signaling components based on mean averages of molecules per cell from quantitative mass spectrometry studies (i.e. Ssa1-45137 to 314830, Ste5-814, Ste7-615, Ste11-1023, Ste20-3959, Fus3-5006, Kss1-3653 [107]).

Loss of Ssa1 and Ssa2 is detrimental to Ste5-Myc9 integrity and results in reducing abundance to 0.17 +/- 0.03 S.E. to 0.2 +/- 0.06 S.E wild type levels at room temperature when expressed from a *CEN* plasmid (Fig 3, S2 Table), greater vulnerability to proteolysis at all temperatures examined and during α factor stimulation (Figs 4 and 5, S2 Table) and greater availability of epitope-tags on the N- and C-termini to monoclonal antibodies (Fig 6). By contrast, Ste7, Tcm1 and many proteins that cross-reacted to antibodies showed no obvious change in abundance in the *ssa1D ssa2D* double mutant. Thus, loss of Ssa1 and Ssa2 results in less protection for Ste5 at multiple levels.

Ste5 normally accumulates largely as monomers in the nucleus and as dimers at the cell cortex. Ste5 localization was impaired in the *ssa1D ssa2D* mutant: Ste5-Myc9 and several Ste5 derivatives were unable to accumulate at the cell cortex (Table 1), although Ste5 was able to oligomerize (Fig 6D–6F), suggesting the capacity of Ste5 dimers to bind to anchors at the plasma membrane is impaired. Less Ste5 accumulated in nuclei including Ste5-Myc9, Ste5(1–242)-GFP2, which has the bipartite NLS and lipid binding motif (Table 1), GFP-Ste5 and TAgNLSK128T-Ste5-Myc9 (Table 1). We infer from these findings that Ssa1 and Ssa2 chaperones may increase directly or indirectly the pool of Ste5 competent to enter the nucleus or protect and sequester a nuclear pool from either export or degradation by SCF$^{Cdc4}$ in the nucleus. Since the cytoplasmic pool of Ste5-Myc9 was also reduced in the *ssa1D ssa2D s*train, Ssa1 and possibly Ssa2 also appear to protect Ste5 from degradation in the cytoplasm. Moreover, overexpression of Ssa1 induced a VWA-like domain mutant Ste5L610/614/634/637-Myc9 to localize in numerous punctate foci and patches in the cytoplasm (Fig 7; Table 3) supporting a role for Ssa1 in quality control of Ste5. Further work is needed to understand how the chaperones protect Ste5 from degradation and to define the Ssa1-induced punctate foci. Collectively, multiple lines of evidence support an active role for Ssa1 in maintaining Ste5 abundance, protection from potentially less functional conformers and degradation and proper localization in the cell.

Ssa1/Ssa2 exert refolding and aggregation functions in the cytoplasm which produces either soluble biologically active protein or, conversely, soluble misfolded protein that can be recognized for ubiquitinylation and degradation [39–41,51,63,64,68]. The simplest interpretation is that Ste5 is protected by Ssa1 and possibly Ssa2 to stay properly folded for biological activity. This chaperone function may block unwanted degradation of Ste5 in the cytoplasm and the nucleus and potentially influence accessibility to binding partners. On the other hand, the more misfolded conformers or those molecules regulated to be turned over are expected to be targeted for degradation either by the proteasome and/or through the vacuole. It is tempting to speculate that Ssa1 (and possibly Ssa2) may also negatively or positively regulate the pathway of Cln/Cdc28 phosphorylated Ste5 that is degraded by the SCF$^{Cdc4}$ ubiquitin ligase complex in the nucleus. A positive regulatory function would be expected to yield an increase in Ste5 abundance in the *ssa1 ssa2* double mutant, and we currently lack this type of evidence. A protective function by Ssa1 would be expected to prevent degradation in the nucleus. On the other hand, we repeatedly found that Ssa1 and Ssa2 promoted overexpressed Ste5 to accumulate as high molecular weight species that could potentially be ubiquitinylated species (Figs 3D, 4C–4E and 7J–7L, [37]). Thus, further work is needed to determine whether Ssa1 and or Ssa2 have direct or indirect roles in either promoting or preventing ubiquitin-dependent degradation of Ste5.

Ssa1 and Ssa2 are known to sequester misfolded and aggregated proteins into a variety of inclusion bodies. We screened for and identified a Ste5 mutant, Ste5L610/614/634/637A-Myc9, with greater propensity to form punctate foci in cells, particularly when Ssa1 was overexpressed (Table 3, Fig 7, Materials and Methods). The Ste5L610/614/634/637A-Myc9 punctate foci induced by Ssa1 appear to be in the cytoplasm and near the nucleus with a few

detected overlapping the DAPI stained nuclear DNA (Figs 7 and S4, Table 3). The cytoplasmic location and shapes of the Ste5L610/614/634/637A-Myc9 punctate foci and their presence during vegetative growth tend to rule out IPOD, cytoplasmic granules [114,115], and aggresomes [91], but they could conceivably be JUNQ/INQ [65–67], Q-bodies/CytoQ bodies [111], stress foci [67], sequestrosomes [60] or protein aggregates that are moved by the polarisome and myosin [116], or HSP-containing aggregates that accumulate during aging [117,118]. We looked for similarly localized proteins in published data including heat shock proteins, aggregated proteins that associate with Ssa1 [e.g. 116,117], various inclusion bodies, and proteins that are markers for organelles and vesicles. The Ste5L610/614/634/637A-Myc9 pattern of punctate foci appears to be most similar to aggregated Ste11DN$^{K444R}$ [66,114,116], and punctate foci of misfolded Ssa1-GFP and von Hippel-Landau protein (VHL-GFP) that accumulate in *hsp104* and *hsp82* mutants [117,118] and Q bodies associated with Ssa1-GFP, Hsp104-mCherry and Hsp82-GFP [41]. Further work is needed to classify the Ste5L610/614/634/637A-Myc9 foci and determine their function.

It was not known whether Ssa1 and Ssa2 regulate the mating pathway. We provide evidence that Ssa1 and Ssa2 are required for full activation of the mating pathway MAPK cascade at the level of activation of Fus3 and Kss1 MAPKs, the ability of Fus3 to efficiently phosphorylate substrates, and ability of the *FUS1* promoter to be fully active basally and during α factor induction, which results in less efficient G1 arrest and shmoo formation (Figs 9–11). Feedback phosphorylation of Ste7 by Fus3 appears to be unaffected. The published high throughput data support potential interaction between Ste5 and Ssa1, Fus3 and Ssa1, but not between Ssa proteins and Kss1 or Ste7, although Kss1 is predicted to interact with Sse2 which binds Ssa1 and Ssa2 (Table 8). Fus3 may be inhibited by Ssa1 and/or Ssa2, because its abundance was slightly elevated 30–50% in the *ssa1D ssa2D* strain (S2 Table, 1.2 +/ 0.10 S.E. to 1.5 +/- 0.65 S.E.). Kss1 may also be inhibited by Ssa1 and/or Ssa2, however we did not obtain good enough quantitation owing to high antibody background. Ste7 and Tcm1 appeared unaffected (Fig 10, S2 Table), but minor effects might not be detected by immunoblot analysis. The relatively equivalent coomassie blue staining of Ste5-Myc9 and Ssa1 in the Ste5-Myc9 immunoprecipitation (Fig 1) and reduced MAPK Fus3 activation in the *ssa1D ssa2D* strain (Figs 9 and 10) raises the possibility that Ssa1 regulates Ste5 while Ste5 is in complex with additional signaling components. Further work may distinguish whether Ssa1 regulates the conversion of Ste5 into forms that can bind signaling partners or directly regulates Ste5-kinase signaling complexes and whether Ssa1 and or Ssa2 promote SCF$^{Cdc4}$ degradation of Ste5 and play a role in degradation of Fus3 and Kss1 independently or while in complex with Ste5.

There is a strong bias towards Ssa1 followed by Ssa2 and Sse1 among the 25 HSP70 network proteins and 7 HSP90 network proteins we examined for potential interactions with mating and invasive growth proteins in published databases. Ssa1 is predicted to regulate over 79% of the 221 proteins examined and 95% of core signaling proteins compared to ~55% for Ssa2 and Sse1, 13% for Ssa3 and 14% for Ssa4, with similar trends for invasive growth (Table 8, Fig 12).

We found, surprisingly, that mating still occurs in all *ssa1-ssa4* combinations tested, despite numerous predicted mating pathway targets for the Ssa proteins, (Fig 10, Table 7). Therefore, compensatory mechanisms must permit mating during loss of Ssa quality control and reduced mating pathway signaling. How might loss of Ssa1 and Ssa2 be compensated? One possibility is that a threshold level of active Ste5 conformers permits mating, perhaps with folding assistance from Ssb1 and Ssb2 during translation; similar reasoning can be applied to other proteins. Ste5 is very rate-limiting and low levels can be sufficient for function [9]. Perhaps unfolding leads to a small pool of active derepressed conformers of Ste5. Perhaps other mating pathway proteins are derepressed, expressed at higher levels (e.g. Fus3 and Ste11) or have fewer inhibitors. Intriguingly, genetic analysis suggests weakened binding of Ste5 to kinase

may enhance mating [91], raising the possibility that less optimal conformers of Ste5 with weakened binding to binding partners may enhance mating. In addition, stress induced from changes in cell wall integrity from *ssa1D ssa2D* mutations may trigger activation of the invasive growth pathway to activate Kss1 which can partially substitute for Fus3 in strains that harbor a functional *KSS1* allele (Fig 1A, [119]). Perhaps the unfolded response (UPR) becomes activated, triggering the cell surface mucin Msb2, which stimulates signaling through Ste11 to Kss1 [120]. Redundant chaperone folding systems such as other Hsp70 and Hsp90 chaperones may provide compensatory functions that promote mating, in addition to known compensatory increases in Ssa4, Hsf1 and Msn2/Msn4 that occur in absence of Ssa1 and Ssa2, or compensatory changes in ubiquitin-dependent degradation [44,64,68]. Notably, we found that Ssz1 Hsp70, Fes1 and three dual regulators of Hsp70 and Hsp90 (i.e. Ydj1, Sse2, and Sti1) were needed for proper shmoo morphogenesis (S5 and S6 Tables, Table 8), providing evidence of multiple compensatory mechanisms that preserve mating during normal growth and stress. Fes1 and both Sse1/Sse2 are also needed for invasive growth "mat" formation [121], indicating functional overlap for mating and invasive growth.

## Supporting information

**S1 Fig. Analysis of full-length Ste5 (1–917) coding sequence with bioinformatics algorithms.**
(TIF)

**S2 Fig. *ssa1D ssa2D* double mutant growth at different temperatures.**
(TIF)

**S3 Fig. Analysis of Ste5ms crystal structures 3FZE and 4F2H overlapping the VWA domain (residues 583–786).**
(TIF)

**S4 Fig. Enlarged image and tally of punctate foci in cells expressing ste5L610/614/634/637A with overexpressed Ssa1-GFP.**
(TIF)

**S5 Fig. Comparison of hydrophobicity, bulkiness, average buried and disorder of amino acid residues of wild type Ste5 and mutant Ste5 L610/614/634/637A.**
(TIF)

**S6 Fig. Enlarged long and short exposures of Fig 8A anti-Kss1 immunoblot and FUS1:: UbiY-lacZ standard error values and p-values.**
(TIF)

**S7 Fig. Representative fields of wild type and Hsp70 mutants treated with 2.5 μM α factor.**
(TIF)

**S8 Fig. Appendix 1. Representative fields of wild type and hsp70 mutants treated with 5 μM αF.**
(TIF)

**S1 Table. Plasmids and strains used in this study.**
(XLSX)

**S2 Table. Summary of ImageJ densitometry values.**
(XLSX)

**S3 Table. List of positive and negative regulators of mating and invasive growth proteins screened for interaction with Ssa1, Ssa2 and other HSP70 network proteins in published databases.**
(XLSX)

**S4 Table. Summary of interaction data on 212 mating pathway and 121 invasive growth pathway proteins with Hsp70 chaperones and co-chaperones.**
(XLSX)

**S5 Table. Morphology of wild type and *fes1D* mutant strains before and after exposure to α factor.**
(XLSX)

**S6 Table. Cell morphology of Hsp70 family mutants before and after α factor treatment.**
(XLSX)

**S1 Appendix. Benchmark molecular weight standards package insert documentation.**
(PDF)

**S1 File. Supporting information text.**
(DOCX)

# Acknowledgments

We greatly thank Yuanyi Feng (Professor, Uniformed Services University of the Health Sciences, Bethesda, MD) and Yunmei Wang (Professor, Case Western Reserve School of Medicine, Cleveland, OH) for research contributions when they were in the lab and William (Bill) S. Lane, John Neveu for the mass spectrometry (Harvard Microchemistry/Mass Spectrometry and Proteomics Resource Laboratory of Harvard University), Maosong Qi (Physician, Lowell General Hospital, MA) for help on kinase assays and Yunmei and Bill for helpful comments. We thank E. Craig (U. Wisconsin), B. Errede (U. North Carolina, Chapel Hill), P. Pryciak (U. Massachusetts Worcester Medical School), A. Truman (U. North Carolina, Charlotte), A. Kumar (U. Michigan), Z. Moqtaderi and K. Struhl (Dept. BCMP, HMS) for strains, plasmids, protocol details or archived published data. E.A.E. designed the project, supervised, analyzed the data, wrote the paper and made the figures and tables using drafts of figures, tables and notebook information from all researchers. Coauthors provided input on manuscript drafts. We thank Henrik Dohlman and colleagues for careful prior review of this manuscript.

# Author Contributions

**Conceptualization:** Paul B. Maslo, Elaine A. Elion.

**Data curation:** Francis W. Farley, Ryan R. McCully, Paul B. Maslo, Mark A. Sheff, Elaine A. Elion.

**Formal analysis:** Ryan R. McCully, Paul B. Maslo, Elaine A. Elion.

**Funding acquisition:** Elaine A. Elion.

**Investigation:** Francis W. Farley, Ryan R. McCully, Paul B. Maslo, Lu Yu, Mark A. Sheff, Homayoun Sadeghi, Elaine A. Elion.

**Methodology:** Francis W. Farley, Ryan R. McCully, Homayoun Sadeghi, Elaine A. Elion.

**Project administration:** Elaine A. Elion.

**Resources:** Elaine A. Elion.

**Software:** Elaine A. Elion.

**Supervision:** Elaine A. Elion.

**Validation:** Francis W. Farley, Ryan R. McCully, Paul B. Maslo, Mark A. Sheff, Homayoun Sadeghi, Elaine A. Elion.

**Visualization:** Francis W. Farley, Ryan R. McCully, Lu Yu, Mark A. Sheff, Elaine A. Elion.

**Writing – original draft:** Elaine A. Elion.

**Writing – review & editing:** Francis W. Farley, Ryan R. McCully, Paul B. Maslo, Lu Yu, Mark A. Sheff, Elaine A. Elion.

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
