## [Decision Letter · Decision Letter 0]

12 Oct 2022

PONE-D-22-25493Effects of HSP70 chaperones Ssa1 and Ssa2 on Ste5 scaffold and the mating mitogen-activated protein kinase (MAPK) Pathway in Saccharomyces cerevisiaePLOS ONE

Dear Dr. Elion,

Thank you for submitting your manuscript to PLOS ONE. After careful consideration, we feel that it has merit but does not fully meet PLOS ONE’s publication criteria as it currently stands. Therefore, we invite you to submit a revised version of the manuscript that addresses the points raised during the review process. Both reviewers note quite a few items that could use attention in the manuscript. Please take this opportunity to clarify any misunderstandings through the addition of more explanatory text and/or additional controls if they are available.

We look forward to receiving your revised manuscript.

Best Regards,

Kathy Borkovich

Katherine A. Borkovich, Ph.D.

Academic Editor

PLOS ONE

Journal Requirements:

3. Please expand the acronym “NIH” (as indicated in your financial disclosure) so that it states the name of your funders in full.

"The NIH RO1 GM46962 funded salaries and research supplies for Elaine Elion, Francis Farley, Ryan McCully, Paul Maslo, Lu Yu, Mark Sheff and Homayoun Sadeghi either directly or indirectly through tenure transfer funds. Once NIH government funding ended Elaine Elion was supported by salary support from Harvard University but was required by her Department to pay out of own pocket for computers, software, computer accessories, cleaning supplies, PPE, and supplies relating to writing and notebook keeping. "

"The authors declare they have no conflicts of interest with the contents of this manuscript. This work was supported by a grant from the N.I.H. (GM46962). A portion of the methods are published in a thesis by Homayoun Sadeghi in the field of Biology for a Master of Liberal Arts in Extension Studies from Harvard Extension School, June 1999. Y.F., Y.W. , F.W.F., R.M., L.Y., M.A.S., W.S.L,, H.S., E.A.E. did the experiments. E.A.E. designed the project, analyzed the data, wrote the paper and made the figures and tables. Coauthors provided input on manuscript drafts. We thank H.D. and colleagues for careful review of this paper."

"The NIH RO1 GM46962 funded salaries and research supplies for Elaine Elion, Francis Farley, Ryan McCully, Paul Maslo, Lu Yu, Mark Sheff and Homayoun Sadeghi either directly or indirectly through tenure transfer funds. Once NIH government funding ended Elaine Elion was supported by salary support from Harvard University but was required by her Department to pay out of own pocket for computers, software, computer accessories, cleaning supplies, PPE, and supplies relating to writing and notebook keeping."

7. PLOS ONE now requires that authors provide the original uncropped and unadjusted images underlying all blot or gel results reported in a submission’s figures or Supporting Information files. This policy and the journal’s other requirements for blot/gel reporting and figure preparation are described in detail at https://journals.plos.org/plosone/s/figures#loc-blot-and-gel-reporting-requirements and https://journals.plos.org/plosone/s/figures#loc-preparing-figures-from-image-files. When you submit your revised manuscript, please ensure that your figures adhere fully to these guidelines and provide the original underlying images for all blot or gel data reported in your submission. See the following link for instructions on providing the original image data: https://journals.plos.org/plosone/s/figures#loc-original-images-for-blots-and-gels. 

8. Please upload a copy of Supporting Information Figure/Table/etc. Table S9 which you refer to in your text on page 23.

Reviewers' comments:

Reviewer's Responses to Questions

**Comments to the Author**

1. Is the manuscript technically sound, and do the data support the conclusions?

Reviewer #1: Partly

Reviewer #2: No

2. Has the statistical analysis been performed appropriately and rigorously? 

Reviewer #1: No

Reviewer #2: No

3. Have the authors made all data underlying the findings in their manuscript fully available?

Reviewer #1: Yes

Reviewer #2: No

4. Is the manuscript presented in an intelligible fashion and written in standard English?

Reviewer #1: Yes

Reviewer #2: No

5. Review Comments to the Author

Reviewer #1: Protein molecular chaperones play multiple roles in cellular signaling pathways, including repressive/regulatory, structural, as well as modulating stability and turnover. The mitogen-activated protein kinase cascade that regulates mating and osmotic/cell wall stress in the yeast Saccharomyces cerevisiae has been a model system for decades to understand ligand-activated signaling in eukaryotes. However, what roles chaperones play in supporting these cellular activities remain incompletely understood. In this report, a proteomics study focused on the Ste5 scaffold protein revealed tight association with the highly expressed constitutive Hsp70-class chaperones Ssa1 and Ssa2. A series of experiments conclusively demonstrate that Ssa1/2 are required for proper folding, stability, targeting and function of Ste5 and by extension, the mating pathway.

This report includes a great deal of work analyzing the interactions between Ssa1/2 and Ste5. So much so that at times it is difficult to parse exactly what question is being asked and whether it is being properly answered. The results, while not always clear, are nevertheless interesting and compelling. Multiple recommendations are made below to tighten up presentation and interpretation of the data to facilitate greater impact for the story, including greater attention to detail in figure presentation and clarity.

However, strangely absent from the paper and the discussion, especially, is the wealth of data demonstrating that the Hsp90 chaperone system (Hsc/p82, Ssa1,2, Ydj1, Sse1, Cdc37, Sti1, etc…) is required for proper functioning of the same pathway via direct interactions with the MAPKKK Ste11. In fact, it would be very difficult to parse many of the outcomes of this new report with previously published data given that the phenotypes are identical. Schmoo formation, mating, Fus1-lacZ expression – all of these parameters are defective in cells lacking the proteins described above, including Ssa1/2. The authors need to address this confounding scenario early on and clearly separate the biochemical work supporting a direct role in chaperoning the Ste5 protein from signaling work that is complicated by the aforementioned Ste11 regulation.

Global recommendations:

1. There are a great many Western blots in this article, and most are not clearly labeled and all are packed so densely on the page it is challenging to discern which panel is which. The authors are advised to reduce the font size on all those panels (and use the same font, preferably) to simplify reading. Lane numbers can be eliminated for all but the largest blots. All blots should be converted to greyscale as well, as some are blue from the original film scan.

2. Figures 7 and 8 have tables within the figures that are difficult to interpret. These should be converted to bar charts, or pulled out into true tables along with the others.

Specific recommendations:

3. Why is the letter “D” used for genetic deletes rather than the traditional “∆”? This is unusual and a little confusing throughout.

4. How precisely was the interpretation that “the majority of Ste5 is associated with Ssa1/2” as stated on p.13 made?

5. P. 15, reference to Fig. 3E is incorrect as Fig. 3D.

6. Fig. 2A, bottom, truncations is mis-typed as “fruncations”.

7. Fig. 2G – the blot image includes handwritten notes on the film that should be removed.

8. Fig. 3B – Blot quality is poor for the CEN-expressed Ste5 – any quantitation of these signals is going to be unreliable. Also not best practice to present four isolated lane strips from a blot. This experiment should be repeated with a proper single gel and better immunoblotting conditions.

9. For the opposite reason, panel 3C is massively overexposed and no densitometry or even visual comparison is possible. Same for Fig. 5D.

10. Fig. 4A – Comparing lanes 1-3 to 4-6, it seems very unlikely that the signals in the former are 4x the latter, and the pellet lanes are so low as to be non-interpretable, generating serious doubts about the usefulness of this experiment.

11. Page 16 – why do the authors refer to D-glucose, D-galactose, and then galactose? Since no one would assume L-sugars are being used this nomenclature seems unnecessary.

12. Fig. 6 – are panels L and M the same image? Overall this figure is very confusingly presented, and the single panels without controls or isogenic strains are incomplete as shown.

13. Fig. 8A,B are unnecessary, at least in the main text. Panel C is uninterpretable.

14. Fig. 9A – plot format is unusual and inconsistent with other plotting styles. It is also well-established that ssa1∆ and ssa2∆ phenotypes are generally non-existent or at least minor compared to loss of both chaperones together. Not sure this point needs to made again and again in the manuscript. Fig. 9B – remove absolute values from chart as the relative levels are what’s most important. The authors should also consider statistical testing these differences with replicates and a t-test, at least.

15. Fig. 9F – where is the signal for Fus3-HA? The interpretation of the data is rather convoluted by normalizing to purported kinase pulled down when the reader cannot see that.

16. Fig. 10. -panel “e” is labeled in lower case.

Reviewer #2: Comments on manuscript titled ” Effects of HSP70 chaperones Ssa1 and Ssa2 on Ste5 scaffold and the mating mitogen-activated protein kinase (MAPK) Pathway in Saccharomyces cerevisiae”

The study focuses on Ste5 interaction with Ssa family of proteins in yeast. Using immunoprecipitation, authors find that Ste5 interacts with Ssa chaperone. Further work aims to examine the role of this interaction in Ste5 stability, and activation of other interactors of Ste5 such as Fus3 and Kss1. Though the study is of interest to the chaperone community, the experiments carried out has various limitations including lack of appropriate controls before a meaningful conclusion could be drawn. Also the manuscript is written with lots or typos, incorrect reference to figures, and in a way that its very difficult to understand in its correct form. Following are some of the comments:

(1) Abstract: Ssa2 is a paralog (rather than ortholog) of Ssa1 .

Figure 1B: “The signal transduction partners ……the wide IgG2a band”. What is the basis of identification of band as Ste7, Ste11, and Fus3 and Kss1. No relevant data is provided that confirms the identity of these bands (such as mass-spect or detection with antibodies). Further the bands itself are not clearly visibile on the Gel image. For example there is no band on the gel-image at 80kDa. May be authors should label the bands.

Figure 1B: Densitometry does not align well with the image shown. In the presence of α-factor Ste5 is still more than Hsp70 though authors mention that Ste 5 is 0.6 time of Hsp70.

Page number 13: “The two other bands……………due to its high abundance”. Instead it should be mentioned as “…..due to its relatively high abundance”….

Page 13: “The tandem mass spectra of two……..Ssa1, Ssa2, Ssa3 and Ssa4 (Fig 1C)”. The HPLC profile should be shown. And why only two peaks were picked for analysis. In general the coverage from Mass spect should be at least more than 40-50% of the protein sequence.

Figure 2F: “Ste5-Myc9 also accumulates ……disappear when Cdc28 CDK is mutated”. The authors should show data in the absence of proteasomal inhibitor to conclude that the shift is because of the inhibitor.

Page 15: “In the wild type……………Ste5 aggregation and degradation”. Figure is wrongly mentioned at 3D instead of 3E.

Figure 3E: Lane 1 and 8 show contradictory results. In lane 8 there is no detectable Ste5 similar to as in case of Ssa1Dssa2D cells.

Densitometry analysis has been reported for many Figures, and the quantitative estimation is reported unti 2nd decimal place. How many times the experiment was carried out for the densitometry analysis to report the data? Further, error in estimation should be reported.

Page 16: “The effects are most likely posttranscriptional ………..by Ssa1, Ssa2 or Hsf1 or Msn2/Msn4”. No evidence in support of posttranscriptional effect is provided. qRT-PCR should be carried out.

Page 16: Figure 3H is mentioned but I could not find the Figure in the version that I received.

Figure 4A: Molecular weight marker is missing.

Figure 5D: “The abundance of Ste5-Myc9 was several-fold less in the ssa1D ssa2D extracts compared to wild type (Fig 5D lanes 9,10) ……… Ste5-Myc9 dimerizes equivalently or slightly more in the ssa1D ssa2D extracts than the wild type extracts (Fig.5D)”. Its not clear that about which lane corresponds to Ssa1DSsa2D double deletion (does not seem to be lane 9 and 10 as reported). Further, the basis on which Ste5-Myc9 dimerization is reported, is not clear and thus should be elaborated for clarity.

Figure 6B: “We found Ssa1-GFP to localize throughout the cytoplasm and nucleus as shown by absence of nuclear exclusion”. The colocalization study should be carried out to support this conclusion.

Page 19: “Moreover, Ssa1-GFP was clearly enriched in the nucleus in addition to being in the cytoplasm in the ssa1D ssa2D strain that lacks endogenous Ssa1 and Ssa2 (Fig 6B)” : Should it be Figure 6C-D?

Figure 6E and 6F: Authors claims that wt Ste5 rarely forms foci however Ste5(1-242)-GFP fusion does form. However this is not apparent from these figures as in both cases similar foci are shown.

Figure 7B is a table. Please show images also.

Page 25: “Thus, Ssa1 and Ssa2 are essential for basal activation of Fus3 and full basal activation of Kss1 and essential for a factor induced activation of both Fus3 and Kss1. Ssa1 and Ssa2 are therefore required for efficient response to mating pheromone in addition to regulating Ste5 abundance and integrity”. – Which data authors are refering to? More clarity is required to support this conclusion.

There are lots of typos such as “Ssa1 Is overexpressed” instead of “Ssa1 is overexpressed”. Page 13: “…….excised from the ge….”

I could not find Table 1-4 in the version I received.

6. PLOS authors have the option to publish the peer review history of their article (what does this mean?). If published, this will include your full peer review and any attached files.

Reviewer #1: **Yes: **Kevin A Morano

Reviewer #2: No

---

## [Author Response · Author response to Decision Letter 0]

11 Jul 2023

Response to Reviewer 1 Dr Morano 

……………………………..

My responses have been inserted after each reviewer comment. Reviewer #1 comments and my responses have been numbered as 1-# to distinguish from other reviewer. Reviewer 1 comments are in original helvitica neue 12 point and my responses are in arial 12 point. Lists of Tables, Figures, SupFigures SupTables and Subheadings with first sentence are after the responses for clarity.

[1-1] Reviewer #1: Protein molecular chaperones play multiple roles in cellular signaling pathways, including repressive/regulatory, structural, as well as modulating stability and turnover. The mitogen-activated protein kinase cascade that regulates mating and osmotic/cell wall stress in the yeast Saccharomyces cerevisiae has been a model system for decades to understand ligand-activated signaling in eukaryotes. However, what roles chaperones play in supporting these cellular activities remain incompletely understood. In this report, a proteomics study focused on the Ste5 scaffold protein revealed tight association with the highly expressed constitutive Hsp70-class chaperones Ssa1 and Ssa2. [1] A series of experiments conclusively demonstrate that Ssa1/2 are required for proper folding, stability, targeting and function of Ste5 and by extension, the mating pathway.

[1-1] Response: We thank reviewer 1 for finding that we have presented a series of experiments that conclusively demonstrate that Ssa1/2 are required for proper folding, stability, targeting and function of Ste5 and by extension, the mating pathway. 

[1-2a] This report includes a great deal of work analyzing the interactions between Ssa1/2 and Ste5. So much so that at times it is difficult to parse exactly what question is being asked and whether it is being properly answered.

[1-2a] Response: I have edited the text to better explain what question is being asked and that the approach being taken is appropriate to answer the question being asked and removed redundancy and tried to add more clarity.. The esteemed reviewer did not offer suggestions on how to improve the manuscript so it is difficult to know exactly where the problems are for him, as the comment is very general. A list of the headings and transition sentences is here

Subsection headings and first sentences: 

1. Ssa1 co-purifies with Ste5-Myc9 

 To identify proteins that associate most prominantly with Ste5, we purified Ste5-Myc9 from logarithmically dividing cells grown at 30oC before and after � factor treatment (Materials and Methods).

2. Ste5-Myc9 undergoes extensive posttranslational modification, aggregation and fragmentation that is influenced by Cdc28, � factor and temperature 

 The mobility of Ste5 in polyacrylamide gels reflects post-translational modifications from phosphorylation and ubiquitylation (33, 34, 38) and a propensity to aggregate and degrade into fragments (this study). 

3. Ssa1 and Ssa2 positively regulate Ste5 abundance especially at higher temperature 

 We determined the effect of a ssa1� ssa2� double mutation on the abundance of Ste5-Myc9. 

4. Loss of Ssa1 and Ssa2 alters the integrity of Ste5 

We examined the effect of ssa1� ssa2� mutations on the integrity of Ste5, by examining a number of its characteristics we have previously defined.

5. Loss of Ssa1 and Ssa2 interferes with cortical localization of Ste5 

 We examined the effect of ssa1� ssa2� mutations on the localization of Ste5-Myc9 expressed from CEN (~1 copy/cell) and 2� (multiple copies/cell) plasmids using haploid strains generously provided by the Craig lab.

6. Ssa1 and Ssa2 enhance nuclear accumulation of Ste5 

 We determined Ste5 localization in the absence of Ssa1 and Ssa2.

7. Overexpression of Ssa1 promotes punctate foci localization of a Ste5 VWA domain mutant predicted to have greater disorder 

 Given that protein quality control can occur in supramolecular foci in the nucleus and the cytoplasm, and the tendency of Ste5 to aggregate, we examined whether Ste5 localizes in punctate structures that are affected by loss of Ssa1 and Ssa2 or by overexpression of Ssa1.

8. Strains derived from Craig lab diploid MW63 lack Kss1 protein by western analysis

During experiments to assess Fus3 and Kss1 activation, we discovered that EY3136 wild type and EY3141 ssa1� ssa2� ascospores from the Craig lab diploid MW63 are kss1- based on absence of detection of Kss1 protein with anti-Kss1 antibody and anti-phospho-p42p44 antibody that recognizes a conserved peptide in human ERK2 that has homology to Fus3 and Kss1 (Fig S2D).

9. Loss of Ssa1 and Ssa2 reduces the ability of Fus3 and Kss1 to be activated

 We assayed activation of Fus3 and Kss1 in a time course experiment in the bar1� Kss1+ strains WT-B and ssa1� ssa2�-B. 

10. Ssa1 and Ssa2 are not required for feedback phosphorylation of Ste7 but are required for efficient Fus3-HA kinase activity i 

Ste7-Myc is hyperphosphorylated in the presence of � factor through feedback phosphorylation by Fus3 and Kss1. We looked at basal feedback phosphorylation of Ste7-Myc by Fus3 using a CYC1prom-STE7-MYC gene that is not regulated by either the mating pathway or heat shock proteins Ssa1, Ssa2, Hsf1.

11. Ssa1 and Ssa2 have either inhibitory or negligible effects on abundance of Ste7, Fus3, and Kss1 

 We determined whether loss of Ssa1 and Ssa2 has obvious effects on the abundance of MAPKK Ste7 and MAPKs Fus3 and Kss1. We did not examine MAPKKK Ste11 whose abundance declines during prolonged � factor stimulation and may be negatively regulated by Ste5, Ubc4 and Ubc5 (100) and is known to be positively regulated by Hsc82, Hsp82, Sti1, Sse1, and Ydj1 (102-104).

12. Loss of Ssa1 and Ssa2 reduces basal and � factor-induced expression of a FUS1::ubiYlacZ reporter gene and causes high standard error 

 We determined the ability of � factor to induce mating pathway responses in the ssa1��ssa2� double mutant and several other ssa mutant strains.

13. Ssa1 and Ssa2 are required for G1 arrest and � factor-induced shmoo formation but not mating 

 Increasing levels of mating MAPK cascade activation are needed for transcriptional activation, G1 arrest and shmoo formation (84). We assayed the contribution of Ssa1 and Ssa2 to G1 arrest and shmoo formation in EY3136 and EY3141 (Table S1). 

14. Interaction data suggest Ssa1 is the chaperone most dedicated to the mating pathway, followed by Ssa2 and the HSP70-HSP90 network 

 To investigate compensatory protein quality control and relative importance of different HSP70 chaperones to the mating and invasive growth pathways, we made an inventory of published predicted physical interactions between Hsp70 and Hsp90 protein networks and 212 mating pathway and 121 invasive growth pathway proteins (Table S4).

15. Hsp70 and Hsp90 regulators Fes1, Ydj1, Snl1, Sse2, Sti1 and Ssz1 are also required for shmoo morphogenesis 

 To further explore compensatory protein quality control that could occur in the absence of Ssa1 and Ssa2, we determined whether mutations in other Hsp70 and Hsp70-Hsp90 network proteins affect the ability of cells to follow through on the multiple G�� (Ste4/Ste18) and G� (Gpa1) -mediated responses that induce cell cycle arrest in G1 phase and formation of shmoos. 

 [1-2b] The results, while not always clear, are nevertheless interesting and compelling. Multiple recommendations are made below to tighten up presentation and interpretation of the data to facilitate greater impact for the story, including greater attention to detail in figure presentation and clarity.

[1-2b] Response: Thank you for finding the results interesting and compelling. From this esteemed reviewers response it is not possible to know what is meant by the very general phrase “The results, while not always clear,” Specifics would be helpful. Nevertheless, I have made a number of changes which I hope improves presentation and impact. 1) I have tightened up presentation in the text and figures and tables with respect to details. 2) I have added standard errors, p-values in numerous places. 3) In response to request for more on HSP90, I have expanded the number of HSP70-HSP90 network proteins in Table S5 from the original 4 to 8 and now include Hsp82, Hsc82, Sba1 and Cdc37. 4) I have added a new comparative analysis that includes Hsp90 network proteins as a chart figure in Figure 12A adjacent to the previous Chart figures in Figure 12B-E that were in supplemental for greater impact, 5) I have moved controls from supplemental to figures in some places to enhance greater impact. 6) I have added additional control tallies for the table on punctate foci. 7) The figures and tables have been reordered in some places for clearer presentation in results. 

[1-3a] However, strangely absent from the paper and the discussion, especially, is the wealth of data demonstrating that the Hsp90 chaperone system (Hsc/p82, Ssa1,2, Ydj1, Sse1, Cdc37, Sti1, etc…) is required for proper functioning of the same pathway via direct interactions with the MAPKKK Ste11. 

[1-3a] Response: I am very heartened that Dr Moreno finds the data interesting and compelling and have taken suggestions to improve clarity of presentation and enhance impact. Because of length and desire to focus on HSP70, I did not write expansively on HSP90 in my descriptions in the introduction or results in the first submission. However, I have taken Dr Morano’s suggestion quite seriously and have made a number of changes as explained here. As Dr Morano knows, Hsp90 and Hsp70 can act alone or together. Hsp82/Hsc82 has numerous co-chaperones including Sti1, Cdc37, Chs1, Sba1 Cpr6, Cpr7, Sse1, Hch1 and Aha1 (SGD). I did mention Hsp90 and did choose several proteins that are considered intersection proteins that associate with and regulate both Hsp70 and Hsp90 in the analysis of putative interactors in Table S5 (i.e. Ydj1, Sse1, Sse2, Sti1, Ssa1, Ssa2) together with the phenotypic analysis of HSP70 network mutants ydj1�, sse1�, sse2�, sti1� together with ssa1� ssa2� now in Fig 11 with improved fields of cells and Figure S7, Figure S8,Table 6, Table S7, Table S8. And a new Figure 12 that highlights comparisons between Hsp70 and Hsp90 proteins and includes relative interactions with the 5 kinases in the pathway. I have increased discussion and reference to work that pertains to mutant Ste11K444R. 

In this second submission I have made the following changes to improve impact:

Abstract last sentence “suggests Ssa1 is the major Hsp70 chaperone for the mating and invasive growth pathways and reveals several Hsp70, Hsp90 and other chaperone-network proteins required for mating morphogenesis. 

Addition to the introduction: “Major functions of Ssa1 and Ssa2 include assisting the ribosome-associated Hsp70s Ssb1 and Ssb2 to fold emerging polypeptides during translation (57-58) and transfer polypeptides to other chaperones such as Hsp90s (i.e. Hsp82 and Hsc82 in S. cerevisiae) which also help fold proteins, into their mature functional forms and collaborate with Hsp70s (61,62).”

CHANGES TO RESULTS SECTION RELEVANT TO STE11 and HSP90:

Interaction data suggest Ssa1 is the chaperone most dedicated to the mating pathway, followed by Ssa2 and the HSP70-HSP90 network 

 To investigate compensatory protein quality control and relative importance of different HSP70 chaperones to the mating and invasive growth pathways, we made an inventory of published predicted physical interactions between Hsp70 and Hsp90 protein networks and 212 mating pathway and 121 invasive growth pathway proteins (Table S4). The list includes eleven Hsp70 proteins Ssa1-Ssa4, Ssb1, Ssb2, Ssz1 (cytosolic), Ecm10 , Ssc1 (mitochondrial) and Kar2, Lhs1 (ER), two Hsp40/DnaJ family proteins (Ydj1, Sis1/BAG-1), two nucleotide exchange factors (Fes1/HspBP1 homolog, Snl1/Bag-1 homolog), 2 Hsp110 proteins (Sse1 and Sse2), two Hsp70-associated disaggregation chaperones (Hsp104, Hsp42), two Hsp90 proteins (Hsc82, Hsp82), and three Hsp90 co-chaperones in the HSP70-HSP90 network, Sti1 (which also interacts with Ssa1), Cdc37 and Sba1. Most data came from unconfirmed high throughput experiments. All cells in the published experiments had been grown at either RT or 30oC in logarithmic phase in either YEPD, SCD or SILAC medium and were not subjected to heat shock. Most of the interaction entries came from references 56, 80,99, and 105. 

 The co-translational chaperones Ssb1 and Ssb2 had the most predicted total interactions (64% and 46% respectively) followed by the posttranslational chaperones Ssa1 (53%) and Ssa2 (39%) and the co-chaperone Ydj1 (22%). Hsp82 is predicted to interact with only 12% of the mating and IG proteins and Hsc82 with 25%, followed by Ydj1 (22%), a J domain co-chaperone for both Hsp70 and Hsp90 (Table S5, Fig 12A). The Hsp110 chaperone Sse1 potentially interacts with 37% of pathway proteins compared to only 5% for Sse2. There were fewer predicted interactions for Ssa3 (9%), Ssa4 (14%), Sse1 (5%), Ssc1 (11%), Ssq1 (1.4%), Ssz1 (15%), Ecm10 (2.3%) and Kar2 (6%), Sis1/BAG1 (15%) (Fig 12) . Ecm10, Kar2, Lhs1, Fes1, Snl1, and Sba1 had no predicted interactions with the 41 core signal transduction and transcription proteins of mating and invasive growth (Fig 12A, Table S5).

We compared the percent distribution of Ssa1, Ssa2, Ssa3, Ssa4, Sse1 and Sse2 interactions with mating pathway and invasive growth pathway proteins and the extent of overlap redundancy between chaperones. Ssa1 had the greatest number of predicted mating pathway and invasive growth pathway clients (i.e. 79% and 77% respectively) followed by Ssa2 (55%, 52% respectively) (Figure 12B-E). In contrast, Ssa3 was 13% for mating and 18% for invasive growth and Ssa4 was 14% for mating and 24% for invasive growth (Figure 12B-E, Table S5). Interestingly, 100% of mating pathway proteins predicted to be regulated by Ssa2 are also predicted to be regulated by Ssa1, and likewise for Ssa3 (100%) and Ssa4 (100%) (Fig 12B-E), with similar overlap for the invasive growth pathway (Fig S8B, 94% for Ssa2, 100% for Ssa3, 100% for Ssa4). The bias towards Ssa1 is pronounced among 29 core mating pathway signaling components (95%) and 25 core invasive growth pathway signaling components (94%) (Fig 12B-E). 

 The literature suggests a balance between Hsp70 promotion of degradation of misfolded proteins and Hsp90 promotion of folding to full maturation (60,64,103,106). Current models suggest that Hsp90 uses both Ssa-dependent and Ssa-independent routes to fold proteins (40, 41, 44, 102-104 107). Hsp70-Hsp90 complexes are thought to act downstream of the Ssa chaperones to mature proteins (104,107). The Hsp70-Hsp90 complexes include Ssa1, Ssa2, Hsp82 and Hsc82, and one or more of Ydj1, Sse1, Sse2, Sti1/HOP, Sba1, Cpr and Cdc37 (61,62,103,104,107). Hsp90 has been classified as mainly regulating protein kinases and has some function in the mating pathway and is required for full stability of Ste11�N, a constitutively active mutant kinase that does not bind Ste5 (102). Ste11�N has reduced ability to induce PRE-lacZ in a �hsp82::LEU2 �hsc82::LEU2 hsp82-G313N triple mutant strain, although a data not shown notation from the Picard laboratory suggests that the �hsp82::LEU2 �hsc82::LEU2 hsp82-G313N strain has no defect in shmoo formation or mating (102, 103). 

We found considerable redundancy for the kinases of the MAPK cascade, with Ste20 predicted to be regulated by Ssb1, Ssb2, Ssa1, Hsp82, Hsc82; Ste11 by Ssb1, Ssb2, Ssa1, Ssa2, Ssa3, Ssc1, Hsp82, Hsc82; Ste7 by Ssb1, Ssb2, Ssc1; Fus3 by Ssb1, Ssb2, Ssa1, Ssa2, Hsc82; and Kss1 by Ssb2, Ssz1, Hsp82 (Table S5). That Ste7 may be regulated by Hsp82 but not by Ssa1 or Ssa2 (Table S5, 102) is consistent with our finding no obvious differences in abundance of Ste7 in the ssa1� ssa2� strain, whereas the changes in abundance for Ste5 and Fus3 correlate with co-immunoprecipitation and/or predicted interaction data.

 The inventory of Hsp70 and Hsp90 interactions (Table S5, Figs 12) indicates that Ssa1 and Ssa2 are the major regulators for mating and invasive growth pathways with some overlap from Hsp82 and Hsc82 that could be either independent of or dependent upon Hsp70-Hsp90 complexes. The Hsp110 protein Sse1 which cooperates with Hsp70 and Hsp90 (46) is predicted to interact with 50% of mating and 57% of invasive growth proteins (Table S5, Fig 12), suggesting these predicted clients could be involved in tripartite complexes with these chaperones.

Hsp70 and Hsp90 regulators Fes1, Ydj1, Snl1, Sse2, Sti1 and Ssz1 are also required for shmoo morphogenesis 

 To further explore compensatory protein quality control that could occur in the absence of Ssa1 and Ssa2, we determined whether mutations in other Hsp70 and Hsp70-Hsp90 network proteins affect the ability of cells to follow through on the multiple G�� (Ste4/Ste18) and G� (Gpa1) -mediated responses that induce cell cycle arrest in G1 phase and formation of shmoos. Fes1 (Factor Exchange For Ssa1) and Snl1 (homolog of human BAG-1) are nucleotide exchange factors for Ssa1 that have few predicted interactions with mating pathway proteins (i.e. Fbp1, Lsg1, Mpk1/Slt2 for Fes1 and Mpt5 for Snl1, Table S5). Fes1 is required for the release of Ssa1-bound misfolded protein to the proteasome and is not thought to function with Hsp90. The fes1��null mutant (39,40) was defective in shmoo formation with many round cells and the few shmoos that formed often had a novel additional morphology of a small bud on the side (Fig 11G-H; Table 6,Table S6, Table S7, Fig S9, Fig S10, 38.9% +/- 9.7 S.E. for fes1� versus 87.4% +/- 0.4 S.E. for wt). In terms of forming a “long shmoo”, the fes1� mutant formed 1.3% +/- 0.7 S.E. compared to 14.1% +/- 0.6 S.E. by wild type (Table S6, Table S7). Thus, blocking processing of Ssa1-client complexes in the fes1� mutant resulted in as strong a mutant phenotype as a ssa1� ssa2� double mutant, with an additional novel phenotype of either concommittant bud formation at a default polarization site or formation of a shmoo on a dividing cell. Given that the major target of Fes1 function is Ssa1 and Ssa2, these findings provide further support for a role for Ssa1 and Ssa2 in regulating the mating pathway. 

By contrast, loss of the Snl1/BAG-1 nucleotide exchange factor that is a co-chaperone for Ssb1 and Ssa1 Hsp70s had little effect on shmoo formation. The snl1� mutant formed 78% the wild type number of shmoos, although some shmoos were wider with irregular peanut shapes (Fig 11L, Table 6, Table S6, Fig S9). Thus, Fes1 has a more important role in regulating folding of clients in the mating pathway. 

Ssb1 is predicted to interact with 142 mating and invasive growth proteins (Fig 12, Table S5). Surprisingly, the ssb1� mutant underwent nearly normal G1 arrest and shmoo formation and made 88.7% the wild type number of shmoos, although some shmoos were longer than wild-type (Fig 11K; Table S6, Table 6). Perhaps paralog Ssb2 compensates for loss of Ssb1. Ssz1 is an Hsp70 that associates with the ribosome via contact with Zuo1, a ribosome-associated J protein, and is predicted to interact with 4 core signaling proteins (Far1, Kss1, Sst2, Tec1) and 29 other mating and invasive growth pathway proteins (Table S5). Mutation of Ssz1 caused defects in both G1 arrest and shmoo formation (Fig 11L, Fig S9F, Table S6, Table 6). Therefore, Ssz1 may provide unique functions for mating morphogenesis that are linked to translation. 

We also assessed mutations in 4 chaperones with dual functions in the Hsp70 and Hsp90 pathways: YDJ1, STI1, SSE1 and SSE2. Ydj1 is the J domain/Hsp40 adapter that helps Hsp70/Hsp90 complexes be recognized by Rsp5 NEDD E3 ubiquitin ligase for ubiquitinylation and subsequent degradation of associated client protein by the proteasome (106). Of all of the HSP proteins we investigated, Ydj1 is the only one predicted to regulate Gpa1 (G�) which negatively regulates the mating signaling pathway (Table S5). Notably, the ydj1� cells were more enlarged and misshapen during vegetative growth with some cells appearing shmoo shaped (Fig S10), a phenotype that is seen with reduced Gpa1 function. The ydj1� mutant did not efficiently arrest in G1 phase and became more severely misshapen (both unbudded and budded cells), enlarged and elongated after � factor treatment and formed fewer shmoos that were often enlarged and aberrant, with projections that were longer, wider, curved or bent (Fig 11M, Fig S9A-B, Table 6, Table S6). The retention of the ability of the ydj1� mutant to enlarge in presence of � factor suggests the cells are not fully blocked in signal transduction, in agreement with reduced FIG1 expression and Axl1 biogenesis (103, 108, 109). 

Sse1 and Sse2 are ATPase components for Hsp90 and nuclear exchange factors for Hsp70 (40,113). Sse1 has 10 predicted interactions with core mating and invasive growth signaling components (i.e. Far1, Kss1, Msg5, Msn5/Ste21, Ras2, Rga1, Rga2, Ste20, Ste21/Msn5, Ste50) and 71 predicted interactions with other mating and invasive growth pathway proteins (Table S5), whereas Sse2 is predicted to interact with 1 core signaling protein (i.e. Far1) and 10 other pathway proteins (Table S5). Surprisingly, the sse1� mutant had a nearly normal pheromone response for G1 arrest and shmoo morphogenesis and formed 74.4% wild type number of shmoos (Table S6). By contrast, the sse2��mutant only formed 16.9% wild type level of shmoos, although it could undergo G1 arrest and cell enlargement (Fig 11N, Table 6, Table S6). Thus, Sse2 must regulate proteins essential for shmoo formation and perhaps compensates in the sse1� mutant, but not vice versa. This finding is striking because Sse2 is not thought to be important for the regulation of Ste11, suggesting the defect is linked to Hsp70 rather than Hsp90 (104). 

Sti1 is a multifunctional HSP co-chaperone that interacts with yeast Hsp90s and Hsp70s Ssa1 and Ssa2 and regulates transfer of clients from Hsp90 to Hsp70. Sti1 inhibits Hsp90 ATPase activity and activates Ssa1 ATPase activity (110-113). Currently, Sti1 has no known interactions with mating proteins (Table S6), although a sti1� mutation reduced FIG2 expression and Ste11�N association with an undefined isoform of Hsp90 (103). Interestingly, the sti1��mutation increased the number of shmoos by 53% as well as the length of shmoo projections compared to wild type (Fig 11O, Fig S9G, Table S6, Table 6), indicating either enhanced shmoo formation or reduced downregulation of shmoo formation. 

In summary, Fes1, Ydj1, Sse2, and Ssz1 are needed for proper shmoo morphogenesis with their loss causing distinct changes in morphology, whereas Sti1 negatively regulates shmoo formation, consistent with a biochemical role in inhibiting the ATPase activity of Hsp90 (110-113). Sti1 is not essential for signal transduction linked to shmoo morphogenesis when signaling is through the native pathway that is dependent on Ste5 and contrasts prior predictions based on a mutant Ste11�N allele that signals independently of Ste5 (103). The severity of defects was greater for the ydj1�, sse2� and sti1� mutations that impact both Hsp70 and Hsp90 networks. The distinctly different morphologies of these mutants underscores compensatory protein quality control mechanisms that operate together with Ssa1 and Ssa2. 

EE COMMENTS RELATIVE TO DR MORANOs COMMENTS ABOUT STE11: 

 With respect to comments about Ste11, I now carefully pointed out that conclusions in several places in the literature viz Hsp90 and Ste11 were based on analysis of a mutant Ste11�N kinase that does not bind Ste5 or Ste50 and that the behavior of this allele does not exactly reflect normal pheromone responses. We have assayed normal pheromone responses in which all the mating signaling components are in their native state and dependent upon Ste5 and its interactions with Ste11. Our analysis of morphology of ydj1� cells during � factor treatment indicates that they do respond and are not fully blocked in signal transduction as concluded from analysis of an artificial pheromone response element promoter-lacZ fusion (Lee et al Caplan Ref102). In the second example, the sse1� mutant had a nearly normal pheromone response for G1 arrest and shmoo morphogenesis. This is in great contrast to previous work with the dominant Ste11�N allele and conclusions thereof that the sse1� mutant is quite blocked for � factor signal transduction (Ref 104). In the third example, we found that the sse2��mutant failed to form shmoos, although it could undergo G1 arrest and cell enlargement. This finding is unexpected given the work of Mandal et al Morano (Ref104) implicating Sse1 but not Sse2 in the regulation of Ste11. In the fourth example, we found that Sti1 negatively regulates shmoo formation. Sti1 has no known interactions with mating proteins, although a sti1� mutation reduced FIG2 expression and His-tagged Ste11�N association with an undefined isoform of Hsp90 (Lee et al Caplan Ref 103). Thus, Sti1’s complex functions with respect to both Hsp70 and Hsp90 in total are not essential for signal transduction linked to morphogenesis. 

I have not found data in the literature on defects in mating or shmoo formation for hsp82 hsc83 mutants single or double as Dr Morano has written. Shmoo formation is the result of signaling events mediated by the G protein (Ste4, Ste18, Gpa1) which flows through Ste5 and the mating MAPK cascade which regulates transcription along with many other targets including the cytoskeleton. The G protein also recruits the Fus3 target Far1, which regulates cell polarization, actin cables and polarized movement of secretory vesicles. Louvion et al Picard (Mol Biol Cell 9(11):3071-83) did their analysis with a hsp82� hsc82� double mutant harboring a HSP82-G313N point mutant on a CEN plasmid (HH1a MATa Dhsc82::LEU2 Dhsp82::LEU2 HSP82 G313N-CEN/ARS-TRP1 ). They found that “the requirement for Hsp90 is “..possibly at the level of Ste11 or downstream of it” based on co-immunoprecipitation of Hsp82 with HA-Ste11, reduced steady state abundance of HA-Ste11, reduced hyperphosphorylation of Ste7-Myc and reduced FUS1-LacZ expression. They did not present data on G1 arrest, shmoo formation or mating. On pg 3079-3080 they state “The Hsp90 mutant strains that we have tested are not completely defective in Ste11 activity. Unlike ste11 deletion strains, they are able to form shmoos in re-sponse to pheromone, and they can mate albeit with reduced efficiency (our unpublished results).” A statement on unpublished results is not even permitted in PLOS ONE. Louvion et al did not offer an explanation for why they did not appear to have gross defects in mating or shmoo formation. Their findings are now mentioned in the discussion section of the paper making sure to be clear that these are an unpublished result citation, but mentioned in the context of our data with emphasis on functional redundancy being one of several explanations for why both ssa1 ssa2 and hp82 hsc82 strains retain ability to mate. The Caplan/Morano labs did not monitor shmoo formation or G1 arrest in their analyses but rather focused on expression of a pheromone responsive target gene expression reporter, abundance of Ste11 or mutant Ste11 derivatives and association of Ste11 mutant with Hsp90 (isotype not defined in the Caplan paper and Dr (also a Dean) Caplan was unable to access his students notebook to figure out the HSP90 isotype owing to being unable to enter the building due to construction based on our very recent correspondance) and cochaperone and with respect to whether mutations in cochaperones affect interaction with Ste11 and Hsp90. 

The analysis from my lab described in this manuscript provides experimental data that mating still occurs in ssa1 ssa2 mutants and other ssa triple deletion mutants. We describe for the first time defects in G1 arrest and shmoo formation in carefully documented data. We carefully document the phenotype of 4 more mutants known to comprise Hsp90, Hsp70 and Hsp70-Hsp90 complexes and show they have more severe phenotypes and present quantification and photos of fields of cells. We describe novel morphological defects based on my careful assessment of cell morphologies. Our work is novel. Moreover, the careful morphological assessment reveals differences from the hypotheses drawn in the Hsp90 papers in several respects. I have explained this more thoroughly in the revised version. 

[1-3b] In fact, it would be very difficult to parse many of the outcomes of this new report with previously published data given that the phenotypes are identical. Schmoo formation, mating, Fus1-lacZ expression – all of these parameters are defective in cells lacking the proteins described above, and clearly separate the biochemical work supporting a direct role in chaperoning the Ste5 protein from signaling work that is complicated by the aforementioned Ste11 regulation.

[1-3b] Response: I am not clear on what the esteemed reviewer is writing about here. The aforementioned work by Louvion did not find defects in shmoo formation and no data were presented. I have not found shmoo and matng data in the literature for the ssa mutants or the other HSP70-HSP90 network mutants we present. The results and discussion have been improved to include discussion of Ste11 and the dual inputs from Hsp70 and Hsp90 viz interactions with pathway proteins.

Hsp90 and Hsp70 can act alone or together. From the literature, the HSP70-HSP90 complex includes includes Hsp82/YPL240c, Hsc82/YMR186W, Sse1, Sse2, Ydj1, Sti1/HOP1, Sba1/YKL117W , Cdc37/YDR168w and Cpr (cyclophilin). To my knowledge, there are no comparative data yet published on ability of mutations in these genes to interfere with G1 arrest and shmoos morphogenesis in response to mating pheromone � factor as we have done in the case of ydj1, sse1, sse2, sti1, ssa1 ssa2 mutants . Therefore our experiments were justified. However, it is beyond the scope of this paper to analyze hsp82 and hsc82 single and double mutants and the sba1 mutant. we show that the phenotypes of ssa1 ssa1, ydj1, sse1, sse2, sti1, ssa1 ssa2 mutants are not identical (Fig. 7, Table 6, SupTable 7, SupTable 8) consistent with their different roles in protein quality control and different sets of predicted interactions with mating and invasive growth pathway proteins (Table S5). The focus in the paper is one of showing that there are overlapping and redundant inputs for protein quality control that permit signaling and mating when ssa1 ssa2 are deleted (or triple ssa mutants) and that the HSP70-HSP90 intersections are important for responses, with greater severity of defect being found for those mutants that impact both Hsp70 and Hsp90 pathways. I hope the revised submission has been improved and clarified sufficiently. 

[1-3c] The authors need to address this confounding scenario early on 

[1-3c] Response: Please see the detailed response under [1-3A]. In a very long paper already, I now mention Hsp90 complexes in the introduction in more detail than what I already had done and have added a section that is focused on Hsp70-Hsp90 complexes, what was known about Hsp90 and its potential interactions including the published work on Ste11 in the results and discussion. Ste11 is one of 9 out of 25 core mating pathway signal transduction proteins that might be interacting with Hsp90 and my comparative analysis and discussion has incorporated this new information. 

The paper has been organized in a way that had put the biochemistry relating to ste5 first before discussing the mating and signaling data at the end before moving back to the concept of functional redundancy from inputs from both Hsp70 and Hsp90 regulators. This last section is revised for clarity for greater impact. Kindly refer back to my responses above for more details. 

Global recommendations:

[1-4] There are a great many Western blots in this article, and most are not clearly labeled and all are packed so densely on the page it is challenging to discern which panel is which. The authors are advised to reduce the font size on all those panels (and use the same font, preferably) to simplify reading. Lane numbers can be eliminated for all but the largest blots. All blots should be converted to greyscale as well, as some are blue from the original film scan.

[1-4] Response: I have clarified labeling, keeping font within the 8-12 range and PLOS ONE size constraints. I have reorganized figures so they are less crowded (there are now 10 figures). All blots are converted to grey scale. I have double checked all figures and if anything appeared unclear. I have improved presentation in text and figures and tables and tried to spread out spacing and in so doing have made an additional figure. I have also replaced RGB scans of blots with black and white/grey scale where needed. For your convenience, a list of the changes to figures and tables is within the cover letter and at the end of this response to reviewers. 

[1-5] Figures 7 and 8 have tables within the figures that are difficult to interpret. These should be converted to bar charts, or pulled out into true tables along with the others.

[1-5] Response: The tables within figures are now in separate Tables with S.E. values. The figure 9A graph has been converted to bar chart to match the Fig 9B and this now called Figure 10A-B. I have previously defined the high level of standard error in the Fus1-lacZ’s for the ssa1 ssa2 double mutant as being a characteristic phenotype A new Fig S6C has normalized standard errors for three sets of data in Fig 10A and 10B. 

The introduction now has a better explanation about Kss1: “Kss1 regulates transcriptional repressors Dig1, Dig2 and transcription activator Ste12 (and represses Ste12 as an inactive kinase via Dig1 (5)), in addition to Tec1 and other transcription factors which together promote invasive growth (Fig 1A). “ 

Specific recommendations:

[1-6] 3. Why is the letter “D” used for genetic deletes rather than the traditional “∆”? This is unusual and a little confusing throughout.

[1-6]. Response: Short response- D’s have been changed to deltas. Long response:

D’s are used by the biotech company Research Genetics that supplies the yeast knockout strain collection shown here: 

“Search for Strains Made by the Saccharomyces Deletion Project 

 Available from 

 Research Genetics and from American Type Culture Collection 

 Strains also available in Europe from EUROSCARF

Genotypes of the Strains

BY4730 MATa leu2D0 met15D0 ura3D0 

BY4739 MATalpha leu2D0 lys2D0 ura3D0 

BY4741 MATa his3D1 leu2D0 met15D0 ura3D0 

BY4742 MATalpha his3D1 leu2D0 lys2D0 ura3D0 

BY4743 4741/4742 

homozygous diploids are in the BY4743 background unless 4730/4739 is indicated”

also see 2009 excerpt at https://www.ncbi.nlm.nih.gov/geo/query/acc.cgi?acc=GSM346111

Friday, April 21, 2023

[1-7] 4. How precisely was the interpretation that “the majority of Ste5 is associated with Ssa1/2” as stated on p.13 made?

[1-8]. Response: Your question is well taken. The interpretation is based on densitometry of Coomassie-stained polyacrylamide gels of immunoprecipitated Ste5-Myc9. I have added to this sentence “in the co-immunoprecipitate” to be more precise. The revised version now reads: “the majority of Ste5 in the co-immunoprecipitate is associated with…” 

[1-9] 5. P. 15, reference to Fig. 3E is incorrect as Fig. 3D.

[1-9]. Response: Thank you this has been corrected. 

[1-10] 6. Fig. 2A, bottom, truncations is mis-typed as “fruncations”.

[1-10]. Response: Thank you this typo is corrected. 

[1-11] 7. Fig. 2G – the blot image includes handwritten notes on the film that should be removed.

[1-11]. Response: The image has been recropped to eliminate as much writing as possible. 

[1-12] 8. Fig. 3B – Blot quality is poor for the CEN-expressed Ste5 – any quantitation of these signals is going to be unreliable. Also not best practice to present four isolated lane strips from a blot. This experiment should be repeated with a proper single gel and better immunoblotting conditions.

[1-12]. Response: The blots, I believe are quite good quality blot is the same as the full blot labeled as long exposure in the same figure as described in figure legend. I have incorporated full blots in as many places as possible and the one you are referring to with the four strips is replaced with full gel blots. I have included the densitometry which includes densitometry of different exposures. The inclusion of full blots helps reader to see they are of good quality. 

[1-13A] 9. For the opposite reason, panel 3C is massively overexposed and no densitometry or even visual comparison is possible. 

[1-13A]. Response: The panel 3C (now 4C) is now next to panel 3B (now 4B) for clarity and full gels are shown as they were in the first version. The overexposed lanes in the long exposure were not used to derive densitometry information and is shown to reveal the various Ste5-Myc9 degradation fragments and high molecular weight forms represented by the upward smear from the full length Ste5-Myc9 and aggregated Ste5-Myc9 in the gel well. This was/is explained in the results description of the figure.

[1-13B] Same for Fig. 5D.

[1-13B] Response: Overexposed bands in 5D (now 6D) were not used for densitometry. Shorter exposures were used and compared to the longer exposures but long exposure was needed to visualize the weakest bands. 

[1-14] 10. Fig. 4A – Comparing lanes 1-3 to 4-6, it seems very unlikely that the signals in the former are 4x the latter, and the pellet lanes are so low as to be non-interpretable, generating serious doubts about the usefulness of this experiment.

[1-14]. Response: The values in 4A (now 5A) represent what was measured using multiple exposures. The values may underrepresent the difference between lanes 1-3 and 4-6 but that is what was found with the signal to noise background within the film. I think it is quite safe to say there is a significant difference between wt and the ssa1 ssa2 double mutant. 

[1-15] 11. Page 16 – why do the authors refer to D-glucose, D-galactose, and then galactose? Since no one would assume L-sugars are being used this nomenclature seems unnecessary.

[1-16]. Response: The D has been deleted. 

[1-17] 12. Fig. 6 – are panels L and M the same image? Overall this figure is very confusingly presented, and the single panels without controls or isogenic strains are incomplete as shown.

[1-18]. This is now figure 7 with improved labeling and percentages that reflect new numbers based on analyzing more data and getting standard error. The experiment has quite a few controls although not everything can be displayed in the figure and is recorded in Table 3 with standard errors and p-values. The part L has two isogenic strains of the VWA quadruple mutant with excess SSA1. Now the panels are labeled as strain 1 IF for immunofluorescence and strain 2 IF and DAPI for the two smaller panels. More controls have been added to the accompanying Table 3. The correct controls are the mutant without excess Ssa1 which is shown in J and K is the same VWA domain mutant with a TAgNLS added to direct more of it to the nucleus to show again that the punctate foci are within the cytoplasmic pool rather than the nuclear pool. Another panel in this figure shows wild type Ste5. Table 3 includes these constructs and others as well as TAgNLS-GFP-GFP and TAgNLS-NES(PKI)-GFP-GFP to compare nuclear and cytoplasmic pools of another protein, with standard errors and p-values. 

[1-18] 13. Fig. 8A,B are unnecessary, at least in the main text. Panel C is uninterpretable.

[1-18] Fig8 has been rearranged and presented with greater detail for better clarity. Parts A, B and D are now SupFig2. Part C is now section C of Table 5 with standard error and p-values and an experimental description at the bottom of Table 5. I hope these changes have improved presentation. I believe the data are quite compelling, particularly at the 6 hour time point. I have tried to be more explanatory in the results section because it is an experiment that requires understanding that we are following what happens after loss of Ssa1 in a strain that only has a functional SSA3 gene. It takes a number of hours for the pool of Ssa1 to decay and be noticeably depleted. The starting strain is lacking SSA2 and SSA4 and does not efficiently form shmoos, only 33.5% at t=0. After 6 hours of decay, there is a large, statistically significant difference without Ssa1 for shmoo formation - 3.15% shmoos without Ssa1 at t= 6 hours compared to 34.55% at t=0, p-value 0.023095

[1-19] 14. Fig. 9A – plot format is unusual and inconsistent with other plotting styles. It is also well-established that ssa1∆ and ssa2∆ phenotypes are generally non-existent or at least minor compared to loss of both chaperones together. Not sure this point needs to made again and again in the manuscript. Fig. 9B – remove absolute values from chart as the relative levels are what’s most important. The authors should also consider statistical testing these differences with replicates and a t-test, at least.

[1-19] Response: I have tried not to be repetitive. Figure 9A has been converted to a chart format that matches part B with standard error instead of standard deviation. Asteriks have been added for p-values (Key: ns (not significant) >0.05, *=<0.05, **<0.01, ***0.001, ****<0.0001). Figure 9B values are relative levels, the reviewer is mistaken. A new Fig S6A has been added that has true values for two sets of data together with p-values. A new Fig S6B has been added showing normalized standard error for three sets of FUS1-lacZs for wt and ssa1 ssa2 to illustrate the characteristic of phenotypic variability that appears to be inherent to the ssa1 ssa2 double mutant and which I mentioned in the first submission. 

[1-20] 15. Fig. 9F – where is the signal for Fus3-HA? The interpretation of the data is rather convoluted by normalizing to purported kinase pulled down when the reader cannot see that.

[1-20] Response: There was a tiny amount of signal detected for the densitometry. 

[1-21] 16. Fig. 10. -panel “e” is labeled in lower case.

[1-21] Response: This has been corrected. 

…………………………………

Reviewer 2 comments have been numbered [2-#] to distinguish from Reviewer 1. Reviewer 2 comments are in original helvitica neue 12 point and my responses are in arial 12 point. Lists of Tables, Figures, SupFigures SupTables and Subheadings with first sentence are after the responses for clarity.

[2-1] Reviewer #2: Comments on manuscript titled ” Effects of HSP70 chaperones Ssa1 and Ssa2 on Ste5 scaffold and the mating mitogen-activated protein kinase (MAPK) Pathway in Saccharomyces cerevisiae”

The study focuses on Ste5 interaction with Ssa family of proteins in yeast. Using immunoprecipitation, authors find that Ste5 interacts with Ssa chaperone. Further work aims to examine the role of this interaction in Ste5 stability, and activation of other interactors of Ste5 such as Fus3 and Kss1. Though the study is of interest to the chaperone community, the experiments carried out has various limitations including lack of appropriate controls before a meaningful conclusion could be drawn. Also the manuscript is written with lots or typos, incorrect reference to figures, and in a way that its very difficult to understand in its correct form. Following are some of the comments:

[2-1] Response: I am glad the reviewer thinks the experiments are of interest to the chaperone community. With all due respect to the esteemed reviewer the comment “various limitations including lack of appropriate controls before a meaningful conclusion could be drawn. “ is not substantiated with specifics, making it not possible to respond. Nevertheless, I have modified text to emphasize controls where it may not have been obvious to the reader and have added several controls in the punctate foci Table. 

With respect to reviewer comments “Also the manuscript is written with lots or typos, incorrect reference to figures, and in a way that its very difficult to understand in its correct form.” 

There were a few typos and incorrect references to figures which have been corrected. 

I am always aided by artificial intelligence in Microsoft word that lets me know if there is a typo or phrase written with incorrect grammar. I hope I have caught them all. I have, for greater clarity and mpact, rearranged some figures and their order of presentation and done relabeling to improve presentation. I hope these modifications make the manuscript an easier read. To aid this reviewer I have listed all Figures, Tables, SupFigures, SupTables and Subsection headings and First sentence of each subsection at the end of my responses to this revieer so she/he can understand the changes and overall points of the paper in a manageable way. 

[2-2] (1) Abstract: Ssa2 is a paralog (rather than ortholog) of Ssa1 .

[2-2] Response: The word ortholog has been changed to paralog. 

[2-3] Figure 1B: “The signal transduction partners ……the wide IgG2a band”. What is the basis of identification of band as Ste7, Ste11, and Fus3 and Kss1. No relevant data is provided that confirms the identity of these bands (such as mass-spect or detection with antibodies). Further the bands itself are not clearly visibile on the Gel image. For example there is no band on the gel-image at 80kDa. May be authors should label the bands.

[2-3] Response: Yes, it is true that these bands could also be proteins other than the kinases that are known to associate with Ste5, although the kinases are obvious candidates. I have rewritten this section to clarify and have mentioned that the faint bands overlap the masses of Ssb1 and Ssb2 which are predicted to interact with Ste5. 

Rewritten text: With limits of detection in Coomassie blue being 50 ng and higher and potentially micromolar binding affinities between Ste5 and associated kinases, it is possible that other proteins that associate with Ste5-Myc9 were not visualized. Bands a, b and c are currently unknown in identity. Several signal transduction partners of Ste5 are close in size to bands “a”, “b” and “c”, including Ste7 (57,723.7 Da) and Ste11 (80,718.8 Da) whereas Fus3 (40,770.5 Da) and Kss1 (42,680.6 Da) might be obscured by the wide IgG2a band. Hsp70s Ssb1 (66,601 Da) and Ssb2 (66,594 Da) are predicted to interact with Ste5 (discussed later in results) and could also be one of the bands. 

[2-4] Figure 1B: Densitometry does not align well with the image shown. In the presence of α-factor Ste5 is still more than Hsp70 though authors mention that Ste 5 is 0.6 time of Hsp70. 

[2-4] Response: Thank you for noticing this nonalignment. I have made the numbers in the text match what is in the Supplemental Table 2. Historically, the text was written before multiple lanes were assessed for densitometry with newer operating systems and the numbers in the table are the correct ones. 

[2-4] Response: This section reads as follows: “ImageJ densitometry of the coomassie stained gel in Fig. 1B indicates a ratio of 1.45 of Ste5-Myc9:70 kDa Hsp70 –� factor and a ratio of 0.67 Ste5-Myc9: 70 kDa Hsp70 +� factor suggesting less Hsp70 associated with Ste5-Myc9 in the +� factor extract (TableS2). “

From SupTable 2:

1.Fig. 1B Relative densitometry of 

ip’d Ste5-Myc9 to co-ip’d 70 kDa band 

(normalized with IgG)

Ste5-Myc9 2� 30oC -�F 0.67

Ste5-Myc9 2� 30oC +�F 1.45

[2-5] Page number 13: “The two other bands……………due to its high abundance”. Instead it should be mentioned as “…..due to its relatively high abundance”….

[2-5]. The word “relatively” has been added. The sentence on page 13 (now page 14) now reads: Focus was placed on band “a” due to its relatively high abundance

[2-6] Page 13: “The tandem mass spectra of two……..Ssa1, Ssa2, Ssa3 and Ssa4 (Fig 1C)”. The HPLC profile should be shown. And why only two peaks were picked for analysis. In general the coverage from Mass spect should be at least more than 40-50% of the protein sequence.

[2-6] Response: The HPLC profile is now shown in Figure 1C and the accompanying details have been added to the Figure legend.

Changes to results on page 13 (now page 14) are: “Analysis of tandem mass spectra of two major HPLC peaks identified residues 170-185 (IINEPTAAAIAYGLDK) of Hsp70 isoforms Ssa1, Ssa2, Ssa3 and Ssa4 (Fig. 1C-D).”

Figure1C legend now reads: 

C. HPLC profile of tandem mass spectra for the 70 kD protein gel band after treatment with trypsin. The HPLC profile of the LCMSMS base peak trace is from time 0 to time 30 minutes for the tryptic digest of the 70 kDa band labeled as STE5-GT by Harvard Microchemistry. The LCMSMS peaks at times 23.91 minutes (STE5-GT239) and 27.01 minutes (STE5-GT270) were manually picked and subjected to manual MSMS interpretation. The T at peak 19.61 minutes is an autolyzed tryptic artefact. This analysis was performed in 1997. 

[2-7] Figure 2F: “Ste5-Myc9 also accumulates ……disappear when Cdc28 CDK is mutated”. The authors should show data in the absence of proteasomal inhibitor to conclude that the shift is because of the inhibitor.

[2-7] Response: The proteasomal inhibitor is not the only ingredient that is different from what was done in Fig1. The Materials and Methods now contain the details of buffer conditions for the various experiments. The presentation has been revised: 

“In a denaturing SDS-PAGE gel, Ste5-Myc9 migrates as a fuzzy broad band near and higher than the expected molecular weight, with multiple phosphorylated sub-bands that shift upward with � factor treatment (Fig 1B, Fig 3A, (3)). In a native gel, the broad diffuse banding pattern is more extensive and more obviously shifts to higher apparent molecular weight with � factor treatment (Fig. 3B; note that Fig 3A-B buffers contain many phosphatase and protease inhibitors including the PSI proteasome inhibitor). The upward shift of Ste5-Myc9 in the presence of � factor requires signaling through the MAPK cascade and is blocked by a ste11� (MAPKKK) mutation (Fig. 3B). High molecular weight forms of Ste5-Myc9 accumulate as a high molecular weight smear and as aggregates in the gel well when Ste5-Myc9 is overexpressed and are more pronounced in extracts from wild type (WT) cells grown at 37oC compared to room temperature (Fig. 3C). Aggregates in a gel well can occur when a protein is polyubiquitinylated or oligomerizes which occurs for Ste5 (34,37,38)...”

[2-8A] Page 15: “In the wild type……………Ste5 aggregation and degradation”. Figure is wrongly mentioned at 3D instead of 3E.

[2-8A] Response: I think the sentence the reviewer is referring to is on page 19. This sentence has been edited and reads: In the wild type control, more Ste5-Myc9 was in the gel well and fragmented at 37oC compared to at RT (Fig 4F-I, lanes 1-3,8-10) suggesting elevated temperature leads to more Ste5 aggregation and degradation.

[2-8B] Figure 3E: Lane 1 and 8 show contradictory results. In lane 8 there is no detectable Ste5 similar to as in case of Ssa1Dssa2D cells.

[2-8B] Response: The room temperature lanes 1 and 8 for the two wild type clones are reduced compared to 30oC and 37oC which is consistent. Lane 8 sample ran somewhat smeary in the gel which in this exposure makes it appear as a lower signal. 

[2-9] Densitometry analysis has been reported for many Figures, and the quantitative estimation is reported unti 2nd decimal place. How many times the experiment was carried out for the densitometry analysis to report the data? Further, error in estimation should be reported.

[2-6] Response: The standard error and number of samples are now in Table S2. 

[2-10] Page 16: “The effects are most likely posttranscriptional ………..by Ssa1, Ssa2 or Hsf1 or Msn2/Msn4”. No evidence in support of posttranscriptional effect is provided. qRT-PCR should be carried out.

[2-10] Response: We provide ample evidence of effects of Ssa1 and Ssa2 on Ste5 protein and evidence of physical association, The sentence is quite accurate. The request for additional experiments is beyond the scope of this manuscript. The editor gave instructions to make responses at the level of text changes. 

[2-11] Page 16: Figure 3H is mentioned but I could not find the Figure in the version that I received.

[2-11] Response: The correct referral is indeed 3H (it is now 4H). It is located bottom right of the figure. 

[1-12] Figure 4A: Molecular weight marker is missing.

[1-12] Response: I have added the mw marker ladder.

[2-13] Figure 5D: “The abundance of Ste5-Myc9 was several-fold less in the ssa1D ssa2D extracts compared to wild type (Fig 5D lanes 9,10) ……… Ste5-Myc9 dimerizes equivalently or slightly more in the ssa1D ssa2D extracts than the wild type extracts (Fig.5D)”. Its not clear that about which lane corresponds to Ssa1DSsa2D double deletion (does not seem to be lane 9 and 10 as reported). Further, the basis on which Ste5-Myc9 dimerization is reported, is not clear and thus should be elaborated for clarity.

[2-13] Response: Figure 5D is now Figure 6D. This section now references the relevant papers that defined oligomerization biochemically and has been improved.

[3-14] Figure 6B: “We found Ssa1-GFP to localize throughout the cytoplasm and nucleus as shown by absence of nuclear exclusion”. The colocalization study should be carried out to support this conclusion.

[3-14] Response: There was no obvious exclusion from any part of the cell, the signal was strong throughout the entire cells. Revised version: “We found Ssa1-GFP to localize throughout the cytoplasm and nucleus.”

[3-15] Page 19: “Moreover, Ssa1-GFP was clearly enriched in the nucleus in addition to being in the cytoplasm in the ssa1D ssa2D strain that lacks endogenous Ssa1 and Ssa2 (Fig 6B)” : Should it be Figure 6C-D?

[3-15] Response: The correct referral was as written, 6B. This is now Figure 7B. 

[3-16] Figure 6E and 6F: Authors claims that wt Ste5 rarely forms foci however Ste5(1-242)-GFP fusion does form. However this is not apparent from these figures as in both cases similar foci are shown.

[3-16] Response: I am now including standard error and number samples in the revised version. In this table am comparing images that were collected on a Zeiss microscope I installed in 1990. Here is what is written:

“By live cell microscopy, Ste5-YFP and GFP-Ste5 localize in numerous punctate foci using a Nikon TE2000E (24), whereas Ste5-Myc9 localizes in an irregular pattern of localization throughout the cytoplasm and nucleus by indirect immunofluorescence using a Zeiss Axioscope 2 microscope that appears mottled with punctae using a BioRad 1024 multiphoton laser confocal microscope (35). We reexamined Ste5-Myc9 and several derivatives for evidence of punctate foci using the Zeiss Axioscope 2 (see Materials and Methods for list of mutants). By indirect immunofluorescence, Ste5-Myc9 was not detected in punctate foci during vegetative growth at 30oC in either W303a or S288c strain backgrounds (Fig 7E, Table 3 % of total cells with punctate foci: W303a ste5� at 30oC: 0 +/-0 S.E. (N=4), W303a 0 +/- 0 S.E. (N=4), S288c 0 +/- 0 S.E. (N=2). Punctate foci were detected at a very low level after � factor treatment, although this was not considered a statistically significant difference by p-value (Table 3, 0.2 +/- 0.2 S.E. (N=7), p-value 0.424224). We could not quantify punctate foci for Ste5 in the In the ssa1� ssa2� strain because of low abundance (i.e. GFP-Ste5, Ste5-Myc9, and TAgNLSK128-Ste5-Myc9).”

“By contrast, it was possible to visualize punctate foci in several mutant derivatives of Ste5 using the Axioscope 2, suggesting these derivatives have a greater ability to aggregate or oligoemerize. Nearly 100% of cells expressing Ste5C180A-Myc9 mutant that can not dimerize through the RING-H2 domain exhibited punctate foci (see Fig 3 in reference 35). In addition, a Ste5(1-242)-GFP2 fusion that oligomerizes through the RING-H2 domain localized in punctate foci in the cytoplasm by live cell imaging (Fig 7F; at 30oC the % cells with punctate foci was 5.3 +/- 3.4 S.E. (N=8), Table 3). These punctate foci were due to the RING-H2 fragment because none were detected for TAgNLS-NES-GFP-GFP and TAgNLS-GFP-GFP fusion proteins that localize to the nucleus and cytoplasm (Table 3). A Ste5-GST fusion that forms more dimers from GST also formed readily detectable punctate foci during vegetative growth (Table 3, 1.4% +/-1.0 S.E. (N=5)) and after � factor stimulation (1.8 +/- 1.8 S.E. (N=3)) by indirect immunofluorescence with anti-GST antibodies (Table 3). Strikingly, we also found that a VWA domain quadruple point mutant, ste5L610/614/634/637A-Myc9 (Fig 7G-I) formed more punctate foci in the cytoplasm ( Fig 7J, Table 3, punctate foci in 1.4 +/- 1.4 S.E. percent of cells (N=3) , p-value .014993, Table 3). A nucleus-enriched derivative, TAgNLS-ste5L610/614/634/637A-Myc9 formed punctate foci in 2 +/-2 S.E.% of cells (N=2) (Fig 7K, p-value .078141, Table 3) and a GST-ste5L610/614/634/637A derivative formed punctate foci in 8.7% of cells (N=1) (Table 3). Thus, the ste5L610/614/634/637A mutations in the VWA-like domain increases the propensity of Ste5 to localize in punctate foci. 

[3-17] Figure 7B is a table. Please show images also.

[3-17] Response: The PLOS ONE guidelines require images of immunoblots and gels and these have been supplied. 

[3-18] Page 25: “Thus, Ssa1 and Ssa2 are essential for basal activation of Fus3 and full basal activation of Kss1 and essential for a factor induced activation of both Fus3 and Kss1. Ssa1 and Ssa2 are therefore required for efficient response to mating pheromone in addition to regulating Ste5 abundance and integrity”. – Which data authors are refering to? More clarity is required to support this conclusion.

[2-18] Response: Thank you for pointing this out. I have improved the presentation as shown below: 

[2-19] There are lots of typos such as “Ssa1 Is overexpressed” instead of “Ssa1 is overexpressed”. Page 13: “…….excised from the ge….”

I could not find Table 1-4 in the version I received.

[2-19] Response: I have checked for typos. I continually make use of the Microsoft word artificial intelligence which tracks typos and grammar and did not find “lots of typos”, although I did find some in addition to the ones you pointed out. I have read the manuscript and checked for typos and corrected those I have found. There are no typos in the two sentences you have written. i.e. “are lots of typos such as “Ssa1 Is overexpressed” instead of “Ssa1 is overexpressed”. Page 13: “…….excised from the ge….”

To assist reviewer 2, I include here a LISTING OF FIGURES, TABLES, SUPFIGURES, SUPTABLES with changes:

The supporting data includes Fig S1, Fig S2, Fig S3, Fig S4, Fig S5, Fig S6, Fig S7, Fig S8, and Table S1, Table S2, Table S3, Table S4, Table S5, Table S6, Table S7, Table S9. This resubmission does not have a Table S9 because information was consolidated (no data were removed). Reviewer 2 mentioned an inability to find all of the supporting materials, I hope she/he is able to do so now.

LISTS of Specific changes to the manuscript figures and tables: 

Figure 1: Immunoprecipitated Ste5-Myc9 is bound to Hsp70 Ssa1 and possibly Ssa2. New Part C Inclusion of requested HPLC data used for mass spectrometry with details in the legend of figure 1. 

Figure 2. Ste5 co-immunoprecipitates with Ssa1-GFP and vice versa. This figure now has parts A-E with movement of several parts to Figure 3 so less crowded (in response to reviewer).

Figure 3. Ste5-Myc9 migrates in polyacrylamide gels as full-length, high molecular weight, aggregated and truncated forms of varying proportions in ste5�, ste5� ste11�, cdc28-4 and SSA1-GFPOP strains. New Figure 3 of panels that were in Fig2. Parts C and D show full gels of whole cell extracts in co-ips in Figure 2. 

Figure 4. The abundance of Ste5 is lower in a ssa1� ssa2��double mutant. This figure has been rearranged to clarify with a new part E in response to reviewer comments. 

Figure 5. N-terminal truncations of Ste5-Myc9 accumulate in the ssa1� ssa2� double mutant. Figure 5 now has part E added in addition to part A, arrangement of figure parts in response to reviewer comments with improved labeling. 

Figure 6. N- and C-terminal tagged forms of Ste5 from ssa1� ssa2� strains have enhanced antibody accessibility. Figure 6A,B,C,D,E has had its labeling improved. 

Figure 7. Ssa1 and Ssa2 alter the localization of wild-type and mutant Ste5. Figure 7 parts G-I has a ball and stick version of the crystal structures i for better clarity. This figure has had its labeling improved and % of punctate foci adjusted after further quantitation for standard error values. The neglibible adjustment in percentages does not change interpretation. 

Figure 8. Effect of ssa1� ssa2� mutations on activation of Kss1 and Fus3. This figure was previously Figure 10. It has three new panels A, B, E. Panel A shows Kss1 levels with anti-Kss1 antibody with the full blot rather than a few lanes and reveals that Craig lab diploid ascospores are kss1- whereas the bar�� derivatives we constructed through crosses are KSS1+. A request for controls by reviewer is solved by this panel and the two other panels that include W303a WT, kss1� and fus3��control strains. Panel B is an additional anti-active MAPK blot with controls at higher a factor concentration. Panels C and D are unchanged and panel E is a new anti-active MAPK blot.

Figure 9. Effect of ssa1� ssa2� mutations on Fus3 abundance, Ste7 feedback phosphorylation and Fus3 kinase activity. Figure 9 has been rearranged. Fig 9A is Ste7M, Fig 9B is Fus3 immunoblot, Fig 9C is Fus3-HA#5 and fus3K42R-HA#5 kinase activity, Fig 9D is Fus3-HA#5 and fus3R42-HA#5 abundance. 

Figure 10. ssa1� ssa2� double mutants have reduced FUS1::ubiYlacZ activity and G1 arrest but can still mate. Figure 10 was previously Figure 8. Figure 10 part A FUS1::UbiY-lacZ graph was put into excel. Parts A and B FUS1::UbiY-lacZ have standard errors instead of standard deviation. P-values have been added to part B per reviewer request. A new companion to A and B is figure S6 which has normalized standard errors from Fig10A-B to highlight large phenotypic variability of the ssa1� ssa2��double mutant. The halo assays have been cropped to make space and the Fus3HA kinase assay has been moved to Fig 9. Labels have been improved. The morphology tally that was previously in this figure has been moved to Table 2. 

Figure 11. ssa1� ssa2� double mutants and several HSP70-HSP90 network mutants are defective in shmoo formation. A-F are images of WT and ssa1 ssa2 cells as before. Parts G-O are new fields of cells of WT S288c and Hsp70 chaperone and HSP70-HSP90 network mutants. 

Table I now has standard errors and p-values (students t-test, 2 independent means, 2 sided). 

Table 2 now has standard errors and p-values 

Table 3 now has standard errors and p-values. 

Table 4 is unchanged

Table 5 has standard errors and p-values and part C morphology that was previously in a figure. 

Table 6 is a new table that summarizes shmoo formation in the various hsp mutants and has Fisher’s exact test p-values. In some instances more cells were tallied for greater power. 

Fig S1 is unchanged.

Fig S2 alignment is new. It has parts that were previously in the main figures. 

Fig S3 is unchanged (was previously Fig S2)

Fig S4 is unchanged (was previously Fig S3)

Fig S5 is unchanged (was previously Fig S4)

Fig S6 is new figure showing A-B larger cropping and long and short exposures of the immunoblot from Fig 8A and in C normalized standard error of FUS1::ubiY-lacZ values

Fig S7 is unchanged

Fig S8 is unchanged

Table S1 has several additions shown in yellow highlight. 

Table S2 has standard error and p-values and improved presentation

Table S3 is unchanged (was Table S4)

Table S4 has been edited to remove several duplications in the list of proteins (was Table S5) in first submission

Table S5 has been updated to include putative physical interaction data on HSP90 network proteins including Hsp82, Hsc82, Cdc37, Sba1 and grouping of core (mating/invasive growth signal transduction protein interactions is included for each HSP network protein listed (Table S5 was Table S6 in first submission) 

Table S6 is unchanged (was Table S7). 

Table S7 is unchanged (was Table S8) 

Scroll up earlier in this response for subsection headings and first sentences.

---

## [Editor Report · Decision Letter 1]

18 Jul 2023

Effects of HSP70 chaperones Ssa1 and Ssa2 on Ste5 scaffold and the mating mitogen-activated protein kinase (MAPK) Pathway in Saccharomyces cerevisiae

PONE-D-22-25493R1

Dear Dr. Elion,

We’re pleased to inform you that your manuscript has been judged scientifically suitable for publication and will be formally accepted for publication once it meets all outstanding technical requirements.

Thank you for submitting this work to PLoS ONE!

Best Regards,

Katherine A. Borkovich, Ph.D.

Academic Editor

PLOS ONE
---

## [Editor Report · Acceptance letter]

18 Sep 2023

PONE-D-22-25493R1 

Effects of HSP70 chaperones Ssa1 and Ssa2 on Ste5 scaffold and the mating mitogen-activated protein kinase (MAPK) Pathway in *Saccharomyces cerevisiae*

Dear Dr. Elion:

I'm pleased to inform you that your manuscript has been deemed suitable for publication in PLOS ONE. Congratulations! Your manuscript is now with our production department. 

Kind regards, 

on behalf of

Dr. Katherine A. Borkovich 

Academic Editor

PLOS ONE